# TE-VLM: Transfer Entropy for Vision Language Model Distillation

## Abstract

Transfer Entropy (TE) is a principled measure of directed information flow, but its direct estimation in high-dimensional multimodal representation spaces is computationally prohibitive. In this work, we propose a practical distillation framework that replaces direct TE estimation with TE-inspired proxy regularization for multimodal vision–language models. Our method introduces proxy objectives that reward student representations for preserving teacher-aligned predictive structure across modalities, while remaining compatible with standard contrastive distillation losses. We instantiate the framework in CLIP-style teacher–student distillation across multiple teacher backbones, including CLIP RN50, ViT-B/16, and RN50×16, and evaluate it on MSCOCO 2014, Flickr8k, Flickr30k, Food-101, and ImageNet-1k. Across retrieval experiments, the proposed TE-inspired objective consistently improves Image-to-Text performance over MI-based and standard distillation baselines, while remaining competitive on Text-to-Image retrieval. Additional recipe-level diagnostics across temperature and batch size show that these gains are reproducible and are not explained solely by a favorable training recipe. Representation-level analyses further show that TE-inspired distillation yields stronger teacher–student agreement in local neighborhood structure, cosine alignment, and joint image–text embedding geometry. Beyond in-dataset evaluation, cross-dataset retrieval from MSCOCO to Flickr8k shows that the proposed objective better preserves transferable multimodal structure under distribution shift. We also observe improvements in zero-shot classification on Food-101 and out-of-dataset evaluation on ImageNet-1k. Together, these results suggest that TE-inspired proxy regularization provides an effective and scalable mechanism for preserving teacher-consistent cross-modal structure during multimodal distillation.

## 1 Introduction

Vision–Language Models (VLMs) have emerged as a powerful framework for learning joint representations of images and text, enabling applications such as image captioning, visual question answering, and cross-modal retrieval (Radford et al., 2021; Jia et al., 2021). However, state-of-the-art VLMs are often computationally expensive, making them impractical for deployment in resource-constrained environments. To address this challenge, knowledge distillation (Hinton et al., 2015) is widely used to transfer knowledge from a large teacher model to a smaller, more efficient student model while preserving performance.

Existing approaches to VLM distillation primarily rely on contrastive learning (Li et al., 2022; Yang et al., 2024) and divergence-based objectives such as Kullback–Leibler (KL) divergence (Li et al., 2024b) to align the student with the teacher. These objectives are effective for matching embeddings, similarity scores, or batch-level distributions at a given optimization step, but they do not directly emphasize whether the student's *updates* move in teacher-consistent directions. This motivates an information-theoretic perspective in which the teacher is viewed not only as a source of target representations, but also as a source of guidance for how the student's multimodal representation geometry should evolve during training.

In this work, we use *transfer entropy* (TE) (Schreiber, 2000) as a conceptual starting point for this perspective. At a high level, optimization-step TE provides a way to describe directed information flow from the

teacher to the student's next-step state beyond what is already explained by the student's current state. This framing is appealing because it captures the intuition that successful distillation should not only reduce static discrepancies, but should also encourage teacher-informed student updates. However, directly instantiating optimization-step TE in high-dimensional CLIP-style embedding spaces is intractable, since it would require estimating conditional mutual information over cross-step joint variables.

Accordingly, our contribution is *not* a direct estimator of optimization-step TE. Instead, we develop a tractable *TE-inspired proxy framework*. We first show that, under a first-order linear–Gaussian approximation for a perturbation-defined one-step setting, the resulting signal is related to cosine alignment between teacher and student Jacobians. We then avoid explicit Jacobian construction by introducing two efficient cosine-based surrogates computed from within-batch finite differences on already-available embeddings: TE1, which measures per-modality alignment, and TE2, which measures joint multimodal alignment. These objectives are lightweight to compute and are the quantities actually optimized in our training algorithm. Therefore, the experiments in this paper primarily validate the effectiveness of these final proxy objectives for VLM distillation, rather than a direct implementation or empirical verification of the ideal optimization-step TE quantity itself.

This distinction is important for correctly interpreting our claims. We use optimization-step TE as a principled conceptual framing for why teacher-consistent update geometry may matter, while the practical method is a geometric proxy objective derived through successive approximation steps. Empirically, we show that these proxies are useful regularizers: they complement standard distillation losses by encouraging the student to align local directional changes in embedding space with those of the teacher. Across retrieval, classification, cross-dataset transfer, and representation-level analyses, the resulting students exhibit stronger teacher–student geometric agreement and improved generalization. Additional recipe-level diagnostics across temperature and batch size further show that these gains are reproducible and are not solely explained by a favorable training recipe.

The key contributions of this work are as follows:

- We introduce a *transfer-entropy-inspired* perspective for VLM distillation that motivates teacher-guided student update geometry, while explicitly distinguishing the ideal optimization-step TE framing from the tractable proxy objective used in practice. This clarifies the conceptual role of TE in our framework and helps align the theoretical motivation with the implemented training objective.

- We show that, under a first-order linear–Gaussian approximation in a perturbation-defined setting, the resulting TE-related signal leads to a computable surrogate based on cosine similarity between teacher and student Jacobians. This provides a principled bridge from the information-theoretic motivation to a practical geometric criterion that can be optimized efficiently.

- We propose two efficient cosine-based proxy objectives, TE1 and TE2, implemented via within-batch finite differences to capture teacher-consistent geometric alignment in high-dimensional multimodal embedding spaces. These proxy objectives avoid explicit Jacobian construction, introduce little additional computational overhead, and are readily compatible with standard CLIP-style distillation pipelines.

- We demonstrate that adding these TE-inspired proxy regularizers to the distillation objective improves retrieval performance over strong baselines including contrastive, KL-divergence, Mean Squared Error (MSE), Interactive Contrastive Learning (ICL), and Mutual Information (MI)-based objectives. The gains are especially consistent for image-to-text retrieval, while remaining competitive for text-to-image retrieval across multiple teacher–student settings.

- We provide extensive empirical evaluation on MSCOCO 2014, Flickr8k, Flickr30k, Food-101, and ImageNet-1k, including recipe-level diagnostics across temperature and batch size, cross-dataset retrieval from MSCOCO to Flickr8k, and representation-level analyses of teacher–student embedding geometry. Together, these experiments show that the proposed proxy objectives are effective, stable, and computationally lightweight, and that they help preserve more transferable multimodal structure during VLM distillation.

## 2 Related Work

### 2.1 Knowledge Distillation

Knowledge distillation enables the transfer of learned representations from a large teacher network to a smaller student model (Hinton et al., 2015). Building on this idea, intermediate feature supervision has been proposed to guide deeper yet more efficient networks (Romero et al., 2015). Other approaches emphasize spatial attention alignment (Zagoruyko & Komodakis, 2017), or address scenarios where original training data is unavailable through data-free distillation (Huang & Wang, 2017).

Further research has focused on aligning internal representations between teacher and student networks. Contrastive objectives harmonize feature spaces (Tian et al., 2020), while attention-based strategies have been tailored for transformer architectures (Touvron et al., 2021). Information-theoretic methods have also been explored, including maximizing mutual information (Ahn et al., 2019) and modeling inter-sample relationships (Park et al., 2019). Another line of work distills the teacher's probability distribution to improve student generalization (Passalis & Tefas, 2018). Mutual-relation distillation techniques have also been applied to face recognition, as demonstrated in CoupleFace (Liu et al., 2022). In multimodal representation learning, Chen et al. (2023) proposed an objective that preserves mutual information between a teacher model and an auxiliary modality to improve distillation.

More recently, several studies have examined frequency-domain representations. Frequency attention modules enable students to adapt feature responses under teacher guidance (Pham et al., 2024), semantic frequency prompts have been leveraged for dense prediction tasks (Zhang et al., 2024), and frequency-aware optimization has also been used to construct compact synthetic datasets (Shin et al., 2023).

Other relevant contributions include self-distillation for general-purpose text embeddings (Chen et al., 2024), sample-efficient distillation through synthetic training data (Liu et al., 2024), dual-teacher frameworks (Li et al., 2024c), orthogonal-projection-based transfer (Miles et al., 2024), and automated search strategies for distillation pipelines in object detection (Li et al., 2024a).

### 2.2 Vision–Language Model Distillation

In the vision–language domain, early work aligned object semantics with textual descriptions to improve cross-modal understanding (Li et al., 2020), while large-scale pre-training frameworks such as UNITER learned universal image–text representations through joint optimization (Chen et al., 2020). More recently, research has increasingly focused on compressing and distilling large vision–language models to obtain efficient students while preserving strong multimodal alignment. For example, Fang et al. (2021) explored model compression for multimodal encoders, Dai et al. (2022) distilled vision–language knowledge to enable multimodal generation, and Wang et al. (2022) refined lightweight CLIP-style models for downstream tasks. TinyCLIP (Wu et al., 2023) further demonstrated that affinity mimicking and weight inheritance can effectively shrink CLIP models while maintaining competitive zero-shot performance.

Distillation has also been leveraged to enhance multimodal reasoning and transferability. Instruction-tuning frameworks distill higher-level reasoning ability from strong multimodal teachers (Hu et al., 2024), and multimodal knowledge transfer has been applied to enable open-vocabulary object recognition (Gu et al., 2021). In parallel, frequency-aware distillation strategies have been proposed to improve robustness under distribution shift (Li et al., 2023), and vision–language representations have been used to strengthen classification across diverse domains (Addepalli et al., 2024).

Very recent work continues to expand the scope and structure of VLM distillation. Mixture-of-Visual-Encoder Knowledge Distillation (MoVE–KD) aggregates complementary information from multiple teacher encoders into a single student model (Cao et al., 2025), while Align–KD encourages students to capture cross-modal interactions earlier in the representation hierarchy (Feng et al., 2025). Several objective functions have also been proposed specifically for CLIP-style distillation: Yang et al. (2024) introduced interactive contrastive learning and MSE alignment, and Li et al. (2024b) employed KL-based distribution matching. Our work is complementary to these formulations, but differs in that we adopt a *transfer-entropy-inspired* perspective and develop tractable geometric proxy objectives that encourage teacher-consistent alignment of student

representation dynamics across modalities. In particular, rather than directly estimating optimization-step transfer entropy, our method optimizes efficient cosine-based proxy rewards computed from within-batch finite differences.

## 3 Introduction to Transfer Entropy

Transfer Entropy is an information-theoretic measure introduced by Schreiber (Schreiber, 2000) to quantify the directed transfer of information between two stochastic processes. It is particularly useful for detecting asymmetrical interactions and causal relationships, as it measures the influence that the past of one process, $X$, has on the future of another process, $Y$, beyond what can be explained by the past of $Y$ alone.

For two discrete-time stochastic processes $X(t)$ and $Y(t)$, the transfer entropy from $X$ to $Y$ is formally defined as (Schreiber, 2000):

$$T_{X \to Y} = \sum_t p(y_{t+1}, y_t, x_t) \log \frac{p(y_{t+1} \mid y_t, x_t)}{p(y_{t+1} \mid y_t)}, \tag{1}$$

where $p(\cdot)$ represents probability distributions of the respective random variables.

Transfer entropy is closely related to conditional mutual information. It can be rewritten as the conditional mutual information between $Y_{t+1}$ and $X_t$, conditioned on $Y_t$ (Shahsavari Baboukani et al., 2020):

$$T_{X \to Y} = I(Y_{t+1}; X_t \mid Y_t), \tag{2}$$

where $I(Y_{t+1}; X_t \mid Y_t)$ is the mutual information between $Y_{t+1}$ and the history of $X$, conditioned on the history of $Y$. The proof is provided in Appendix A. This formulation reveals that transfer entropy measures the additional information that $X_t$ provides about the future state $Y_{t+1}$, over and above the information provided by $Y$'s own history $Y_t$. In Appendix B, we present an overview of prior work on mutual information and TE estimation.

## 4 Method

### 4.1 Why Is Transfer Entropy Beneficial for VLM Distillation

In VLM distillation, the goal is not only to match the teacher's representations, but also to encourage the student's updates to move in teacher-consistent directions across both modalities. Transfer Entropy (TE) provides an information-theoretic *conceptual framing* for this idea: at the level of optimization dynamics, it describes how much the teacher could explain the student's *next-step* representations beyond what is already explained by the student's current state. However, this optimization-step TE is not directly tractable in high-dimensional VLM embedding spaces. Accordingly, in this work we do not directly estimate or optimize that ideal quantity; instead, we use it to motivate efficient geometric proxy objectives that are implemented through cosine-based alignment of teacher and student finite-difference directions.

**Motivation for teacher–student directional alignment.** The purpose of the TE-inspired proxy is not to replace standard knowledge distillation objectives, but to add a complementary geometric constraint. Conventional KD losses encourage the student to match the teacher's output distribution, embeddings, or batch-level similarity structure. However, in CLIP-style retrieval, performance depends not only on pointwise embedding agreement, but also on the relative geometry of the joint image–text space. A student with limited capacity may match individual teacher embeddings or similarity scores while still distorting local neighborhoods and pairwise directions. Our proxy addresses this by encouraging teacher–student alignment of finite-difference directions: for paired examples, the student representation change is encouraged to be parallel to the corresponding teacher representation change. Thus, the enforced property is local geometric alignment rather than causal information transfer. We view this as a teacher-consistent relational regularizer that helps preserve the teacher's neighborhood and ranking structure, which is important for retrieval and transfer.

**What properties are enforced by the proxy.** We emphasize that the proposed TE-inspired proxy does not directly enforce causal information flow in the strict information-theoretic sense. Instead, the implemented TE1/TE2 objectives enforce teacher–student alignment of local representation changes. Specifically, for paired examples within a mini-batch, the proxy encourages the finite-difference direction of the student embedding space to be parallel to the corresponding finite-difference direction of the teacher embedding space. Thus, the property optimized in practice is teacher-consistent local geometric alignment: TE1 measures this alignment separately for image and text modalities, while TE2 measures joint multimodal alignment after concatenating image and text directional changes. This alignment can be interpreted as encouraging the student to preserve the teacher's local neighborhood and ranking structure, rather than as estimating exact transfer entropy or establishing causal information transfer.

**Notation.** For a mini-batch $\mathcal{B}$, let $(U^{(T)}, V^{(T)})$ denote the teacher's (frozen) text and image representations on $\mathcal{B}$, and let $(U_t^{(S)}, V_t^{(S)})$ and $(U_{t+1}^{(S)}, V_{t+1}^{(S)})$ denote the student's text and image representations on the *same* mini-batch $\mathcal{B}$ at optimization steps $t$ and $t+1$, respectively.

**Transfer entropy across optimization steps.** We define the teacher→student TE on $\mathcal{B}$ at step $t$ as the conditional mutual information

$$T_{\text{T}\to\text{S}}(t; \mathcal{B}) = I\left( \left(U_{t+1}^{(S)}, V_{t+1}^{(S)}\right); \left(U^{(T)}, V^{(T)}\right) \Big| \left(U_t^{(S)}, V_t^{(S)}\right)\right), \tag{3}$$

where $I(\cdot; \cdot \mid \cdot)$ is conditional mutual information. Intuitively, $T_{\text{T}\to\text{S}}(t; \mathcal{B})$ captures how much *additional* predictive information the teacher provides about the student's updated representations at step $t+1$, beyond what is already contained in the student's current representations at step $t$.

A high TE indicates that the teacher is meaningfully shaping the student's refinement from $(U_t^{(S)}, V_t^{(S)})$ to $(U_{t+1}^{(S)}, V_{t+1}^{(S)})$. This is especially expected early in distillation, when the student representations are underdeveloped and the teacher's signals provide substantial guidance. Tracking TE over training therefore provides a principled diagnostic of whether and when the student continues to benefit from the teacher, and whether the two modalities receive balanced guidance.

**Clarification on "time" and practicality.** One might worry that (3) assumes temporal correlation within a shuffled mini-batch. We emphasize that the index $t$ in (3) is not wall-clock time and does not refer to temporally correlated samples (e.g., videos). Instead, $t$ denotes the student's optimization step and (3) serves as a conceptual information-flow diagnostic.

At CLIP-scale dimensions, directly estimating (3) is intractable because it would require high-dimensional density estimation over cross-step joint variables. In practice, we therefore introduce the TE proxies in Section 4.2, which are tractable geometric surrogates motivated by Theorem 1. The proxy computation does not require any additional teacher evaluations beyond standard distillation: the teacher forward is computed once per mini-batch (without gradients) and reused to form within-batch finite-difference directions. TE then adds only lightweight difference and cosine-alignment computations on the current batch embeddings (no cache-and-replay and no extra forward/backward passes). Such information-theoretic perspectives on learning dynamics are consistent with prior work (Goldfeld et al., 2019; Achille & Soatto, 2018).

## 4.2 TE-Inspired Proxies via Cosine Similarity

In VLM distillation, directly estimating transfer entropy is challenging because image and text representations are high-dimensional. For example, CLIP ResNet-50 produces 1024-dimensional embeddings (Radford et al., 2021). Since transfer entropy can be written as conditional mutual information (Eq. 2), evaluating $T_{X\to Y}$ requires estimating the joint density $p(y_{t+1}, y_t, x_t)$ and the conditionals $p(y_{t+1} \mid y_t, x_t)$ and $p(y_{t+1} \mid y_t)$. When $x_t$ and $y_t$ are continuous representations with $D \approx 10^3$, this becomes a high-dimensional density-estimation problem: even coarse discretization scales exponentially with dimension (the curse of dimensionality) (Köppen, 2000), and common TE/CMI estimators either rely on local neighborhood statistics (whose sample complexity grows rapidly in $D$) or impose parametric assumptions that are difficult to justify in deep feature spaces (Shahsavari Baboukani et al., 2020; Gowri et al., 2025). Moreover, TE depends

on *joint* dependencies among $(y_{t+1}, y_t, x_t)$; capturing these interactions often entails estimating dependency structures with $\Theta(D^2)$ parameters (e.g., covariances or precision matrices) and expensive linear-algebra operations, or performing high-dimensional neighbor searches that become unreliable as $D$ increases. Collectively, these factors make direct TE estimation prohibitive for CLIP-scale embeddings, motivating the efficient TE-inspired proxies we introduce next. We therefore introduce the following theorem as the theoretical basis for the TE-inspired geometric proxy objectives used in our method.

**Theorem 1** (First-order TE–Jacobian relation)**.** *Let $x \in \mathbb{R}^d$ be an input (image–caption pair), and let $f_T, f_S : \mathbb{R}^d \to \mathbb{R}^D$ denote the teacher and student encoders with Jacobians*

$$J_T(x) \;=\; \nabla_x f_T(x), \quad J_S(x) \;=\; \nabla_x f_S(x) \in \mathbb{R}^{D \times d}. \tag{4}$$

*Under a first-order linear–Gaussian approximation of the conditional mutual information, the one-step transfer entropy from the teacher to the student satisfies*

$$T_T^S(x) \;\propto\; \cos\big(\widetilde{J}_S(x),\, \widetilde{J}_T(x)\big), \tag{5}$$

*where the Frobenius-normalized Jacobians are*

$$\widetilde{J}_S(x) = \frac{J_S(x)}{\|J_S(x)\|_F}, \qquad \widetilde{J}_T(x) = \frac{J_T(x)}{\|J_T(x)\|_F}, \tag{6}$$

*and $\cos(A, B) = \langle A, B \rangle_F$ denotes the cosine similarity (Frobenius inner product) between matrices $A$ and $B$.*

In Appendix C, we provide the proof for this theorem.

In practice, we approximate the Jacobian actions using finite differences (Nocedal & Wright, 1999; Baydin et al., 2018):

$$J_S\, \delta\mathbf{x} \rightsquigarrow f_S(\mathbf{x} + \delta\mathbf{x}) - f_S(\mathbf{x}), \qquad J_T\, \delta\mathbf{x} \rightsquigarrow f_T(\mathbf{x} + \delta\mathbf{x}) - f_T(\mathbf{x}),$$

where $\delta\mathbf{x}$ is a small input perturbation. Based on this first-order approximation, we propose two cosine-based TE-inspired proxy objectives.

**Ideal TE vs. our TE proxy.** Equation (3) defines an ideal teacher→student transfer-entropy quantity across optimization steps: it measures how much information the teacher's representation $(U^{(T)}, V^{(T)})$ transfers to the student's next-step state $(U_{t+1}^{(S)}, V_{t+1}^{(S)})$, beyond what is already explained by the student's current state $(U_t^{(S)}, V_t^{(S)})$. At CLIP-scale dimensions, directly estimating this conditional mutual information is intractable, as it would require high-dimensional density estimation over cross-step joint variables. Accordingly, Eq. (3) should be understood as the conceptual target (the teacher drives the student update), while our method uses a tractable geometric proxy motivated by first-order local approximation. In particular, Eq. (5) characterizes a perturbation-defined one-step teacher→student TE and motivates a cosine-based proxy via Jacobian alignment (proved in Appendix C). In our implementation (Algorithm 1), we instantiate this proxy using input-induced finite differences within the current mini-batch for both teacher and student.

**When is this alignment meaningful?** The TE proxies reward teacher-consistent geometry in representation space: for paired examples within a batch, they encourage the student's directional changes between examples to align with the corresponding teacher directional changes. This provides a relational learning signal that complements standard matching losses by transferring local structure of the teacher embedding space to the student. Throughout, we refer to Eq. (5) and TE1/TE2 as TE proxies rather than the exact optimization-step TE of Eq. (3).

**From Theorem 1 to practical TE proxies.** Theorem 1 provides a geometric characterization of a perturbation-defined one-step teacher→student transfer entropy, where the state transition is induced by a small input perturbation $x \mapsto x + \delta x$ (Appendix C). Under a first-order linear–Gaussian approximation, this quantity satisfies $T_T^S(x) \propto \cos(\widetilde{J}_S(x), \widetilde{J}_T(x))$. In contrast, Eq. (3) defines an ideal TE across optimization steps (parameter updates), which is generally intractable to estimate directly in high-dimensional feature

spaces. We therefore use the Jacobian-alignment signal from Theorem 1 as an efficient TE proxy without claiming to directly estimate the optimization-step TE in Eq. (3).

In practice, we avoid explicit Jacobian construction by using finite differences on representations. Within a mini-batch $\mathcal{B} = \{\mathbf{x}_i\}_{i=1}^{B}$, the index $i$ enumerates examples and is not treated as time; thus our proxy should not be interpreted as estimating temporal TE from shuffled batches. Instead, we form finite-difference directional changes across paired inputs within the same mini-batch: $\Delta f_T(\mathbf{x}_i) := f_T(\mathbf{x}_{i'}) - f_T(\mathbf{x}_i)$ and $\Delta f_S(\mathbf{x}_i) := f_S(\mathbf{x}_{i'}) - f_S(\mathbf{x}_i)$ for paired examples $(\mathbf{x}_i, \mathbf{x}_{i'})$. Under local linearization, these differences serve as practical surrogates for Jacobian actions, $\Delta f(\mathbf{x}) \approx J(\mathbf{x})(\mathbf{x}_{i'} - \mathbf{x}_i)$. Cosine similarity between teacher and student differences therefore provides a tractable estimator of the Jacobian-alignment term suggested by Theorem 1.

We refer to the resulting cosine-based objectives as TE1 (Section 4.2.1) and TE2 (Section 4.2.2), and treat them as geometric TE proxies motivated by Theorem 1 and implemented in Algorithm 1.

### 4.2.1 TE-Inspired Proxy via Cosine Similarity of Differences

Let $\mathbf{v}^{(S)}$ and $\mathbf{u}^{(S)}$ denote the image and text embeddings from the student model, and $\mathbf{v}^{(T)}$ and $\mathbf{u}^{(T)}$ denote the corresponding embeddings from the teacher model. The TE-inspired proxies are based on computing cosine similarity between local finite-difference directions in the student and teacher embedding spaces on the same mini-batch. Since the teacher is frozen, any variation in teacher embeddings arises from changing inputs, not from training-time dynamics.

To construct the TE-inspired proxy, we first compute differences between adjacent embeddings in the internally permuted mini-batch for both image and text modalities. The embedding differences are computed as

$$\Delta \mathbf{v}_i^{(S)} = \mathbf{v}_{i+1}^{(S)} - \mathbf{v}_i^{(S)}, \quad \Delta \mathbf{v}_i^{(T)} = \mathbf{v}_{i+1}^{(T)} - \mathbf{v}_i^{(T)} \tag{7}$$

for images, and

$$\Delta \mathbf{u}_i^{(S)} = \mathbf{u}_{i+1}^{(S)} - \mathbf{u}_i^{(S)}, \quad \Delta \mathbf{u}_i^{(T)} = \mathbf{u}_{i+1}^{(T)} - \mathbf{u}_i^{(T)} \tag{8}$$

for text embeddings.

Here, the index $i$ enumerates examples within a mini-batch and does not represent time. We form differences between pairs of examples in the *same* batch (e.g., adjacent elements after an internal random permutation) and use TE1/TE2 as *geometric* finite-difference surrogates, not as temporal TE estimators from shuffled data. Thus, TE1 and TE2 are geometric surrogates that encourage the student to match the teacher's local embedding-space geometry, and do not assume temporal structure in the data ordering.

Once the differences are obtained, the next step is to compute the cosine similarity between the student's and teacher's directional changes. Cosine similarity (Xia et al., 2015) serves as a measure of alignment between the two models, so that closer agreement indicates stronger teacher-consistent geometric behavior in the student representations. The cosine similarity for images is given by

$$\cos \theta_i^{(v)} = \frac{\langle \Delta \mathbf{v}_i^{(S)}, \Delta \mathbf{v}_i^{(T)} \rangle}{\|\Delta \mathbf{v}_i^{(S)}\| \|\Delta \mathbf{v}_i^{(T)}\| + \epsilon}, \tag{9}$$

where $\epsilon$ is a small constant to prevent division by zero. While for text embeddings, it is given by

$$\cos \theta_i^{(u)} = \frac{\langle \Delta \mathbf{u}_i^{(S)}, \Delta \mathbf{u}_i^{(T)} \rangle}{\|\Delta \mathbf{u}_i^{(S)}\| \|\Delta \mathbf{u}_i^{(T)}\| + \epsilon}. \tag{10}$$

To obtain a modality-level TE-inspired proxy score, the method computes the mean cosine similarity across all paired differences in the batch. The image-based TE is computed as

$$\text{TE}_{\text{img}} = \frac{1}{B-1} \sum_{i=1}^{B-1} \cos \theta_i^{(v)}, \tag{11}$$

while the text-based TE follows the same formulation:

$$\text{TE}_{\text{txt}} = \frac{1}{B-1} \sum_{i=1}^{B-1} \cos \theta_i^{(u)}. \tag{12}$$

The final TE approximation (TE1) is obtained by averaging the image and text TE values:

$$\text{TE1} = \frac{1}{2} \left( \text{TE}_{\text{img}} + \text{TE}_{\text{txt}} \right). \tag{13}$$

### 4.2.2  TE-Inspired Proxy via Cosine Similarity of Concatenated Differences

An alternative TE-inspired proxy combines the directional changes from both image and text modalities before computing cosine similarity. In this method, we first calculate the differences between consecutive embeddings for both modalities, same as (7)(8). Instead of computing the cosine similarity for each modality independently and then averaging the results, we concatenate the difference vectors from both modalities into a single vector. That is, for each index $i$, we define the concatenated difference vectors as

$$\Delta \mathbf{c}_i^{(S)} \;=\; \left[ \Delta \mathbf{v}_i^{(S)} \,\|\, \Delta \mathbf{u}_i^{(S)} \right], \tag{14}$$

$$\Delta \mathbf{c}_i^{(T)} \;=\; \left[ \Delta \mathbf{v}_i^{(T)} \,\|\, \Delta \mathbf{u}_i^{(T)} \right], \tag{15}$$

where $\|$ denotes concatenation along the feature dimension.

The cosine similarity between the concatenated difference vectors is then computed as

$$\cos \theta_i^{(\text{cat})} = \frac{\langle \Delta \mathbf{c}_i^{(S)}, \Delta \mathbf{c}_i^{(T)} \rangle}{\|\Delta \mathbf{c}_i^{(S)}\| \, \|\Delta \mathbf{c}_i^{(T)}\| + \epsilon}, \tag{16}$$

with $\epsilon$ being a small constant for numerical stability. Finally, the overall TE approximation (TE2) is obtained by averaging these cosine similarities over all consecutive pairs in the batch:

$$\text{TE2} = \frac{1}{B-1} \sum_{i=1}^{B-1} \cos \theta_i^{(\text{cat})}. \tag{17}$$

This concatenation-based surrogate captures the joint geometric alignment of image and text representation changes, providing a single metric that reflects how well the student model's combined multimodal directional structure matches that of the teacher.

**Why concatenation is meaningful (and not equivalent to averaging).**  We emphasize that TE2 is *not* intended to be algebraically equivalent to averaging the per-modality cosine similarities (TE1), since cosine similarity is nonlinear. Rather, TE2 provides a *single joint geometric score* by measuring the angle between the *concatenated* update vectors $\Delta \mathbf{c}_i^{(S)} = [\Delta \mathbf{v}_i^{(S)} \,\|\, \Delta \mathbf{u}_i^{(S)}]$ and $\Delta \mathbf{c}_i^{(T)} = [\Delta \mathbf{v}_i^{(T)} \,\|\, \Delta \mathbf{u}_i^{(T)}]$. Geometrically, this is equivalent to computing cosine similarity in the direct-sum space $\mathbb{R}^{d_v} \oplus \mathbb{R}^{d_u}$, i.e., enforcing *joint* alignment of the image and text update directions. By contrast, TE1 treats the two modalities independently and averages their alignment scores.

Writing $a = \Delta \mathbf{v}_i^{(S)}$, $b = \Delta \mathbf{u}_i^{(S)}$, $c = \Delta \mathbf{v}_i^{(T)}$, $d = \Delta \mathbf{u}_i^{(T)}$, we can expand TE2 as

$$\cos \theta_i^{(\text{cat})} = \frac{\langle a, c \rangle + \langle b, d \rangle}{\sqrt{\|a\|^2 + \|b\|^2} \, \sqrt{\|c\|^2 + \|d\|^2} + \epsilon}. \tag{18}$$

Equation (18) shows that TE2 couples the modalities through the *shared normalization*: a modality with large update magnitude (e.g., image) will dominate the joint direction unless the other modality (e.g., text) is also aligned. This coupling is desirable in VLM distillation because it encourages the student to align

the *combined* multimodal update geometry to the teacher, rather than achieving high alignment in only one modality.

In Appendix D, we present evaluation results for our two TE-inspired proxy methods and compare them with an analytic Gaussian TE reference in a simple synthetic setting. In Appendix E, we analyze the computational cost of direct TE estimation versus the proposed TE-inspired proxies. In all experiments, TE and its surrogates are computed on the *global* CLIP embeddings. For images, $V$ denotes the pooled visual embedding (e.g., global representation / CLS token for ViT). For text, $U$ denotes the embedding at the end-of-text (EOT) token position in the final Transformer layer. We do not operate on token-level features; all TE quantities are defined on these pooled representations.

### 4.3 Loss Functions for VLM Distillation

**Motivating the combined objective.** Our total objective combines complementary distillation signals that operate at different granularities. The contrastive loss (CL) preserves CLIP-style global alignment by learning a discriminative cross-modal embedding space within a batch, matching image–text pairs while separating negatives. The KL divergence term further distills the teacher's soft similarity distribution over the batch, transferring relative-neighbor structure beyond hard positives. The feature regression term (MSE/L2) anchors the student to the teacher in representation space to stabilize training and reduce collapse or drift, especially when the student has lower capacity. Interactive contrastive learning (ICL) explicitly enforces cross-modal consistency between student and teacher representations, comparing student image embeddings with teacher text embeddings and student text embeddings with teacher image embeddings, thereby strengthening modality coupling and helping the student inherit the teacher's cross-modal correspondences.

**What the TE-inspired proxy contributes beyond CL/KL/MSE/ICL.** CL, KL, MSE, and ICL primarily encourage static matching of embeddings and similarity structure at a given optimization step. In contrast, the TE-inspired proxy term targets a different signal: it rewards teacher-consistent update geometry by encouraging alignment between the student's representation change and the teacher-implied change through an efficient geometric surrogate. Empirically, we find that this term complements the losses above by improving update dynamics, that is, whether the student moves in teacher-consistent directions, rather than only reducing instantaneous discrepancies. This distinction is especially useful under capacity gaps and training noise, where matching current embeddings alone may be insufficient.

To transfer knowledge from the teacher model to the student in the VLM distillation setting, we use a weighted combination of CL, KL divergence, MSE feature distillation, ICL, a mutual-information (MI) loss (Tian et al., 2020), an MSE-diff (MSE-$\Delta$) loss, and a TE-inspired proxy reward. All loss terms except the TE proxy are defined in Appendix F. We include MI and MSE-diff as established geometric baselines for fair comparison and ablation.

To incorporate the TE-inspired component, we treat its surrogate score as a reward and subtract it from the overall loss. Combining all terms, the total objective is

$$\mathcal{L}_{\text{total}} = \mathcal{L}_{\text{contrastive}} + \alpha\,\mathcal{L}_{\text{KL}} + \beta\,\mathcal{L}_{\text{MSE}} + \delta\,\mathcal{L}_{\text{ICL}} + \eta\,\mathcal{L}_{\text{MI}} + \lambda\,\mathcal{L}_{\text{MSE-}\Delta} - \gamma\,\text{TE}, \qquad (19)$$

where $\alpha$, $\beta$, $\delta$, $\eta$, $\lambda$, and $\gamma$ balance the contributions of the KL, MSE, ICL, MI, MSE-$\Delta$, and TE reward terms, respectively. Here, $\mathcal{L}_{\text{MI}}$ is implemented as an InfoNCE-style contrastive objective that lower-bounds mutual information. Minimizing $\mathcal{L}_{\text{MI}}$ encourages matched teacher–student representations to score higher than mismatched pairs within the batch (Tian et al., 2020).

Overall, this composite objective encourages the student to match the teacher's similarity distributions and feature space while also capturing change-aware signals in representation evolution. In this way, the loss combines distributional alignment, feature matching, and teacher-consistent geometric supervision within a single distillation framework for VLMs.

---

**Algorithm 1:** One distillation step with TE1/TE2 (within-batch finite differences)

---

**Input:** Student $f_S$, frozen teacher $f_T$, batch $\mathcal{B} = \{(\mathbf{x}_i, y_i)\}_{i=1}^B$, Adam optimizer with learning rate $\rho$
**Output:** Updated student parameters $\theta_S$
Sample mini-batch $\mathcal{B}$;
Sample a random permutation $\pi$ over $\{1, \ldots, B\}$;
Form permuted batch $\mathcal{B}_\pi = \{(\mathbf{x}_{\pi(i)}, y_{\pi(i)})\}_{i=1}^B$;
**Teacher forward (once, no grad):** $(\mathbf{v}^{(T)}, \mathbf{u}^{(T)}) \leftarrow f_T(\mathcal{B}_\pi)$;
Detach teacher features;
**Student forward:** $(\mathbf{v}^{(S)}, \mathbf{u}^{(S)}) \leftarrow f_S(\mathcal{B}_\pi)$;
Compute base distillation loss

$$\mathcal{L}_{\text{base}} \leftarrow \mathcal{L}_{\text{CL}} + \alpha \mathcal{L}_{\text{KL}} + \beta \mathcal{L}_{\text{L2}} + \delta \mathcal{L}_{\text{ICL}} + \eta \mathcal{L}_{\text{MI}} + \lambda \mathcal{L}_{\text{MSE-}\Delta}$$

`// Within-batch finite differences using adjacent pairs after permutation`
**for** $i = 1$ **to** $B - 1$ **do**
    $\Delta \mathbf{v}_i^{(S)} \leftarrow \mathbf{v}_{i+1}^{(S)} - \mathbf{v}_i^{(S)}$;
    $\Delta \mathbf{u}_i^{(S)} \leftarrow \mathbf{u}_{i+1}^{(S)} - \mathbf{u}_i^{(S)}$;
    $\Delta \mathbf{v}_i^{(T)} \leftarrow \mathbf{v}_{i+1}^{(T)} - \mathbf{v}_i^{(T)}$;
    $\Delta \mathbf{u}_i^{(T)} \leftarrow \mathbf{u}_{i+1}^{(T)} - \mathbf{u}_i^{(T)}$;
`// TE1: per-modality alignment`
$\text{TE}_{\text{img}} \leftarrow \frac{1}{B-1} \sum_{i=1}^{B-1} \cos(\Delta \mathbf{v}_i^{(S)}, \Delta \mathbf{v}_i^{(T)})$;
$\text{TE}_{\text{txt}} \leftarrow \frac{1}{B-1} \sum_{i=1}^{B-1} \cos(\Delta \mathbf{u}_i^{(S)}, \Delta \mathbf{u}_i^{(T)})$;
$\text{TE1} \leftarrow \frac{1}{2}(\text{TE}_{\text{img}} + \text{TE}_{\text{txt}})$;
`// TE2: joint multimodal alignment`
**for** $i = 1$ **to** $B - 1$ **do**
    $\Delta \mathbf{c}_i^{(S)} \leftarrow [\Delta \mathbf{v}_i^{(S)} \| \Delta \mathbf{u}_i^{(S)}]$;
    $\Delta \mathbf{c}_i^{(T)} \leftarrow [\Delta \mathbf{v}_i^{(T)} \| \Delta \mathbf{u}_i^{(T)}]$;
$\text{TE2} \leftarrow \frac{1}{B-1} \sum_{i=1}^{B-1} \cos(\Delta \mathbf{c}_i^{(S)}, \Delta \mathbf{c}_i^{(T)})$;
Set total loss: $\mathcal{L} \leftarrow \mathcal{L}_{\text{base}} - \gamma(\text{TE1} + \text{TE2})$;
Backpropagate $\mathcal{L}$ and update $\theta_S$ using Adam with learning rate $\rho$;

---

### 4.4 Description of a Full Training Step with TE Surrogates

**One training step with TE surrogates (within-batch finite differences).** Algorithm 1 summarizes one full distillation iteration and makes explicit how the TE-inspired surrogate terms (TE1/TE2) are computed in practice. At each iteration, we sample a mini-batch $\mathcal{B} = \{(\mathbf{x}_i, y_i)\}_{i=1}^B$ and perform one student parameter update using Adam. The teacher $f_T$ is frozen, so its embeddings provide a stable reference geometry for the current batch, while the student $f_S$ is updated by backpropagation through the full objective.

Crucially, TE1/TE2 do not require any additional teacher evaluation beyond standard distillation. We run the teacher forward once, without gradients, on the current batch and detach the resulting embeddings $(\mathbf{v}^{(T)}, \mathbf{u}^{(T)})$. We also run the student forward once on the same permuted batch to obtain $(\mathbf{v}^{(S)}, \mathbf{u}^{(S)})$. To construct finite-difference proxies for local directional structure, we sample a random permutation $\pi$ of the batch indices and reorder the batch as $\mathcal{B}_\pi$. This permutation is used only to define a diverse set of within-batch finite-difference pairs; it does not impose any temporal or semantic ordering on the samples.

We then form within-batch differences using adjacent pairs in the permuted order: $\Delta \mathbf{v}_i = \mathbf{v}_{i+1} - \mathbf{v}_i$ and $\Delta \mathbf{u}_i = \mathbf{u}_{i+1} - \mathbf{u}_i$ for both teacher and student. These differences should be interpreted as sampled local directional changes in embedding space. The index $i$ enumerates permuted samples and has no temporal meaning; accordingly, TE1/TE2 are not temporal transfer-entropy estimators computed from shuffled data.

Instead, TE1 measures per-modality cosine alignment between teacher and student directions, while TE2 measures joint multimodal alignment by concatenating image and text directions before computing cosine similarity. Together, these surrogate rewards encourage the student to match the teacher's local representation geometry on each batch in a relational, direction-field sense.

Algorithm 1 presents the most general form in which both TE1 and TE2 are included in the loss. TE1-only and TE2-only ablations are obtained by setting the other surrogate term to zero.

**Compute footprint.** Per iteration, we run the teacher forward once without gradients and the student forward once with gradients, as in standard distillation. TE1/TE2 add only lightweight vector-difference and cosine-similarity computations on already-computed embeddings, and therefore introduce negligible overhead: no extra teacher forward passes, no Jacobian construction, and no cache-and-replay procedure.

We confirm that all reported results in this paper were obtained using the implementation described in Algorithm 1. The corresponding code is provided in the supplementary material to support reproducibility and to eliminate ambiguity about the training procedure.

## 5 Experiments

We evaluate the proposed method under four teacher–student configurations: (1) teacher: ResNet-50, student: ResNet-34; (2) teacher: ViT-B/16, student: ResNet-34; (3) teacher: ResNet-50×16, student: ResNet-34; and (4) teacher: ResNet-50, student: ResNet-18. Brief background on CLIP RN50, ViT-B/16, and RN50×16 is provided in Appendix G.

Our main retrieval experiments are conducted on MSCOCO 2014 (Lin et al., 2014), Flickr8k (Hodosh et al., 2013; Marco et al., 2023), and Flickr30k (Young et al., 2014). We further evaluate transfer to classification through experiments on Food-101 (Bossard et al., 2014), and we assess out-of-dataset zero-shot classification on ImageNet-1k (Deng et al., 2009; Russakovsky et al., 2015).

### 5.1 Teacher: RN50, Student: RN34, Dataset: MSCOCO

#### 5.1.1 Experimental Settings

The student VLM consists of an RN34 image encoder and a lightweight Transformer text encoder. The RN34 backbone contains approximately 21.8 million parameters, with its final fully connected layer modified to output 1024-dimensional features while keeping the parameter count relatively stable (He et al., 2016). The text encoder uses a vocabulary of size 49,408 and hidden dimension 1024, contributing approximately 25.3 million parameters (Mehta et al., 2020). In addition, the student Transformer has 2 encoder layers with 8-head attention (Vaswani et al., 2017). Combining the image and text encoders, the total parameter count of the student is approximately 55–60 million, substantially smaller than the teacher model while retaining sufficient capacity for multimodal distillation.

Our experiments are conducted on the MSCOCO 2014 dataset (Lin et al., 2014), which contains 82,783 training images and 40,504 validation images, each paired with multiple textual descriptions. MSCOCO is widely used in vision–language research because of its large scale and diverse image–caption pairs.

The student is trained with combinations of contrastive loss (CL), KL divergence, MSE loss, ICL loss, MI-based objectives, and the proposed TE-inspired proxy regularizers. The total training objective is given in Eq. (19). We study multiple combinations of these components to evaluate how the TE-inspired proxy interacts with standard distillation losses and whether it provides consistent improvements beyond existing objectives. Importantly, the proposed method is not a direct estimator of optimization-step transfer entropy; rather, it is a tractable proxy objective motivated by that perspective and implemented through within-batch finite-difference surrogates.

The hyperparameters $\alpha$, $\beta$, $\delta$, $\eta$, $\lambda$, and $\gamma$ in Eq. (19) control the relative contributions of the CL, KL, MSE, ICL, MI, and TE-inspired terms. Because these components can have different numerical scales, we choose weights so that no single term dominates optimization solely due to magnitude differences. Training

dynamics for representative loss combinations are shown in Fig. 1, and additional recipe-level diagnostics on temperature and batch size are provided later in Section 5.1.3.

Unless otherwise specified, we optimize the student using Adam with learning rate $\rho = 10^{-4}$, and use $\alpha = 1.0$, $\beta = 50$, $\delta = 1.0$, temperature $\tau = 0.07$, and batch size $|B| = 64$. The training data is shuffled at each epoch. For computational efficiency, each model is trained for 10 epochs per configuration.

All experiments in this subsection were run on Google Colab Pro using a T4 GPU with High-RAM. Under this setup, each training run and evaluation took approximately 14 hours.

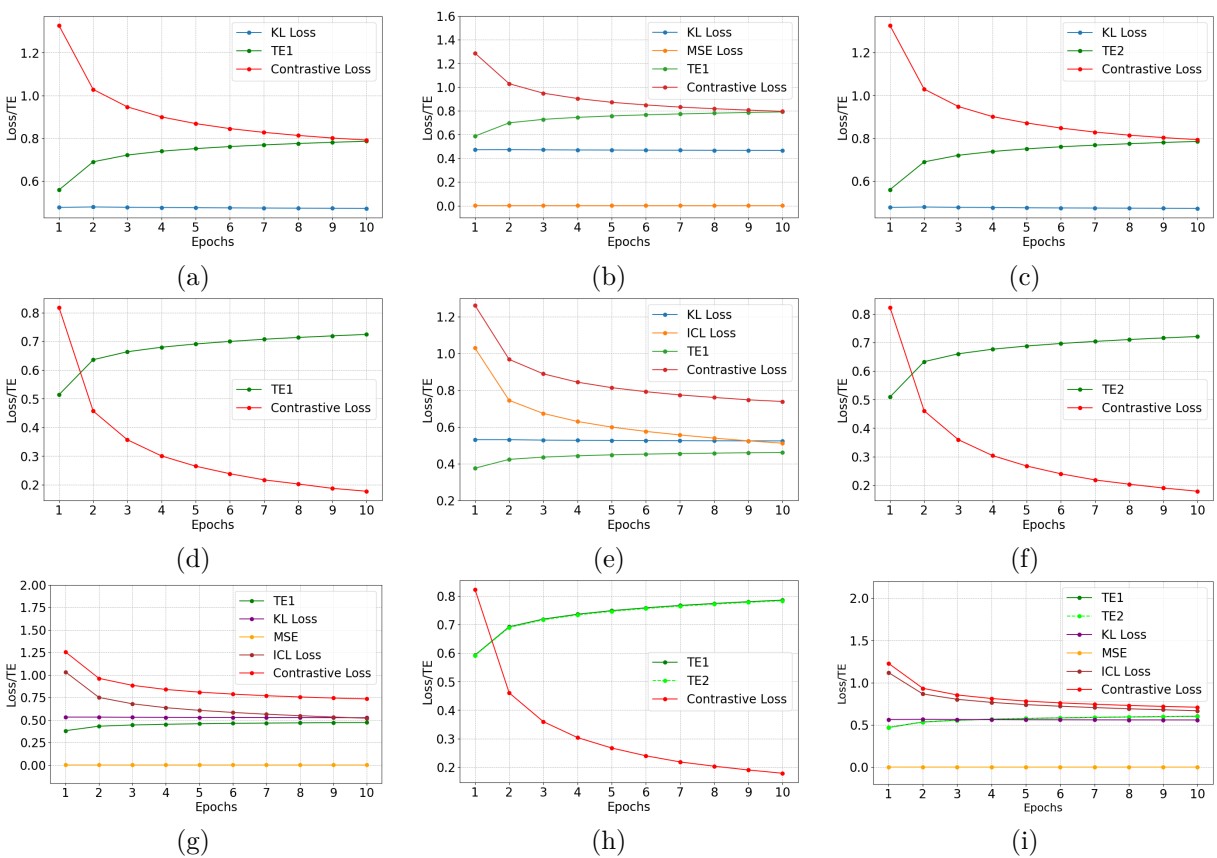

Figure 1: The training losses and TE for different loss functions in the distillation training of RN34-based VLM student with teacher CLIP RN50 using MSCOCO dataset. (a) Contrastive + KL - TE1, (b) Contrastive + KL + MSE - TE2, (c) Contrastive +KL - TE2, (d) Contrastive - TE1, (e) Contrastive + KL + ICL - TE1, (f) Contrastive - TE2, (g) Contrastive + KL +MSE + ICL - TE1, (h) Contrastive - TE1 - TE2, (i) Contrastive + KL + ICL + MSE -TE1 - TE2.

### 5.1.2 Experimental Results and Hyperparameter Selection

Figure 1 shows that the total training loss generally decreases over epochs, while the TE surrogate rewards tend to increase. This indicates that the optimization process is successfully reducing the overall objective while increasing teacher–student geometric alignment under the proxy terms. For experiments that include both TE1 and TE2, such as Fig. 1h and Fig. 1i, the two surrogate rewards increase monotonically with similar but not identical trajectories. At the same time, KL and MSE do not always decrease monotonically, suggesting that the different objective components interact in nontrivial ways during training. We therefore interpret these curves as training-dynamics diagnostics rather than as evidence that each individual component should decrease monotonically in isolation.

The interactions among CL, KL, ICL, MSE, and the TE-inspired proxy terms are also reflected in the final surrogate values. For example, in `CL - TE1` (Fig. 1d), TE1 reaches an average value of 0.7242 at epoch 10; in `CL + KL - TE1` (Fig. 1a), it reaches 0.7865; while in `CL + KL + ICL - TE1` (Fig. 1e), it reaches 0.4611. These comparisons suggest that adding KL is associated with higher TE1 values in this setting, whereas adding ICL on top of CL+KL is associated with lower TE1 values. We emphasize that these observations are empirical properties of the training objective in this experimental regime, rather than general causal claims about the losses themselves.

We evaluate the trained student models using Recall@K for both image-to-text (I2T) and text-to-image (T2I) retrieval. Recall@K measures the percentage of queries for which the correct match appears among the top-$K$ retrieved results (Manning et al., 2008). Higher Recall@1 indicates stronger fine-grained alignment between images and texts, while Recall@5 and Recall@10 provide a broader view of retrieval quality. Table 1 summarizes the performance of the RN34 student under different loss combinations.

Table 1: Comparison of zero-shot retrieval performance (Recall@k) of RN34-based VLM student with teacher CLIP RN50 for different loss function combinations in VLM distillation using **MSCOCO**. All Loss Function: CL + KL + MSE + ICL - TE1 - TE2. The weighting factors for loss functions: $\alpha = 1.0$, $\beta = 50$, $\delta = 1.0$, $\gamma = 1.0$.

| Model and Loss Function | I2T Retrieval (R) | | | T2I Retrieval (R) | | |
|---|---|---|---|---|---|---|
| | R@1 | R@5 | R@10 | R@1 | R@5 | R@10 |
| **Teacher Model (RN50)** | 25.29% | 45.68% | 55.37% | 12.02% | 26.44% | 34.51% |
| **Student Models (RN34)** | | | | | | |
| CL Only (Oord et al., 2018) | 4.94% | 14.60% | 22.51% | 3.96% | 12.67% | 19.45% |
| CL + MSE (Yang et al., 2024) | 5.13% | 15.41% | 23.17% | 4.00% | 12.79% | 19.53% |
| CL + KL (Li et al., 2024b) | 5.42% | 16.20% | 24.55% | 5.06% | 15.35% | 22.92% |
| CL + ICL (Yang et al., 2024) | 5.75% | 16.86% | 24.83% | 5.07% | 15.14% | 22.44% |
| CL - TE1 | 6.91% | 19.48% | 28.12% | 5.68% | 16.33% | 23.93% |
| CL - TE2 | 7.04% | 19.22% | 27.97% | 5.46% | 15.90% | 23.49% |
| CL - TE1 - TE2 | **8.24**% | **22.43**% | **31.73**% | 6.53% | 18.13% | 26.02% |
| CL + KL - TE1 | 7.81% | 21.31% | 30.65% | 6.42% | 18.18% | 26.33% |
| CL + KL - TE2 | 7.77% | 21.10% | 30.22% | 6.21% | 17.95% | 26.03% |
| CL + KL + MSE - TE1 | 7.62% | 21.05% | 30.34% | 6.53% | 18.48% | 26.66% |
| CL + KL + ICL - TE1 | 7.59% | 20.87% | 29.95% | 6.78% | 19.11% | 27.33% |
| CL + KL + MSE + ICL - TE1 | 7.51% | 20.62% | 29.81% | 6.76% | 19.02% | 27.32% |
| All Loss Function | 8.11% | 22.05% | 31.57% | **7.18**% | **19.75**% | **28.14**% |

Table 1 shows that adding TE-inspired proxy terms substantially improves retrieval relative to CL-only and standard auxiliary-loss baselines. In particular, introducing TE1 or TE2 on top of the base contrastive objective yields the largest single-step gains among the tested additions. The strongest I2T result is obtained by `CL - TE1 - TE2`, while the full loss combination `CL + KL + MSE + ICL - TE1 - TE2` achieves the best T2I performance. These results suggest that the TE-inspired proxy terms provide strong regularization on their own and can also complement more conventional distillation losses when combined appropriately. Overall, the best-performing configurations consistently include TE components, indicating that the proxy objective helps the student align more effectively with the teacher's multimodal structure.

Table 2 reports the sensitivity of retrieval performance to different hyperparameter settings. Overall, the method is reasonably stable under moderate changes in loss weights, while the TE reward weight $\gamma$ has the clearest effect on retrieval quality. Starting from the baseline without TE ($\gamma = 0$), introducing a nonzero TE weight improves retrieval in both directions, and Image-to-Text performance generally improves as $\gamma$ increases from 1 to 7.5. In particular, $\gamma = 7.5$ achieves the best I2T result, reaching R@1/R@5/R@10 of 10.27/26.36/36.30. Increasing $\gamma$ further to 10 yields slightly weaker performance, suggesting diminishing returns and mild over-regularization when the TE reward is too strong.

By comparison, varying $\alpha$ (KL), $\beta$ (MSE), and $\delta$ (ICL) within the tested ranges leads to relatively smaller fluctuations, indicating that the objective is not overly sensitive to these terms once they are set to reasonable values. Adding the MI loss also improves retrieval, especially for T2I: setting $\eta = 2.5$ with $\gamma = 0$ yields the strongest T2I performance in Table 2, reaching 7.66/20.69/29.25. In contrast, using the MSE-$\Delta$ loss alone ($\lambda \in \{50, 100\}$ with $\gamma = 0$) provides only marginal gains over the baseline, and combining multiple auxiliary terms does not always produce additive improvements. Overall, these results suggest that moderate TE weighting is the most effective way to strengthen I2T retrieval in this setting, while MI-based supervision remains competitive for T2I.

Table 2: Comparison of zero-shot retrieval performance (Recall@k) in percentage of RN34-based VLM student with teacher CLIP RN50 on MSCOCO.

| $\alpha$ | $\beta$ | $\delta$ | $\eta$ | $\lambda$ | $\gamma$ | I2T R@1 | I2T R@5 | I2T R@10 | T2I R@1 | T2I R@5 | T2I R@10 |
|---|---|---|---|---|---|---|---|---|---|---|---|
| 1 | 50 | 1 | 0 | 0 | 0 | 6.08% | 18.14% | 26.93% | 5.92% | 17.06% | 24.89% |
| 1 | 50 | 1 | 0 | 0 | 1 | 8.11% | 22.05% | 31.57% | 7.18% | 19.75% | 28.14% |
| 1 | 50 | 1 | 0 | 0 | 2.5 | 9.48% | 24.68% | 34.54% | 7.57% | 20.55% | 29.03% |
| 1 | 50 | 1 | 0 | 0 | 5 | 10.02% | 25.80% | 35.74% | 7.32% | 20.04% | 28.50% |
| 1 | 50 | 1 | 0 | 0 | 7.5 | **10.27%** | **26.36%** | **36.30%** | 6.98% | 18.95% | 26.95% |
| 1 | 50 | 1 | 0 | 0 | 10 | 9.77% | 25.25% | 35.04% | 6.94% | 18.78% | 26.67% |
| 5 | 50 | 1 | 0 | 0 | 5 | 10.19% | 26.15% | 36.09% | 6.81% | 18.84% | 26.88% |
| 1 | 100 | 1 | 0 | 0 | 5 | 10.25% | 25.76% | 35.69% | 7.30% | 20.55% | 29.03% |
| 1 | 50 | 5 | 0 | 0 | 5 | 8.20% | 22.14% | 31.63% | 7.14% | 19.61% | 28.05% |
| 1 | 50 | 1 | 0 | 50 | 0 | 6.67% | 18.48% | 27.30% | 6.01% | 17.27% | 25.24% |
| 1 | 50 | 1 | 0 | 100 | 0 | 6.52% | 18.90% | 27.57% | 6.29% | 17.71% | 25.67% |
| 1 | 50 | 1 | 1 | 1 | 0 | 8.35% | 22.62% | 31.70% | 7.43% | 20.23% | 28.87% |
| 1 | 50 | 1 | 2.5 | 0 | 0 | 8.56% | 23.01% | 32.41% | **7.66%** | **20.69%** | **29.25%** |
| 1 | 50 | 1 | 5 | 0 | 0 | 8.65% | 23.29% | 33.05% | 7.65% | 20.63% | 29.05% |
| 1 | 50 | 1 | 7.5 | 0 | 0 | 8.25% | 22.41% | 32.04% | 7.44% | 20.27% | 28.61% |
| 1 | 50 | 1 | 2.5 | 50 | 2.5 | 9.12% | 24.25% | 33.88% | 7.31% | 19.98% | 28.40% |
| 1 | 50 | 1 | 2.5 | 0 | 5 | 9.94% | 25.73% | 35.54% | 6.87% | 18.92% | 27.09% |
| 1 | 50 | 1 | 2.5 | 50 | 5 | 9.26% | 24.26% | 34.44% | 7.05% | 19.47% | 27.78% |
| 1 | 50 | 1 | 1 | 0 | 7.5 | 9.67% | 24.85% | 34.78% | 6.91% | 18.95% | 27.07% |
| 1 | 50 | 1 | 1 | 50 | 7.5 | 10.23% | 25.93% | 35.81% | 6.87% | 18.91% | 27.04% |
| 1 | 50 | 1 | 5 | 50 | 7.5 | 9.03% | 24.15% | 33.87% | 6.85% | 18.80% | 26.77% |

We also observe that the TE-inspired gains are stronger and more consistent for image-to-text retrieval than for text-to-image retrieval. This asymmetry is already present in the teacher and is partly inherited by the student through distillation. In Table 1, the RN50 teacher obtains 25.29% I2T R@1 but only 12.02% T2I R@1, indicating that the teacher itself provides a stronger signal for image-to-text retrieval. Since the student is trained to preserve the teacher's cross-modal geometry, it naturally reflects this direction-dependent behavior. In addition, I2T and T2I are not identical ranking problems: I2T ranks captions for a given image, while T2I ranks images for a given caption, and the two directions can have different sensitivity to image-side, text-side, and joint embedding geometry. In our current formulation, TE1/TE2 act as shared teacher-student geometric regularizers rather than retrieval-direction-specific objectives. Therefore, the same regularization strength may improve I2T more consistently while yielding more mixed effects on T2I.

**Hyperparameter selection and reproducibility.** The weights $\alpha, \beta, \delta, \eta, \lambda$, and $\gamma$ control the relative influence of the KL, MSE (feature regression), ICL, MI, MSE-$\Delta$, and TE surrogate terms with respect to the base contrastive objective. By "assigning larger weights to losses with smaller magnitudes," we use the following heuristic: during a short warm-up run (approximately the first epoch) with a fixed reference setting, we log the *batch-averaged* value of each unweighted component $\{\mathcal{L}_{CL}, \mathcal{L}_{KL}, \mathcal{L}_{MSE}, \mathcal{L}_{ICL}, \mathcal{L}_{MI}, \mathcal{L}_{MSE\Delta}, TE_1, TE_2\}$, and choose weights so that the corresponding *weighted* components have roughly comparable scale:

$$\alpha \, \mathbb{E}[\mathcal{L}_{KL}] \approx \beta \, \mathbb{E}[\mathcal{L}_{MSE}] \approx \delta \, \mathbb{E}[\mathcal{L}_{ICL}] \approx \eta \, \mathbb{E}[\mathcal{L}_{MI}] \approx \lambda \, \mathbb{E}[\mathcal{L}_{MSE\Delta}] \approx \gamma \, \mathbb{E}[TE] \approx \mathbb{E}[\mathcal{L}_{CL}], \quad (20)$$

where TE denotes the bounded cosine-based surrogate ($\text{TE}_1/\text{TE}_2$). Intuitively, when a term is naturally small, such as the MSE between normalized embeddings, a larger coefficient is needed for it to meaningfully affect optimization; conversely, terms already on the same scale as $\mathcal{L}_{\text{CL}}$ require smaller coefficients.

After setting a reasonable scale using Eq. (20), we perform a small grid search around these values and select the final configuration by validation Recall@K on MSCOCO. Table 2 summarizes this sweep. In the main RN50→RN34 MSCOCO setting, we fix $\alpha = 1$ and $\delta = 1$, use $\beta \in \{50, 100\}$ depending on the experiment, and tune $\gamma$ over $\{1, 2.5, 5, 7.5, 10\}$, with $\gamma = 7.5$ giving the strongest I2T performance. For MI baselines, we tune $\eta$ analogously, typically over $\{1, 2.5, 5, 7.5\}$, and we set $\lambda = 0$ unless explicitly evaluating the MSE-$\Delta$ ablation.

### 5.1.3 Training Dynamics and Recipe-Level Diagnostics

We further examine both the training dynamics of the auxiliary objectives and the sensitivity of the method to recipe-level parameters. Our results show that the behavior of KL and MSE is strongly recipe-dependent, especially with respect to the contrastive temperature. In the original $\tau = 0.07$ setting (e.g., Fig. 1b), the KL term decreases only slightly, from 0.4721 to 0.4662 (about 1.25%), whereas the MSE term decreases from $5.93 \times 10^{-4}$ to $3.19 \times 10^{-4}$ (about 46.2%). This previously suggested that direct feature-level alignment was easier to optimize than similarity-distribution matching (e.g., KL). However, the lower-temperature results in Fig. 2 show that this interpretation is not universal: at $\tau = 0.03$, both TE- and MI-based objectives exhibit substantial KL reduction. In particular, under TE-based training ($\gamma = 7.5$), KL decreases from 0.5800 to 0.3407 (a 41.3% reduction), while MSE decreases from $5.48 \times 10^{-4}$ to $1.78 \times 10^{-4}$ (a 67.5% reduction). Under MI-based training ($\eta = 5$), KL decreases from 0.6148 to 0.3403 (a 44.7% reduction), although the MSE term decreases more modestly, from $1.132 \times 10^{-3}$ to $9.37 \times 10^{-4}$ (a 17.2% reduction). Thus, the weaker KL trend at $\tau = 0.07$ should not be interpreted as an inherent limitation of the objectives themselves, but rather as a consequence of recipe choice.

These results suggest that the contrastive temperature plays a major role in determining how strongly teacher-structure matching is optimized. A lower temperature sharpens the induced similarity distribution, making mismatches among positive pairs and hard negatives more salient and thereby providing a stronger signal for reducing KL. At the same time, the two objectives still differ in feature-level alignment: TE yields much stronger MSE reduction than MI at $\tau = 0.03$, indicating that TE more effectively preserves direct teacher–student feature correspondence even when both objectives achieve comparable final KL values. We therefore interpret the training dynamics as follows: KL reduction is highly temperature-sensitive, whereas the stronger MSE reduction under TE reflects a more persistent advantage in feature-level teacher alignment. This distinction helps explain why the proposed TE regularizer can produce more teacher-faithful geometry and stronger image-to-text retrieval, even when KL alone does not fully separate the two objectives.

Table 3: Temperature $\tau$ ablation on MSCOCO under $\alpha = 1$, $\beta = 50$, $\delta = 1$. TE denotes ($\gamma = 7.5, \eta = 0$), and MI denotes ($\eta = 5, \gamma = 0$). TE consistently outperforms MI on image-to-text retrieval across all tested temperatures, with the strongest TE image-to-text performance observed at $\tau = 0.03$. Best result within each temperature and metric is boldfaced.

| $\tau$ | Method | Image-to-Text | | | Text-to-Image | | |
|---|---|---|---|---|---|---|---|
| | | R@1 | R@5 | R@10 | R@1 | R@5 | R@10 |
| 0.03 | TE | **10.85** | **26.47** | **36.29** | **6.84** | **18.76** | **26.94** |
| 0.03 | MI | 9.14 | 23.67 | 33.14 | 6.49 | 18.23 | 26.44 |
| 0.07 | TE | **10.27** | **26.36** | **36.30** | 6.98 | 18.95 | 26.95 |
| 0.07 | MI | 8.65 | 23.29 | 33.05 | **7.65** | **20.63** | **29.05** |
| 0.15 | TE | **9.68** | **24.92** | **34.64** | 6.86 | 18.14 | 25.80 |
| 0.15 | MI | 8.73 | 23.10 | 32.42 | **6.96** | **18.54** | **26.23** |

We next study the sensitivity of the proposed objective to the contrastive temperature $\tau \in \{0.03, 0.07, 0.15\}$ and the batch size $|B| \in \{64, 128\}$ on MSCOCO-2014, while keeping the remaining hyperparameters fixed at

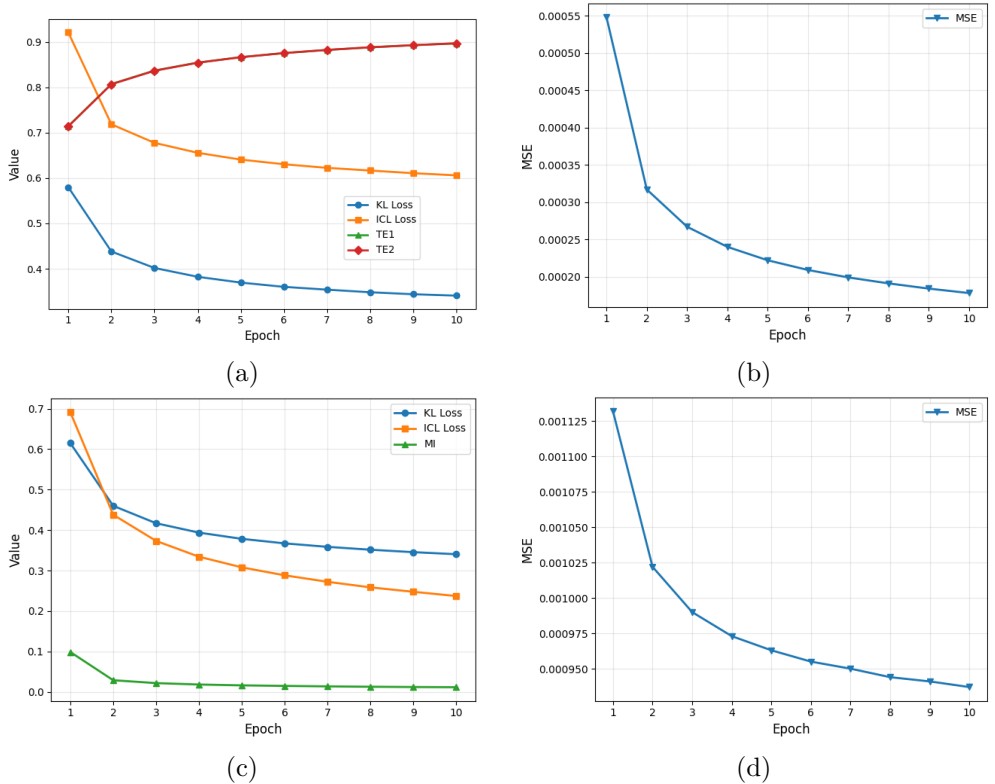

Figure 2: Training dynamics at $\tau = 0.03$ for TE-based ($\alpha = 1, \beta = 50, \delta = 1, \gamma = 7.5$) and MI-based ($\alpha = 1, \beta = 50, \delta = 1, \eta = 5$) distillation. (a) KL, ICL, TE1, and TE2 in TE-based distillation. (b) MSE in TE-based distillation. (c) KL, ICL, and MI in MI-based distillation. (d) MSE in MI-based distillation. At the lower temperature, both objectives exhibit substantial KL reduction, while TE-based yields markedly stronger MSE reduction than MI-based.

Table 4: Batch-size ablation on MSCOCO at fixed $\tau = 0.07$ under $\alpha = 1$, $\beta = 50$, $\delta = 1$. TE denotes ($\gamma = 7.5, \eta = 0$), and MI denotes ($\eta = 5, \gamma = 0$). Best result within each batch size and metric is boldfaced.

| $|B|$ | Method | Image-to-Text | | | Text-to-Image | | |
|---|---|---|---|---|---|---|---|
| | | R@1 | R@5 | R@10 | R@1 | R@5 | R@10 |
| 64 | TE | **10.27** | **26.36** | **36.30** | 6.98 | 18.95 | 26.95 |
| 64 | MI | 8.65 | 23.29 | 33.05 | **7.65** | **20.63** | **29.05** |
| 128 | TE | **10.69** | **26.94** | **37.18** | 7.07 | 19.25 | 27.52 |
| 128 | MI | 9.64 | 25.60 | 35.50 | **7.62** | **20.46** | **29.01** |

$\alpha = 1$, $\beta = 50$, and $\delta = 1$. We compare the TE-regularized objective ($\gamma = 7.5, \eta = 0$) against the MI-based baseline ($\eta = 5, \gamma = 0$). As shown in Table 3, the main pattern is stable across temperatures: TE consistently improves image-to-text retrieval over the MI baseline at all tested values of $\tau$. In particular, TE improves I2T Recall@1 from 9.14% to 10.85% at $\tau = 0.03$, from 8.65% to 10.27% at $\tau = 0.07$, and from 8.73% to 9.68% at $\tau = 0.15$, with matching gains at Recall@5 and Recall@10. Among the tested values, $\tau = 0.03$ yields the strongest image-to-text performance for TE in this setting. This lower-temperature regime is also consistent with the training-dynamics results in Fig. 2, where both KL and MSE decrease much more substantially than under the original $\tau = 0.07$ recipe, indicating that the earlier weak KL trend was largely a recipe effect rather than an inherent limitation of the objective. On text-to-image retrieval, the pattern is more mixed:

TE is slightly better at $\tau = 0.03$, while the MI baseline is stronger at $\tau = 0.07$ and marginally stronger at $\tau = 0.15$.

We then vary the batch size at fixed $\tau = 0.07$, which changes the effective negative-pool strength in the contrastive objective while keeping the original reference recipe unchanged. Table 4 shows that the TE advantage on image-to-text retrieval persists for both $|B| = 64$ and $|B| = 128$. At $|B| = 64$, TE improves I2T Recall@1 from 8.65% to 10.27%; at $|B| = 128$, it improves I2T Recall@1 from 9.64% to 10.69%, with consistent gains at Recall@5 and Recall@10 as well. Increasing the batch size from 64 to 128 also slightly improves the TE model itself, from 10.27/26.36/36.30 to 10.69/26.94/37.18 on I2T Recall@1/5/10, suggesting that the proposed regularizer remains compatible with a stronger contrastive setup rather than relying on a weak negative pool.

The KL and MSE trajectories remain qualitatively similar across batch sizes. For example, under the TE-inspired setting ($\gamma = 7.5$), KL decreases from 0.6508 to 0.6422 and MSE decreases from 0.000632 to 0.000338 at $|B| = 64$, while at $|B| = 128$ KL decreases from 0.7556 to 0.7484 and MSE decreases from 0.000699 to 0.000404. Under the MI-based setting ($\eta = 5$), KL decreases from 0.6597 to 0.6047 and MSE decreases from 0.001004 to 0.000904 at $|B| = 64$, while at $|B| = 128$ KL decreases from 0.4657 to 0.4351 and MSE decreases from 0.000901 to 0.000690. Thus, changing the batch size does not substantially alter the overall KL/MSE behavior, suggesting that the main TE retrieval gains are not explained by a special batch-size-dependent training dynamic.

Overall, these experiments show that the TE-based gains on image-to-text retrieval are reproducible across multiple temperatures and batch sizes, although the improvements remain stronger for image-to-text retrieval than for text-to-image retrieval.

### 5.1.4 Cross-Dataset Retrieval Experiment

To further evaluate whether the proposed TE-inspired proxy regularization preserves transferable multimodal structure beyond the training distribution, we conducted a cross-dataset retrieval experiment. Specifically, we distilled an RN34 student from an RN50 teacher using MSCOCO for training and then evaluated zero-shot retrieval on Flickr8k without any additional fine-tuning. We compare TE-based settings ($\eta = 0, \gamma > 0$) against MI-based settings ($\eta > 0, \gamma = 0$), while fixing the other hyperparameters to $\alpha = 1$, $\beta = 50$, $\delta = 1$, and $\lambda = 0$.

Table 5: Cross-dataset zero-shot retrieval performance (%) for RN34 distilled from RN50. Training dataset: MSCOCO. Evaluation dataset: Flickr8k. TE-based settings use $\eta = 0, \gamma > 0$; MI-based settings use $\eta > 0, \gamma = 0$. Best TE and best MI results within each retrieval direction are boldfaced separately.

| $\tau$ | $\alpha$ | $\beta$ | $\delta$ | $\eta$ | $\lambda$ | $\gamma$ | Image-to-Text | | | | Text-to-Image | | | |
|---|---|---|---|---|---|---|---|---|---|---|---|---|---|---|
| | | | | | | | R@1 | R@5 | R@10 | MRR | R@1 | R@5 | R@10 | MRR |
| 0.07 | 1 | 50 | 1 | 0 | 0 | 2.5 | 15.36 | 34.33 | 44.77 | 25.03 | 11.40 | 27.59 | 36.84 | 19.90 |
| 0.07 | 1 | 50 | 1 | 0 | 0 | 5 | 16.35 | 35.85 | 46.41 | 26.23 | **12.17** | **28.44** | **37.98** | **20.74** |
| 0.07 | 1 | 50 | 1 | 0 | 0 | 7.5 | 16.66 | 36.02 | 46.22 | 26.43 | 11.79 | 27.41 | 36.95 | 20.15 |
| 0.03 | 1 | 50 | 1 | 0 | 0 | 5 | 17.19 | 36.15 | 46.67 | 26.85 | 11.22 | 27.14 | 36.47 | 19.59 |
| 0.03 | 1 | 50 | 1 | 0 | 0 | 7.5 | **17.74** | **38.26** | **48.76** | **27.92** | 11.80 | 27.50 | 36.88 | 20.10 |
| 0.07 | 1 | 50 | 1 | 2.5 | 0 | 0 | 13.85 | 31.95 | 42.62 | 23.26 | 11.08 | 26.45 | 35.62 | 19.23 |
| 0.07 | 1 | 50 | 1 | 5 | 0 | 0 | **15.51** | 33.58 | 44.14 | **24.82** | **11.89** | 27.70 | 36.93 | 20.23 |
| 0.07 | 1 | 50 | 1 | 7.5 | 0 | 0 | 14.58 | **33.68** | **45.20** | 24.48 | 11.79 | **27.79** | **37.31** | **20.24** |
| 0.03 | 1 | 50 | 1 | 5 | 0 | 0 | 13.51 | 30.08 | 41.19 | 22.41 | 9.70 | 23.96 | 32.70 | 17.38 |
| 0.03 | 1 | 50 | 1 | 7.5 | 0 | 0 | 13.22 | 30.08 | 40.09 | 22.11 | 9.68 | 23.81 | 32.62 | 17.33 |

Table 5 shows that the TE-inspired proxy objective transfers more effectively across datasets than the MI-based baseline in this setting. Beyond Recall@K, we also evaluate Mean Reciprocal Rank (MRR) (Schütze et al., 2008) for both I2T and T2I retrieval in Table 5. MRR is a ranking metric that evaluates how early the first relevant item appears in a retrieved list. For each query, the reciprocal rank is the inverse of the

rank of the first correct result. Averaging across $N$ queries gives

$$\text{MRR} = \frac{1}{N} \sum_{i=1}^{N} \frac{1}{\text{rank}_i}, \tag{21}$$

where $\text{rank}_i$ denotes the position of the first relevant item for query $i$. Higher MRR indicates that correct results tend to appear earlier in the ranking. For I2T, each image is associated with multiple ground-truth captions: we compute similarities to all caption embeddings, determine the 1-indexed rank of each correct caption, and use the best (minimum) rank to accumulate the reciprocal rank 1/rank. For T2I, each caption has a single correct image; we rank all images by similarity to the caption and again use 1/rank to compute MRR. All metrics are computed over the full validation split so that MRR follows the same protocol as Recall@K while providing a more fine-grained assessment of ranking quality across the entire list.

Across temperatures, TE-based configurations consistently outperform MI-based ones for image-to-text retrieval. At $\tau = 0.07$, the best MI setting is $\eta = 5$, which reaches 15.51/33.58/44.14 with MRR 24.82, while TE improves this to 16.35/35.85/46.41 with MRR 26.23 at $\gamma = 5$, and further to 16.66/36.02/46.22 with MRR 26.43 at $\gamma = 7.5$. At the lower temperature $\tau = 0.03$, the TE gains become even larger: $\gamma = 5$ achieves 17.19/36.15/46.67 with MRR 26.85, and $\gamma = 7.5$ gives the strongest overall I2T result, reaching 17.74/38.26/48.76 with MRR 27.92. In contrast, the MI baseline degrades substantially at $\tau = 0.03$, where $\eta = 5$ drops to 13.51/30.08/41.09 with MRR 22.41. These results indicate that the TE-inspired objective yields not only stronger cross-dataset transfer, but also greater robustness to temperature changes for I2T retrieval.

For text-to-image retrieval, the picture is more mixed, but TE remains competitive and attains the best overall result. At $\tau = 0.07$, the best TE setting is $\gamma = 5$, which achieves 12.17/28.44/37.98 with MRR 20.74, outperforming the best MI setting at the same temperature, $\eta = 7.5$ or $\eta = 5$, which reaches at most 11.89/27.79/37.31 with MRR 20.24. At $\tau = 0.03$, however, MI performance drops markedly to 9.70/23.96/32.70 with MRR 17.38, while TE remains relatively stable, with T2I results around 11.22–11.80 in R@1 and MRR around 19.59–20.10. Thus, although the TE advantage is stronger for I2T than for T2I, the TE-based objective still provides the most reliable cross-dataset behavior overall and avoids the larger degradation observed for MI under the lower-temperature setting.

Overall, these results suggest that the proposed TE-inspired proxies better preserve transferable multimodal structure under distribution shift, rather than merely fitting the source training distribution. The $\tau = 0.03$ results strengthen this conclusion by showing that TE continues to improve or maintain retrieval quality when the temperature changes, whereas MI becomes much less stable, especially for cross-dataset generalization.

## 5.2 Representation-Level Diagnostics

This section studies how the implemented TE-inspired proxy regularizer changes the learned embedding geometry, rather than attempting to directly validate the ideal optimization-step TE quantity in Eq. (3). We compare how TE- and MI-based distillation shape the learned embedding geometry on MSCOCO after 10 epochs of training. We evaluate teacher (CLIP RN50) and student (RN34) embeddings on the same MSCOCO validation samples and use a teacher-only PCA protocol. For each modality, PCA is fit only on the teacher validation embeddings, and both teacher and student embeddings are projected into this shared 2D basis. Because PCA is fit separately for the TE and MI experiments, absolute coordinates are not directly comparable across figures; accordingly, we treat these visualizations as qualitative evidence and rely primarily on quantitative teacher–student agreement metrics.

### 5.2.1 Temperature=0.07

**Modality-specific teacher–student geometry.** Figures 3 and 4 show teacher–student embedding geometry for image and text modalities under the shared teacher-defined basis. In both modalities, the TE-distilled student appears visually closer to the teacher than the MI-distilled student. Since visual inspection alone is insufficient, we additionally quantify teacher–student agreement using local-neighborhood and per-sample similarity metrics.

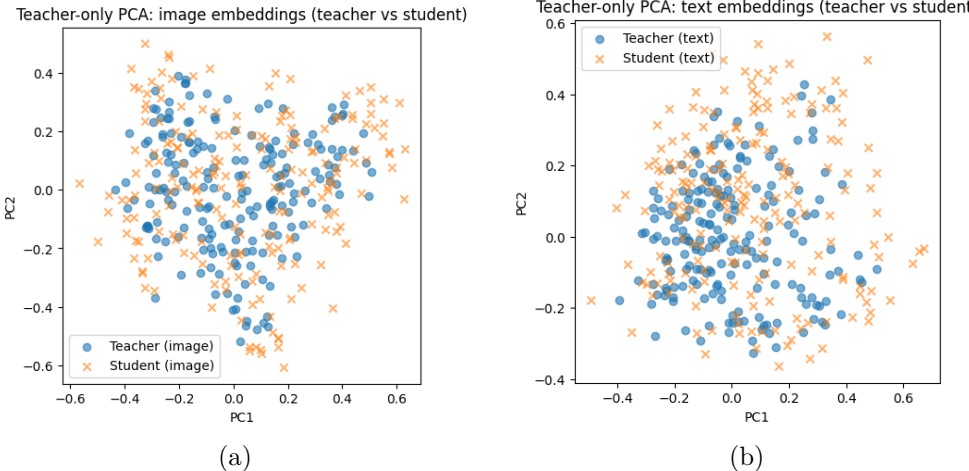

Figure 3: For $\tau = 0.07$, teacher-only PCA visualization of teacher–student embedding geometry on MSCOCO val after 10 epochs of distillation on MSCOCO train (TE-based; $\gamma = 7.5$, with $\alpha = 1$, $\beta = 50$, $\delta = 1$, $\eta = 0$, $\lambda = 0$). For each modality, PCA is fit on the teacher validation embeddings and both teacher and student embeddings are projected using this shared 2D basis. (a) Image embeddings. (b) Text embeddings.

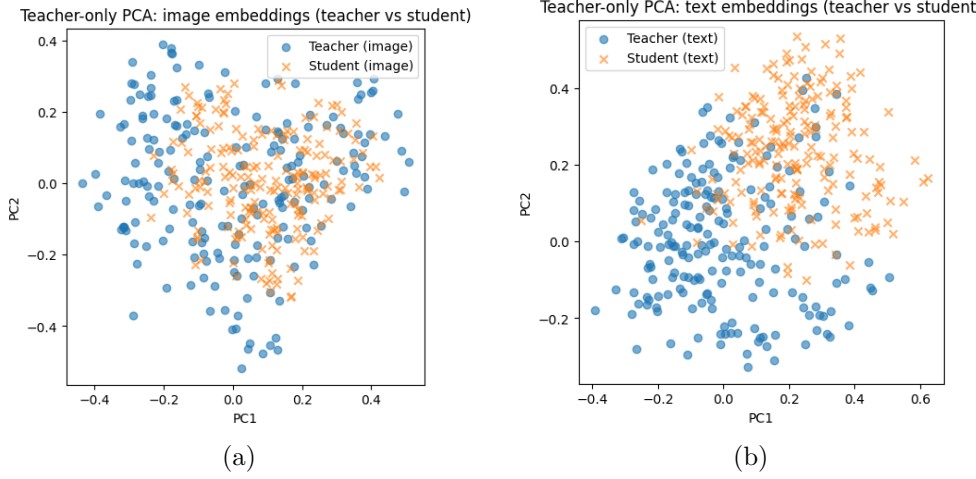

Figure 4: For $\tau = 0.07$, teacher-only PCA visualization of teacher–student embedding geometry on MSCOCO val after 10 epochs of distillation on MSCOCO train (MI-based; $\eta = 5$, with $\alpha = 1$, $\beta = 50$, $\delta = 1$, $\lambda = 0$, $\gamma = 0$). For each modality, PCA is fit on the teacher validation embeddings and both teacher and student embeddings are projected using this shared 2D basis. (a) Image embeddings. (b) Text embeddings.

**kNN neighborhood agreement and cosine similarity.** To measure local-geometry preservation, we compute kNN overlap@10 between teacher and student *image* embeddings on MSCOCO val. For each sample $i$, let $\mathcal{N}_{10}^{T}(i)$ and $\mathcal{N}_{10}^{S}(i)$ denote its 10 nearest neighbors under the teacher and student embeddings, respectively. We report

$$\frac{1}{N} \sum_{i=1}^{N} \frac{\left| \mathcal{N}_{10}^{T}(i) \cap \mathcal{N}_{10}^{S}(i) \right|}{10}, \tag{22}$$

along with the standard deviation across samples. We also report per-sample cosine similarity between matched teacher and student embeddings for both modalities. As summarized in Table 6, TE achieves higher image-embedding kNN overlap@10 with the teacher ($0.605 \pm 0.202$ vs. $0.546 \pm 0.200$), indicating stronger preservation of local neighborhood structure. TE also yields substantially higher per-sample cosine

Table 6: Retrieval and embedding-alignment metrics on MSCOCO val after 10 epochs of distillation on MSCOCO train (RN50 teacher, RN34 student) for $\tau = 0.07$. Retrieval metrics are in %, higher is better. kNN overlap@10 measures local neighborhood agreement between teacher and student image embeddings; cosine similarities are per-sample teacher–student agreement in each modality.

| | Image→Text | | | | Text→Image | | | |
|---|---|---|---|---|---|---|---|---|
| Method | R@1 | R@5 | R@10 | MRR | R@1 | R@5 | R@10 | MRR |
| TE ($\gamma = 7.5$) | 10.28 | 25.88 | 35.83 | 18.66 | 6.88 | 18.94 | 27.07 | 13.67 |
| MI ($\eta = 5$) | 8.74 | 23.19 | 33.18 | 16.71 | 7.62 | 20.65 | 29.16 | 14.82 |

| | kNN overlap@10 (img) | | $\cos(T_{\text{img}}, S_{\text{img}})$ | | $\cos(T_{\text{txt}}, S_{\text{txt}})$ | |
|---|---|---|---|---|---|---|
| Method | mean | std | mean | std | mean | std |
| TE ($\gamma = 7.5$) | 0.605 | 0.202 | 0.726 | 0.078 | 0.840 | 0.044 |
| MI ($\eta = 5$) | 0.546 | 0.200 | 0.487 | 0.054 | 0.528 | 0.048 |

agreement for both image embeddings ($0.726\pm0.078$ vs. $0.487\pm0.054$) and text embeddings ($0.840\pm0.044$ vs. $0.528\pm0.048$). Beyond Recall@K, we also evaluate MRR (Schütze et al., 2008) for both I2T and T2I retrieval in Table 6. Taken together, these results show that TE more faithfully preserves both local geometry and per-sample teacher–student correspondence than MI.

**Joint image–text structure.** We next visualize the joint image+text embedding space by pooling image and text embeddings and projecting them to 2D with teacher-only PCA fit separately per panel; circles denote image embeddings and crosses denote text embeddings. Figure 5 shows that the teacher space exhibits a clear bimodal organization, with image and text embeddings occupying distinct regions. The TE-distilled student largely preserves this global structure, maintaining a teacher-like separation between modalities. In contrast, the MI-distilled student shows heavier cross-modal mixing and weaker separation, indicating a greater departure from the teacher's joint embedding geometry. As with the modality-specific PCA plots, we interpret relative separation and overlap within each panel rather than comparing absolute coordinates across panels.

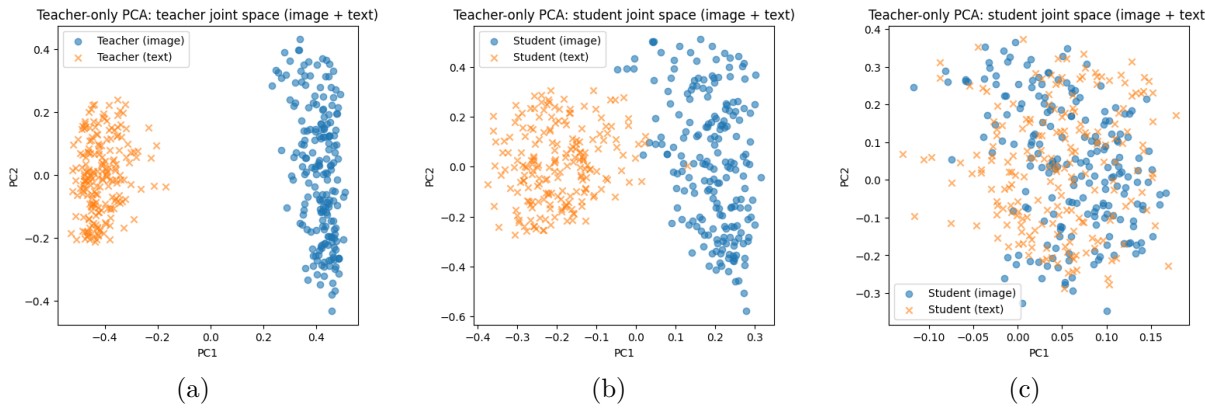

(a)  (b)  (c)

Figure 5: For $\tau = 0.07$, joint (image+text) embedding visualization on MSCOCO val after 10 epochs of distillation on MSCOCO train. We pool image (circles) and text (crosses) embeddings and project them to 2D with teacher-only PCA fit separately per panel. (a) Teacher CLIP RN50. (b) TE-distilled RN34 student ($\gamma = 7.5$; $\alpha = 1$, $\beta = 50$, $\delta = 1$, $\eta = 0$, $\lambda = 0$). (c) MI-distilled RN34 student ($\eta = 5$; $\alpha = 1$, $\beta = 50$, $\delta = 1$, $\gamma = 0$, $\lambda = 0$).

**Connection to retrieval.** The geometric differences above are also reflected in retrieval performance. Table 6 shows that TE improves Image→Text retrieval relative to MI (R@1 10.28% vs. 8.74%, R@5 25.88% vs. 23.19%, R@10 35.83% vs. 33.18%, MRR 18.66% vs. 16.71%). MI remains slightly stronger on Text→Image (R@1 7.62% vs. 6.88%, R@5 20.65% vs. 18.94%, R@10 29.16% vs. 27.07%, MRR 14.82% vs. 13.67%), indicating that improved teacher-faithfulness does not translate uniformly into gains in both retrieval directions. Overall, however, these diagnostics suggest that the proposed TE regularizer changes not only scalar training dynamics, but also the learned embedding geometry, making the student representation more teacher-faithful in both local and global structure.

### 5.2.2   Temperature=0.03

**Modality-specific teacher–student geometry.** Figures 6 and 7 show teacher–student embedding geometry for image and text modalities under the shared teacher-defined basis at the lower temperature $\tau = 0.03$. As in the $\tau = 0.07$ case, the TE-distilled student appears visually closer to the teacher than the MI-distilled student in both modalities. For image embeddings, the TE student better preserves the teacher's broad cluster arrangement, including the denser central-right region and the more diffuse left-side structure, whereas the MI student exhibits a more noticeable shift and compression relative to the teacher. For text embeddings, the difference is even clearer: the TE student remains largely co-located with the teacher's dominant clusters, while the MI student shows a systematic upward shift and stronger separation from the teacher cloud, suggesting weaker preservation of the teacher-defined text geometry. Although these PCA plots are qualitative, they are consistent with the view that TE better maintains teacher-faithful embedding structure under the lower-temperature regime.

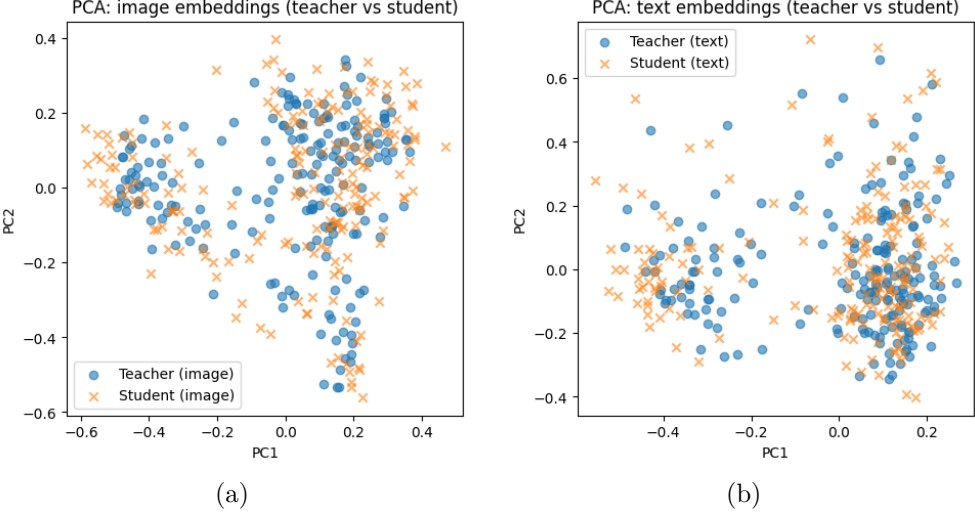

Figure 6: For $\tau = 0.03$, teacher-only PCA visualization of teacher–student embedding geometry on MSCOCO val after 10 epochs of distillation on MSCOCO train (TE-based; $\gamma = 7.5$, with $\alpha = 1$, $\beta = 50$, $\delta = 1$, $\eta = 0$, $\lambda = 0$). For each modality, PCA is fit on the teacher validation embeddings and both teacher and student embeddings are projected using this shared 2D basis. (a) Image embeddings. (b) Text embeddings.

**kNN neighborhood agreement and cosine similarity.** We next quantify teacher–student agreement using the same local-neighborhood and per-sample similarity diagnostics as above. Table 7 shows that TE again achieves stronger alignment with the teacher than MI across all geometry-based metrics. In particular, TE yields higher image-embedding kNN overlap@10 ($0.487 \pm 0.174$ vs. $0.429 \pm 0.173$), indicating better preservation of local neighborhood structure. The gap is even larger for per-sample cosine agreement: for image embeddings, TE reaches $0.809 \pm 0.058$ versus $0.469 \pm 0.042$ for MI, and for text embeddings, TE achieves $0.882 \pm 0.048$ versus $0.522 \pm 0.048$. These improvements are substantial and suggest that, at $\tau = 0.03$, the TE-distilled student remains much more closely aligned with the teacher in both modalities, especially at the level of pointwise correspondence.

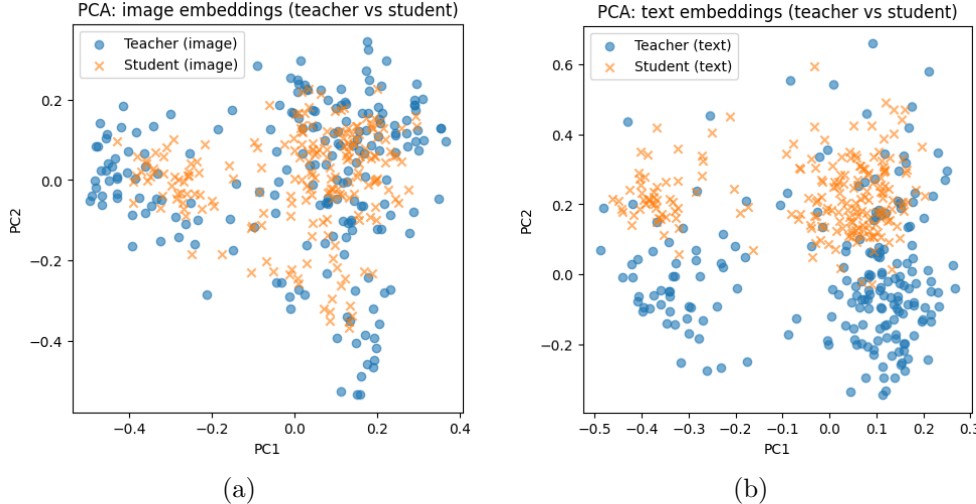

Figure 7: For $\tau = 0.03$, teacher-only PCA visualization of teacher–student embedding geometry on MSCOCO val after 10 epochs of distillation on MSCOCO train (MI-based; $\eta = 5$, with $\alpha = 1$, $\beta = 50$, $\delta = 1$, $\lambda = 0$, $\gamma = 0$). For each modality, PCA is fit on the teacher validation embeddings and both teacher and student embeddings are projected using this shared 2D basis. (a) Image embeddings. (b) Text embeddings.

Table 7: For $\tau = 0.03$, retrieval and embedding-alignment metrics on MSCOCO val after 10 epochs of distillation on MSCOCO train (RN50 teacher, RN34 student). Retrieval metrics are in %, higher is better. kNN overlap@10 measures local neighborhood agreement between teacher and student image embeddings; cosine similarities are per-sample teacher–student agreement in each modality.

| | Image→Text | | | | Text→Image | | | |
|---|---|---|---|---|---|---|---|---|
| Method | R@1 | R@5 | R@10 | MRR | R@1 | R@5 | R@10 | MRR |
| TE ($\gamma = 7.5$) | 10.85 | 26.47 | 36.29 | 19.21 | 6.84 | 18.76 | 26.94 | 13.59 |
| MI ($\eta = 5$) | 9.14 | 23.67 | 33.14 | 17.09 | 6.49 | 18.23 | 26.44 | 13.18 |

| | kNN overlap@10 (img) | | $\cos(T_{\text{img}}, S_{\text{img}})$ | | $\cos(T_{\text{txt}}, S_{\text{txt}})$ | |
|---|---|---|---|---|---|---|
| Method | mean | std | mean | std | mean | std |
| TE ($\gamma = 7.5$) | 0.487 | 0.174 | 0.809 | 0.058 | 0.882 | 0.048 |
| MI ($\eta = 5$) | 0.429 | 0.173 | 0.469 | 0.042 | 0.522 | 0.048 |

**Joint image–text structure.** Figure 8 further compares the joint image+text embedding space by pooling both modalities and projecting them to 2D under teacher-only PCA. The teacher space again exhibits a clear bimodal organization, with image and text embeddings occupying distinct regions. The TE-distilled student preserves this overall geometry more faithfully: the image and text clouds remain clearly separated, and the student layout broadly mirrors the teacher's two-cluster structure. In contrast, the MI-distilled student shows a less teacher-like organization, with more distortion in the placement of the text cluster relative to the image cluster and weaker preservation of the teacher's global arrangement. Thus, beyond modality-specific alignment, TE also appears to better preserve the teacher's cross-modal global geometry at $\tau = 0.03$.

**Connection to retrieval.** These geometric differences are reflected in retrieval performance. Table 7 shows that TE outperforms MI in both retrieval directions at $\tau = 0.03$. For Image→Text, TE improves over MI from 9.14/23.67/33.14 with MRR 17.09 to 10.85/26.47/36.29 with MRR 19.21. For Text→Image, TE also remains slightly better, improving from 6.49/18.23/26.44 with MRR 13.18 to 6.84/18.76/26.94 with

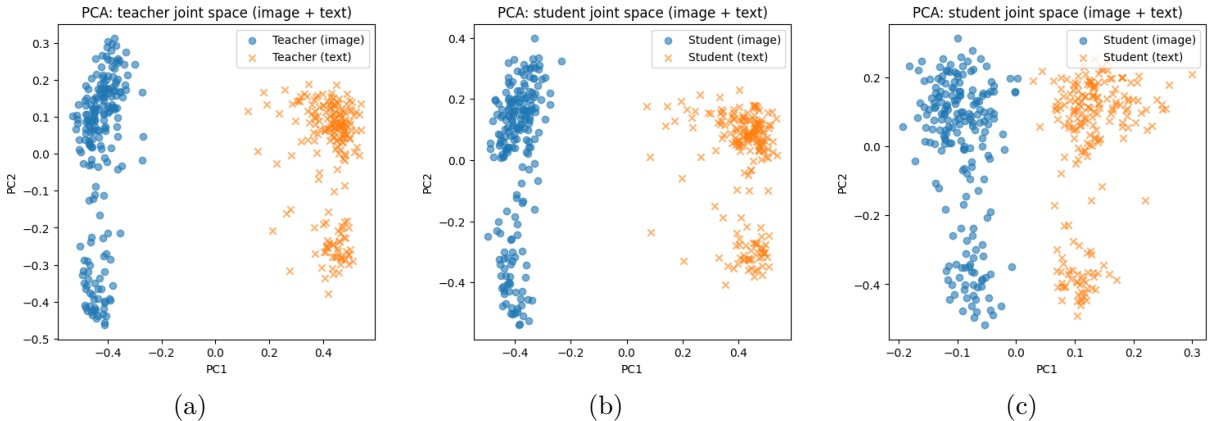

Figure 8: For $\tau = 0.03$, joint (image+text) embedding visualization on MSCOCO val after 10 epochs of distillation on MSCOCO train. We pool image (circles) and text (crosses) embeddings and project them to 2D with teacher-only PCA fit separately per panel. (a) Teacher CLIP RN50. (b) TE-distilled RN34 student ($\gamma = 7.5$; $\alpha = 1$, $\beta = 50$, $\delta = 1$, $\eta = 0$, $\lambda = 0$). (c) MI-distilled RN34 student ($\eta = 5$; $\alpha = 1$, $\beta = 50$, $\delta = 1$, $\gamma = 0$, $\lambda = 0$).

MRR 13.59. Compared with the $\tau = 0.07$ setting, where MI was slightly stronger on T2I retrieval, the lower-temperature regime yields a cleaner advantage for TE across both directions. Taken together, the PCA visualizations, neighborhood-overlap statistics, cosine similarities, and retrieval metrics all support the same conclusion: at $\tau = 0.03$, TE produces a more teacher-faithful student representation than MI, both locally and globally, and this stronger geometric fidelity is accompanied by more consistent retrieval gains.

### 5.3 Teacher: ViT-B/16, Student: ResNet-34

We distill an RN34-based VLM student from a CLIP ViT-B/16 teacher. For this experiment, the loss function incorporates weighting factors $\alpha = 1.0$, $\beta = 100$, $\delta = 1.0$, $\gamma = 5.0$, along with a temperature parameter $\tau = 0.07$. These weighting parameters were chosen based on the relative contribution of each loss term to the total loss during training. For the experiments on the MSCOCO dataset, due to the large scale of both the model and the dataset, we trained the student model for 6 epochs. Each experiment (i.e., each row in Table 8) required approximately 10 hours on a Google Colab T4 GPU with high-RAM. For the Flickr8k experiments, we used a Google Colab A100 GPU and trained for 10 epochs. Given the smaller dataset size, each experiment (i.e., each row in Table 10) took around 30 minutes to complete.

Our experiments (Tables 8 and 10) show that maximizing the information flow from teacher to student via TE delivers the single largest boost among all losses. Loss functions with TE leading to the 3–4 percentage point (pp) gains on MSCOCO and the 8–12pp gains on the low-resource Flickr8k benchmarks. These results establish TE as a principled and highly effective regularizer for cross-modal knowledge distillation.

Table 9 summarizes the hyperparameter sensitivity of zero-shot retrieval when distilling a student RN34 from a ViT-B/16 teacher on MSCOCO. Overall, introducing the TE reward is consistently beneficial, especially for T2I retrieval: compared to the baseline without TE ($\gamma = 0$), a small TE weight ($\gamma = 1$) yields the strongest T2I scores in the table (R@1/5/10 of 6.75/18.74/26.91), indicating that TE-based regularization can effectively strengthen bidirectional alignment under this teacher-student pairing. Increasing $\gamma$ beyond this regime provides diminishing returns and can reduce performance (e.g., $\gamma \geq 5$), suggesting that overly strong TE weighting may over-regularize the student. We also observe that increasing the MSE weight $\beta$ can improve I2T retrieval: the best I2T performance is achieved at $\beta = 100$ with $\gamma = 5$ (R@1/5/10 of 8.48/22.02/30.99), highlighting the importance of feature-level matching for transferring visual semantics from ViT-B/16 to RN34. In contrast, varying the MI weight $\eta$ in isolation ($\gamma = 0$) produces moderate gains over the baseline but does not surpass the best TE-regularized setting for T2I, and combining MI, MSE-$\Delta$ ($\lambda > 0$), and TE does not yield additive improvements in this sweep. Taken together, these results indicate

Table 8: Comparison of zero-shot retrieval performance (Recall@k) of RN34-based VLM student with teacher CLIP ViT-B/16 on MSCOCO in VLM distillation with different loss functions. All Loss Function: CL + KL + MSE + ICL - TE1 - TE2. The hyperparameter settings: $\alpha = 1.0$, $\beta = 100$, $\delta = 1.0$, $\gamma = 5.0$.

| Model and Loss Function | I2T Retrieval (R) | | | T2I Retrieval (R) | | |
|---|---|---|---|---|---|---|
| | R@1 | R@5 | R@10 | R@1 | R@5 | R@10 |
| **Teacher Model (ViT-B/16)** | 29.60% | 50.95% | 60.35% | 15.41% | 31.54% | 39.96% |
| **Student Models (RN34)** | | | | | | |
| CL Only (Oord et al., 2018) | 4.66% | 14.10% | 21.28% | 3.78% | 11.95% | 18.40% |
| CL + MSE (Yang et al., 2024) | 4.55% | 14.27% | 21.36% | 3.79% | 11.99% | 18.44% |
| CL + KL (Li et al., 2024b) | 4.70% | 14.46% | 22.21% | 4.58% | 14.15% | 21.32% |
| CL + ICL (Yang et al., 2024) | 5.52% | 16.25% | 23.92% | 4.55% | 13.86% | 20.79% |
| CL - TE1 | 7.24% | 19.88% | 28.55% | 5.68% | 16.22% | 23.71% |
| CL - TE2 | 7.02% | 20.26% | 29.46% | 5.83% | 16.54% | 24.27% |
| CL - TE1 - TE2 | 7.44% | 20.24% | 29.01% | 5.78% | 16.35% | 23.90% |
| CL + KL + MSE - TE1 | 7.74% | 21.05% | 30.14% | 5.86% | 16.85% | 24.64% |
| CL + KL + ICL - TE1 | 7.64% | 21.07% | 30.16% | **6.20%** | **17.64%** | **25.66%** |
| CL + KL + MSE + ICL - TE1 | 7.90% | 21.49% | 30.42% | 6.15% | 17.48% | 25.35% |
| ALL Loss Function | **8.48%** | **22.02%** | **30.99%** | 6.00% | 17.01% | 24.80% |

that (i) modest TE regularization is most helpful for T2I, while (ii) stronger feature distillation (larger $\beta$) primarily benefits I2T, and careful balancing is required to avoid over-regularization when multiple auxiliary terms are enabled.

Table 9: Comparison of zero-shot retrieval performance (Recall@k) of RN34-based VLM student with teacher CLIP ViT-B/16 on MSCOCO.

| $\alpha$ | $\beta$ | $\delta$ | $\eta$ | $\lambda$ | $\gamma$ | I2T R@1 | I2T R@5 | I2T R@10 | T2I R@1 | T2I R@5 | T2I R@10 |
|---|---|---|---|---|---|---|---|---|---|---|---|
| 1 | 50 | 1 | 0 | 0 | 0 | 5.81% | 17.30% | 25.89% | 5.60% | 16.64% | 24.46% |
| 1 | 50 | 1 | 0 | 0 | 1 | 7.32% | 20.41% | 29.58% | **6.75%** | **18.74%** | **26.91%** |
| 1 | 50 | 1 | 0 | 0 | 2.5 | 7.68% | 21.16% | 30.20% | 6.70% | **18.74%** | 26.78% |
| 1 | 50 | 1 | 0 | 0 | 5 | 7.87% | 21.47% | 30.74% | 5.98% | 17.21% | 24.96% |
| 1 | 100 | 1 | 0 | 0 | 5 | **8.48%** | **22.02%** | **30.99%** | 6.00% | 17.01% | 24.80% |
| 1 | 50 | 1 | 0 | 0 | 7.5 | 7.90% | 21.27% | 30.22% | 5.92% | 16.69% | 24.42% |
| 1 | 50 | 1 | 0 | 0 | 10 | 7.49% | 20.56% | 29.51% | 5.47% | 16.01% | 23.38% |
| 1 | 50 | 1 | 1 | 0 | 0 | 7.08% | 19.72% | 28.44% | 6.40% | 17.94% | 25.86% |
| 1 | 50 | 1 | 2.5 | 0 | 0 | 7.09% | 20.00% | 29.04% | 6.31% | 17.97% | 25.96% |
| 1 | 50 | 1 | 5 | 0 | 0 | 7.28% | 20.05% | 29.24% | 6.08% | 17.37% | 25.30% |
| 1 | 100 | 1 | 5 | 0 | 0 | 7.04% | 20.17% | 29.08% | 6.21% | 17.43% | 25.37% |
| 1 | 50 | 1 | 7.5 | 0 | 0 | 6.69% | 19.53% | 28.25% | 6.01% | 16.96% | 24.70% |
| 1 | 50 | 1 | 2.5 | 0 | 5 | 7.29% | 20.45% | 29.37% | 5.68% | 16.56% | 24.13% |
| 1 | 50 | 1 | 2.5 | 50 | 2.5 | 7.33% | 20.31% | 29.40% | 6.00% | 17.12% | 24.91% |
| 1 | 50 | 1 | 1 | 0 | 7.5 | 7.43% | 20.43% | 29.56% | 5.75% | 16.48% | 24.00% |
| 1 | 50 | 1 | 1 | 50 | 7.5 | 7.42% | 20.51% | 29.25% | 5.65% | 16.33% | 24.02% |

Performance continues to improve as $\gamma$ increases up to 5, with the best I2T results observed at $\gamma = 7.5$ (7.90% Recall@1, 30.22% Recall@10). However, the T2I results peak earlier, with $\gamma = 1$ providing the strongest Recall@1 and Recall@5 values, while larger $\gamma$ values cause a mild decline. This indicates that while TE is generally beneficial, excessively weighting it can distort the loss balance and harm retrieval performance on certain tasks. Overall, these results demonstrate two key points: (i) TE is a crucial component of the loss, consistently lifting performance above the no-TE baseline, and (ii) the optimal $\gamma$ value is task-

dependent, suggesting that moderate TE weighting is sufficient to maximize the gains from information-theoretic regularization.

Table 10: Zero-shot retrieval performance (Recall@k) on Flickr8k. The RN34-based VLM student is distilled from the teacher model (CLIP ViT-B/16). All Loss Function: CL + MSE + KL + ICL - TE1 - TE2. The weighting factors for loss functions: $\alpha = 1.0$, $\beta = 100$, $\delta = 1.0$, $\gamma = 5.0$.

| Model and Loss Function | I2T Retrieval (R) | | | T2I Retrieval (R) | | |
|---|---|---|---|---|---|---|
| | R@1 | R@5 | R@10 | R@1 | R@5 | R@10 |
| **Teacher Model (ViT-B/16)** | 57.41% | 82.70% | 90.61% | 55.02% | 81.63% | 87.64% |
| **Student Models (RN34)** | | | | | | |
| CL (Oord et al., 2018) | 21.09% | 46.95% | 59.47% | 17.53% | 42.59% | 55.26% |
| CL + MSE (Yang et al., 2024) | 21.17% | 46.46% | 58.98% | 16.26% | 42.83% | 55.37% |
| CL + KL (Li et al., 2024b) | 24.38% | 51.32% | 63.84% | 19.59% | 46.97% | 60.03% |
| CL + ICL (Yang et al., 2024) | 26.44% | 52.14% | 65.32% | 20.44% | 47.69% | 61.61% |
| CL - TE1 | 28.91% | 57.17% | 69.19% | 22.98% | 50.69% | 63.79% |
| CL - TE2 | 30.07% | 57.41% | 68.45% | 23.67% | 52.04% | 65.44% |
| CL - TE1 - TE2 | 28.42% | 58.90% | 70.02% | 22.59% | 51.10% | 64.79% |
| CL + KL + MSE - TE1 | 29.90% | 59.97% | 71.83% | 22.34% | 51.73% | 65.44% |
| CL + KL + ICL - TE1 | 28.42% | 56.84% | 71.17% | 23.05% | 51.53% | 65.09% |
| CL + KL + MSE + ICL - TE1 | 29.82% | 60.71% | 71.75% | 23.53% | 51.47% | 64.99% |
| All Loss Function | **33.28%** | **64.33%** | **73.97%** | **26.36%** | **56.18%** | **69.64%** |

Flickr8k has a small number of training/evaluation samples which can amplify variance and make performance more sensitive to hyperparameter choices. Motivated by this, we additionally evaluate on Flickr30k (Young et al., 2014), a larger and more diverse captioned dataset, to provide a more robust assessment of our distillation objective and to verify that the observed benefits of TE-based regularization persist under a less data-limited setting. Table 11 summarizes the zero-shot retrieval results on Flickr30k when distilling an RN34 student from a ViT-B/16 teacher under different loss weightings. Several consistent trends emerge. First, introducing the TE reward ($\gamma > 0$) generally improves retrieval over the baseline without TE, particularly for moderate TE weights. Increasing $\gamma$ from 0 to 2.5–5 yields clear gains in both I2T and T2I Recall@K, with I2T R@1 improving from 34.71% to over 40%, and T2I R@1 rising from 29.35% to above 31%. However, very large TE weights (e.g., $\gamma = 7.5$) begin to slightly degrade performance, indicating that excessive regularization may oversuppress the contrastive signal. Second, adjusting the KL weight shows that a stronger feature-matching term ($\alpha = 5$) in combination with moderate TE ($\gamma = 5$) leads to the strongest overall I2T performance (R@1 = 41.32%, R@10 = 80.47%), suggesting complementary effects between KL alignment and TE-based guidance. In contrast, increasing $\delta$ or $\lambda$ yields smaller and less consistent gains, whereas introducing MI ($\eta > 0$) can modestly improve T2I performance. Overall, these results reinforce the pattern observed on Flickr8k: moderate TE regularization enhances the semantic structure preserved in the distilled model, while careful balancing with KL produces the most robust improvements.

### 5.4 Computational Overhead and Robustness of the Empirical Evaluation

Appendix E explains why computing exact TE is intractable, but it does not quantify the additional cost of our TE surrogates (TE1 and TE2) relative to a standard distillation run. To make this overhead explicit, we measured wall-clock time and FLOPS for two settings on Google Colab A100 GPUs: training with the baseline loss (CL + MSE + KL + ICL) and training with the objective including TE (CL + MSE + KL + ICL - TE). We report the total time for 10 training epochs plus evaluation, as well as the total FLOPS and the portion attributable to the TE terms, in Table 12.

The reported FLOPS correspond to the entire training and evaluation run with the loss CL+KL+MSE+ICL-TE. The FLOPS contributed by the TE surrogates are six orders of magnitude smaller than the total, and the wall-clock times with and without TE only have 1 minute difference, indicating that the additional

Table 11: Comparison of zero-shot retrieval performance (Recall@k) of RN34-based VLM student with teacher CLIP ViT-B/16 on **Flickr30k**.

| $\alpha$ | $\beta$ | $\delta$ | $\eta$ | $\lambda$ | $\gamma$ | I2T R@1 | I2T R@5 | I2T R@10 | T2I R@1 | T2I R@5 | T2I R@10 |
|---|---|---|---|---|---|---|---|---|---|---|---|
| 1 | 50 | 1 | 0 | 0 | 0 | 34.71% | 62.43% | 74.85% | 29.35% | 57.91% | 70.02% |
| 1 | 50 | 1 | 0 | 0 | 1 | 38.36% | 66.67% | 76.43% | 31.40% | 60.95% | 71.93% |
| 1 | 50 | 1 | 0 | 0 | 2.5 | 40.34% | 67.16% | 77.51% | 31.79% | 60.89% | 72.01% |
| 1 | 50 | 1 | 0 | 0 | 5 | 40.24% | 68.54% | 78.80% | 31.01% | 60.37% | 72.19% |
| 1 | 50 | 1 | 0 | 0 | 7.5 | 37.28% | 66.86% | 78.11% | 30.02% | 58.93% | 71.28% |
| 1 | 50 | 1 | 0 | 0 | 10 | 39.35% | 66.67% | 76.73% | 30.20% | 58.64% | 70.43% |
| 5 | 50 | 1 | 0 | 0 | 5 | **42.31%** | **69.92%** | **80.37%** | 31.24% | 61.08% | 72.19% |
| 1 | 100 | 1 | 0 | 0 | 5 | 37.87% | 67.36% | 78.90% | 31.12% | 61.24% | 72.31% |
| 1 | 50 | 5 | 0 | 0 | 5 | 37.38% | 67.26% | 77.61% | 31.36% | 60.37% | 71.05% |
| 1 | 50 | 1 | 0 | 50 | 0 | 36.49% | 64.20% | 75.54% | 31.10% | 58.78% | 70.26% |
| 1 | 50 | 1 | 0 | 100 | 0 | 36.59% | 64.99% | 76.63% | 30.08% | 59.90% | 71.76% |
| 1 | 50 | 1 | 1 | 1 | 0 | 36.59% | 66.77% | 78.30% | 32.29% | **61.40%** | **73.06%** |
| 1 | 50 | 1 | 2.5 | 0 | 0 | 37.77% | 66.47% | 77.22% | **32.96%** | 60.79% | 71.64% |
| 1 | 50 | 1 | 5 | 0 | 0 | 39.25% | 65.78% | 76.73% | 31.52% | 60.02% | 71.91% |
| 1 | 50 | 1 | 7.5 | 0 | 0 | 39.55% | 68.05% | 78.40% | 31.95% | 60.00% | 71.48% |
| 5 | 50 | 1 | 7.5 | 0 | 0 | 37.77% | 67.36% | 77.91% | 30.14% | 58.21% | 69.92% |
| 1 | 50 | 1 | 2.5 | 50 | 2.5 | 40.43% | 67.06% | 78.11% | 32.64% | 60.65% | 72.21% |
| 1 | 50 | 1 | 2.5 | 0 | 5 | 38.66% | 66.77% | 78.80% | 30.51% | 59.47% | 71.95% |
| 1 | 50 | 1 | 2.5 | 50 | 5 | 39.94% | 68.84% | 78.40% | 31.32% | 59.68% | 71.34% |
| 1 | 50 | 1 | 1 | 0 | 7.5 | 36.39% | 67.65% | 77.91% | 30.73% | 59.39% | 71.48% |
| 1 | 50 | 1 | 1 | 50 | 7.5 | 38.17% | 65.78% | 77.02% | 30.08% | 58.09% | 70.91% |
| 1 | 50 | 1 | 5 | 50 | 7.5 | 36.79% | 65.48% | 76.82% | 30.41% | 58.86% | 70.49% |
| **Teacher (ViT-B/16)** | | | | | | 81.76% | 97.14% | 98.92% | 62.66% | 86.23% | 91.36% |

Table 12: Computation and time cost comparison for training (10 epochs) and evaluation on MSCOCO.

| Teacher | Student | CL+KL+MSE+ICL–TE | CL+KL+MSE+ICL | FLOPS | FLOPS from TE |
|---|---|---|---|---|---|
| RN50 | RN34 | 6h 19m | 6h 18m | $3.44 \times 10^{16}$ | $1.79 \times 10^{10}$ |
| ViT-B/16 | RN34 | 5h 33m | 5h 32m | $2.60 \times 10^{16}$ | $1.49 \times 10^{10}$ |

computational cost of TE1/TE2 is negligible in practice. The experiment with teacher ViT-B/16 is faster because its image encoder has hidden dimension 768, whereas RN50 uses hidden dimension 1024.

To assess the robustness of our empirical findings, we run each configuration with five independent random seeds and report mean ± standard deviation for all retrieval metrics in Table 13. On MSCOCO, the standard deviations are very small (typically on the order of 0.1–0.3), indicating that both I2T and T2I performance are highly stable under different initializations and data shuffling. Flickr30k exhibits slightly larger but still modest variability, suggesting mild sensitivity while remaining well-behaved. In contrast, Flickr8k shows noticeably larger standard deviations (around 1.0), which is expected given its much smaller size and higher sampling noise. Overall, these trends confirm that our conclusions are robust on larger benchmarks, with variability naturally increasing on smaller datasets such as Flickr8k.

Beyond Recall@K, we also evaluate MRR (Schütze et al., 2008) for both I2T and T2I retrieval in Table 14. Table 13 and Table 14 also compare the TE-based distillation term (settings with $\eta = 0$, $\gamma > 0$) against the MI-based baseline ($\eta > 0$, $\gamma = 0$). Across all four teacher–student–dataset combinations, TE consistently achieves higher I2T performance than MI, both in Recall@K and in MRR. For T2I, TE is competitive with or better than MI on most metrics, with MI only occasionally matching or slightly exceeding TE on a single Recall@K value. On the larger MSCOCO and Flickr30k benchmarks, the gains from TE are especially clear,

Table 13: Zero-shot retrieval performance (mean $\pm$ std, %) of RN34-based VLM students distilled from different CLIP teachers on multiple datasets ($\delta = 1, \lambda = 0$).

| Teacher | Dataset | $\alpha$ | $\beta$ | $\eta$ | $\gamma$ | I2T Recall@K | | | T2I Recall@K |
| --- | --- | --- | --- | --- | --- | --- | --- | --- | --- |
| | | | | | | R@1 | R@5 | R@10 | R@1 / R@5 / R@10 |
| RN50 | MSCOCO | 1 | 50 | 0 | 7.5 | 10.08±0.17 | 26.10±0.19 | 36.22±0.21 | 7.12±0.19 / 19.35±0.27 / 27.85±0.23 |
| RN50 | MSCOCO | 1 | 50 | 5 | 0 | 8.35±0.22 | 23.34±0.16 | 33.25±0.24 | 7.35±0.28 / 20.57±0.18 / 28.95±0.26 |
| RN50 | Flickr8k | 1 | 50 | 0 | 5 | 35.81±1.12 | 65.67±1.13 | 77.47±1.19 | 26.74±1.11 / 56.02±1.03 / 69.22±1.02 |
| RN50 | Flickr8k | 1 | 50 | 5 | 0 | 33.22±1.15 | 64.18±1.09 | 76.90±1.08 | 26.32±1.21 / 57.12±1.12 / 69.31±1.06 |
| ViT-B/16 | MSCOCO | 1 | 100 | 0 | 5 | 8.01±0.21 | 21.71±0.18 | 30.75±0.16 | 6.03±0.23 / 17.27±0.15 / 25.03±0.18 |
| ViT-B/16 | MSCOCO | 1 | 50 | 5 | 0 | 7.07±0.16 | 19.98±0.22 | 28.97±0.17 | 6.13±0.26 / 17.46±0.16 / 25.30±0.21 |
| ViT-B/16 | Flickr30k | 5 | 50 | 0 | 5 | 42.24±0.79 | 69.42±0.84 | 79.84±0.89 | 31.46±0.82 / 60.56±0.69 / 72.11±0.62 |
| ViT-B/16 | Flickr30k | 1 | 50 | 5 | 0 | 38.75±0.81 | 65.92±0.78 | 76.23±0.83 | 31.21±0.72 / 60.14±0.64 / 71.70±0.68 |

Table 14: Comparison of zero-shot MRR (mean $\pm$ std) with different settings ($\delta = 1$, $\lambda = 0$).

| Teacher | Student | Dataset | $\alpha$ | $\beta$ | $\eta$ | $\gamma$ | I2T MRR (%) | T2I MRR (%) |
| --- | --- | --- | --- | --- | --- | --- | --- | --- |
| RN50 | RN34 | MSCOCO | 1 | 50 | 0 | 7.5 | 18.32±0.12 | 13.93±0.19 |
| RN50 | RN34 | MSCOCO | 1 | 50 | 5 | 0 | 16.46±0.14 | 14.34±0.23 |
| RN50 | RN34 | Flickr8k | 1 | 50 | 0 | 5 | 49.03±1.02 | 39.09±1.15 |
| RN50 | RN34 | Flickr8k | 1 | 50 | 5 | 0 | 44.62±1.17 | 38.47±1.12 |
| ViT-B/16 | RN34 | MSCOCO | 1 | 100 | 0 | 5 | 15.52±0.15 | 12.46±0.11 |
| ViT-B/16 | RN34 | MSCOCO | 1 | 50 | 5 | 0 | 14.23±0.22 | 12.63±0.09 |
| ViT-B/16 | RN34 | Flickr30k | 5 | 50 | 0 | 5 | 54.79±0.74 | 45.38±0.65 |
| ViT-B/16 | RN34 | Flickr30k | 1 | 50 | 5 | 0 | 52.21±0.62 | 45.02±0.71 |

indicating that the TE surrogate provides a stronger and more reliable signal for shaping the multimodal embedding space. On the smaller Flickr8k dataset, TE still yields higher I2T MRR and Recall@K than MI despite the higher variance, suggesting that geometric alignment of teacher–student updates is more effective than MI-based matching under capacity mismatch. Overall, TE outperforms MI in our settings, improving the ranking quality of the student model while maintaining stable training dynamics.

## 5.5 More Competitive Teacher Model

Table 15 reports zero-shot retrieval results on MSCOCO at $\tau = 0.07$ when distilling an RN34 student from a substantially stronger teacher, CLIP RN50×16. Across the explored settings, the student consistently outperforms the corresponding RN50/RN34 and ViT-B/16/RN34 configurations, suggesting that a more competitive teacher provides richer supervisory signal under the same distillation framework. As in the earlier experiments, moderate TE regularization remains beneficial. In particular, with $\gamma \in \{2.5, 5\}$, the student achieves the best overall trade-offs: $\gamma = 2.5$ yields the strongest T2I performance, reaching R@1/R@5/R@10 of 8.40/22.45/31.50, while increasing the feature distillation weight to $\beta = 100$ with $\gamma = 5$ gives the best I2T result, reaching 10.75/27.20/37.41.

Larger TE weights, such as $\gamma = 7.5$, slightly reduce both I2T and T2I performance, again suggesting diminishing returns and mild over-regularization when the reward term becomes too strong. By comparison, MI-only settings ($\eta \in \{1, 2.5, 5\}$ with $\gamma = 0$) remain competitive but are consistently slightly weaker than the best TE-regularized configurations. Adding further auxiliary terms, such as MI together with MSE-$\Delta$, does not yield clear additional gains within this sweep. Finally, despite the strong teacher performance (I2T R@1 of 32.14% and T2I R@1 of 17.08%), a substantial teacher–student gap remains, highlighting the difficulty of compressing RN50×16 into RN34. Even so, these results show that carefully tuned TE regularization and feature distillation can still produce meaningful retrieval improvements under a substantially stronger teacher.

We emphasize that this experiment should be interpreted as a capacity-gap stress test rather than a full scaling study. Although the RN50×16 teacher is substantially stronger than the RN34 student, a large

Table 15: Comparison of zero-shot retrieval performance (Recall@k) in percentage of RN34-based VLM student with teacher CLIP RN50×16 on MSCOCO.

| $\alpha$ | $\beta$ | $\delta$ | $\eta$ | $\lambda$ | $\gamma$ | I2T R@1 | I2T R@5 | I2T R@10 | T2I R@1 | T2I R@5 | T2I R@10 |
|---|---|---|---|---|---|---|---|---|---|---|---|
| 1 | 50 | 1 | 0 | 0 | 2.5 | 10.29% | 26.18% | 36.56% | **8.40%** | **22.45%** | **31.50%** |
| 1 | 50 | 1 | 0 | 0 | 5 | 10.52% | 26.88% | 37.19% | 7.85% | 21.11% | 29.80% |
| 1 | 100 | 1 | 0 | 0 | 5 | **10.75%** | **27.20%** | **37.41%** | 8.01% | 21.32% | 29.86% |
| 1 | 50 | 1 | 0 | 0 | 7.5 | 9.99% | 26.47% | 36.48% | 7.66% | 20.51% | 29.06% |
| 1 | 50 | 1 | 1 | 0 | 0 | 9.64% | 25.14% | 35.55% | 7.64% | 20.97% | 29.79% |
| 1 | 50 | 1 | 2.5 | 0 | 0 | 9.93% | 25.94% | 35.97% | 7.84% | 21.26% | 30.22% |
| 1 | 50 | 1 | 5 | 0 | 0 | 9.81% | 25.57% | 35.87% | 7.79% | 20.99% | 29.80% |
| 1 | 50 | 1 | 2.5 | 50 | 5 | 10.01% | 25.97% | 36.29% | 7.77% | 20.88% | 29.58% |
| 1 | 50 | 1 | 1 | 50 | 7.5 | 10.08% | 26.00% | 36.21% | 7.49% | 20.21% | 28.73% |
| **Teacher (RN50×16)** | | | | | | 32.14% | 53.62% | 62.67% | 17.08% | 33.54% | 42.00% |

absolute teacher-student performance gap remains. Thus, the result does not show that TE-based distillation closes the gap to a much larger teacher. Instead, it shows that the TE-inspired regularizer can still provide student-side improvements over comparable objectives under a larger teacher. Establishing true scaling behavior will require experiments with larger students, stronger VLM teachers, and larger-scale training data, which we leave for future work.

## 5.6 Distillation for Classification

### 5.6.1 Within-Dataset Zero-Shot Classification on Food-101

We have evaluated our TE-based distillation on Food-101 (Bossard et al., 2014), a challenging benchmark dataset for large-scale food recognition. Food-101 contains 101 categories with a total of 101,000 images, split into 75,750 images for training and 25,250 images for testing. This dataset is particularly suitable for evaluating knowledge transfer since it combines significant intra-class variation with a large number of categories, which makes direct zero-shot transfer difficult for a smaller-capacity student network.

In our setup, the teacher is a ResNet-50 (RN50) model, and the student is a smaller ResNet-34 (RN34). Importantly, during distillation, the student is trained without direct access to the ground truth labels. Instead, it learns only from the outputs of the teacher, thereby relying entirely on the transferred information. This design allows us to directly measure the effectiveness of the proposed TE-based framework in capturing and transferring generalizable knowledge from teacher to student.

Table 16 summarizes the zero-shot classification accuracy of the student RN34 under different weightings of the loss components (cf. Eq. 19), alongside the teacher RN50 baseline. Several key observations emerge. First, the naive baseline where $\gamma = 0$ (i.e., without TE) performs better than the teacher in terms of Top-1 accuracy but slightly underperforms in Top-5 accuracy. Second, once TE is introduced ($\gamma > 0$), we observe consistent improvements across both Top-1 and Top-5 accuracy. For instance, setting $\gamma = 2.5$ increases the student's Top-1 accuracy to 82.46% and Top-5 accuracy to 96.23%, surpassing the teacher by significant margins. Larger $\gamma$ values generally sustain these gains, with $\gamma = 7.5$ yielding the best Top-5 performance (96.62%), and an alternative setting with $\alpha = 5$ and $\gamma = 2.5$ providing the overall best Top-1 accuracy (82.91%). These trends suggest that TE contributes complementary signal during distillation that is not fully captured by conventional loss terms. Each experiment (each row) in Table 16 takes around 45 minutes using Colab with GPU A100.

Overall, our results demonstrate that the student RN34, despite its smaller capacity, is able to not only match but even surpass the teacher RN50 under several configurations. This improvement cannot be attributed to overfitting, since no ground truth labels are used during distillation, but instead highlights the effectiveness of TE-based distillation in transferring structured, generalizable information. This experiment thus provides strong evidence that TE is a valuable component for enhancing knowledge transfer in classification tasks.

Table 16: Zero-shot classification accuracy (%) of RN34-based VLM student and teacher CLIP RN50 on Food-101.

| $\alpha$ | $\beta$ | $\delta$ | $\eta$ | $\lambda$ | $\gamma$ | Top-1 Acc. | Top-5 Acc. |
|---|---|---|---|---|---|---|---|
| 1 | 50 | 1 | 0 | 0 | 0 | 80.23% | 95.22% |
| 1 | 50 | 1 | 0 | 0 | 2.5 | 82.46% | 96.23% |
| 1 | 50 | 1 | 0 | 0 | 5 | 82.37% | 96.44% |
| 1 | 50 | 1 | 0 | 0 | 7.5 | 82.27% | **96.62%** |
| 1 | 50 | 1 | 0 | 0 | 10 | 82.01% | 96.38% |
| 5 | 50 | 1 | 0 | 0 | 2.5 | **82.91%** | 96.47% |
| 1 | 100 | 1 | 0 | 0 | 2.5 | 82.54% | 96.10% |
| 1 | 50 | 5 | 0 | 0 | 2.5 | 81.07% | 95.30% |
| 1 | 50 | 1 | 2.5 | 0 | 0 | 82.13% | 96.10% |
| 1 | 50 | 1 | 5 | 0 | 0 | 82.37% | 96.50% |
| 1 | 50 | 1 | 7.5 | 0 | 0 | 81.39% | 96.24% |
| 1 | 50 | 1 | 5 | 50 | 7.5 | 81.63% | 96.29% |
| Teacher (RN50) | | | | | | 79.80% | 96.17% |

### 5.6.2 Out-of-Dataset Zero-Shot Classification on ImageNet-1k

To further assess how much semantic structure the student retains beyond Food-101, we evaluate zero-shot classification on the standard ImageNet-1k benchmark (Deng et al., 2009)(Russakovsky et al., 2015). We distill an RN34 student from an RN50 teacher using MSCOCO and Food-101, and then perform zero-shot classification on the ImageNet-1k validation set (50K images) following the standard CLIP-style protocol. Importantly, the student is *never* trained on ImageNet images; thus, this experiment measures out-of-domain semantic transfer under a substantial domain shift.

Table 17: Zero-shot classification accuracy (%) of distilled RN34-based VLM student on ImageNet-1k.

| $\alpha$ | $\beta$ | $\delta$ | $\eta$ | $\lambda$ | $\gamma$ | Top-1 Acc. | Top-5 Acc. |
|---|---|---|---|---|---|---|---|
| 1 | 50 | 1 | 0 | 0 | 5 | 10.23% | 24.08% |
| 1 | 50 | 1 | 0 | 0 | 7.5 | 10.62% | 24.46% |
| 1 | 50 | 1 | 2.5 | 0 | 0 | 8.09% | 19.92% |
| 1 | 50 | 1 | 5 | 0 | 0 | 8.61% | 21.24% |

Table 17 reports Top-1 and Top-5 accuracies for representative hyperparameter settings. Overall, incorporating the TE reward yields the strongest ImageNet performance among the tested configurations: using $\gamma \in \{5, 7.5\}$ achieves Top-1 accuracy of 10.23%–10.62% and Top-5 accuracy of 24.08%–24.46%, outperforming MI-only variants (e.g., $\eta \in \{2.5, 5\}$ with $\gamma = 0$), which attain Top-1 accuracy of 8.09%–8.61% and Top-5 accuracy of 19.92%–21.24%. This trend is consistent with our retrieval ablations, where moderate TE regularization improves alignment and yields better generalization. While absolute ImageNet accuracy remains well below the RN50 teacher, as expected given the student's smaller capacity and the fact that distillation is performed only on MSCOCO and Food-101, the improvement from TE indicates that TE-guided distillation helps preserve transferable semantic information under a challenging out-of-domain evaluation.

### 5.7 Additional Experiments

In Appendix H, we report five additional experimental analyses to further evaluate the proposed TE-inspired proxy regularization. Appendix H.1 compares our method with a relational geometry-matching baseline based on within-batch Gram similarity alignment. Appendix H.2 isolates the contribution of the TE-inspired proxy by fixing the base distillation objective and varying only the TE-related component. Appendix H.3 studies the finite-difference pairing strategy by comparing random adjacent-pairing with nearest-neighbor

pairing in the teacher embedding space. Appendix H.4 reports additional results for teacher RN50 and student RN34 on Flickr8k, and Appendix H.5 reports results for teacher RN50 and a smaller student RN18 on MSCOCO. Together, these experiments provide further evidence that the proposed TE-inspired proxy regularization is effective across related geometry-matching baselines, proxy-component ablations, pairing strategies, datasets, and student capacities.

## 6 Conclusions and Future Work

In this work, we introduced a *TE-inspired* regularization framework for VLM distillation. Rather than directly estimating optimization-step transfer entropy, which is intractable in high-dimensional multimodal embedding spaces, we derive and implement tractable geometric proxy objectives based on first-order Jacobian alignment and within-batch finite differences. These TE-inspired proxies encourage teacher-consistent local representation changes across image and text modalities while remaining lightweight to compute, making them practical to incorporate into standard CLIP-style teacher–student training pipelines.

Across CLIP-style teacher–student settings, we show that the proposed proxy regularizers improve retrieval performance and transfer behavior over strong baselines. In particular, the empirical results indicate that the benefits are most pronounced for image-to-text retrieval, while remaining competitive for text-to-image retrieval. Beyond standard in-dataset retrieval results, our revised experiments include recipe-level diagnostics across temperature and batch size, cross-dataset retrieval from MSCOCO to Flickr8k, and representation-level analyses of teacher–student embedding geometry. These additional studies strengthen the evidence-to-claim mapping by showing not only that the proposed proxy objective improves downstream metrics, but also that it changes the learned representation structure in a more teacher-faithful way and continues to provide gains under changes in training recipe and evaluation distribution.

Taken together, these results support a consistent picture: the implemented TE-inspired proxy objective improves teacher–student geometric alignment, preserves more transferable multimodal structure, and provides a stable and effective regularizer for VLM distillation in practice. More broadly, our findings suggest that viewing distillation through the lens of directed teacher-guided update geometry can be useful even when the ideal information-theoretic quantity itself is not directly estimable. In this sense, optimization-step TE serves as a principled conceptual motivation, while the practical contribution of the paper lies in the final tractable proxy objectives that are actually optimized.

Our work focuses on fixed-teacher distillation and on first-order local geometric surrogates. An important direction for future work is to extend this framework to co-distillation or jointly optimized teacher–student settings, where the interaction between two evolving models may yield richer forms of directed information flow. Another direction is to explore stronger geometric or information-theoretic surrogates that better capture nonlinear dependencies beyond the local first-order signal used here. It would also be valuable to evaluate whether similar TE-inspired proxy objectives remain effective for larger VLMs, stronger students, and broader multimodal tasks such as visual question answering, instruction tuning, and generative multimodal modeling.

## 7 Limitations

Despite the demonstrated effectiveness of our TE-inspired proxy regularization for VLM distillation, the approach has several limitations. First, the method does not directly estimate optimization-step transfer entropy; instead, it relies on tractable geometric surrogates based on cosine alignment of local finite-difference directions. While this makes the method computationally practical in high-dimensional multimodal embedding spaces, it also means that the surrogate may not fully capture the complex nonlinear dependencies that motivate the original TE perspective. Accordingly, the empirical results in this paper should be interpreted as validating the usefulness of the final proxy objective, rather than as a direct empirical verification of the ideal TE quantity.

Second, while TE1/TE2 consistently improve retrieval, the student can still remain far from the teacher because these surrogates capture only a *restricted* geometric signal: alignment of *local* representation vari-

ations between teacher and student on a batch. Several important aspects of the teacher geometry are not directly constrained. TE1/TE2 do not enforce *global* isometry of the embedding space, such as preservation of long-range pairwise distances, full-distribution neighborhood structure, or higher-order manifold curvature, so a student may match local directional trends while remaining globally mis-scaled or warped. They also emphasize *directional* alignment without directly controlling the *magnitude* or anisotropy spectrum of variation, such as principal directions or singular-value structure, which can still affect retrieval quality.

Third, TE1/TE2 operate only on final embeddings and therefore do not explicitly constrain *intermediate* representations, cross-modal interaction mechanisms, or capacity-dependent inductive biases that may contribute to the teacher's stronger semantic organization. In addition, because the finite-difference directions are formed from within-batch pairings, the surrogate samples only a limited subset of directions in the input space. Directions that are especially important for hard negatives, rare concepts, or fine-grained attributes may be underrepresented, leaving residual teacher–student mismatch. The current framework also assumes access to paired teacher–student representations for each sample, which may be less suitable in settings with partial supervision, noisy teacher signals, or missing modalities.

Fourth, although the proposed TE-inspired proxy is evaluated across multiple teacher backbones and datasets, the current experiments are primarily conducted in CLIP-style dual-encoder distillation settings with compact student models. This setting is appropriate for studying global image–text representation alignment, retrieval, and zero-shot transfer, but it does not directly cover multimodal reasoning benchmarks such as VQA, NLVR2, or visual commonsense reasoning. These tasks typically require additional cross-modal fusion, task-specific prediction heads, or generative reasoning architectures, whereas our method operates on pooled image and text embeddings. Extending the proposed teacher–student directional-alignment objective to fusion-based or generative VLMs is an important direction for future work. In addition, while our experiments include stronger teacher backbones, further evaluation with larger students and larger-scale VLMs would provide a more complete understanding of the scaling behavior of the proposed proxy regularizer.

These limitations suggest that TE1/TE2 are best viewed as efficient geometric regularizers that complement, rather than replace, standard distributional matching losses such as contrastive, KL, MSE, or ICL-based objectives. Closing the remaining teacher–student gap will likely require richer direction sampling, stronger global-structure constraints, improved intermediate-level alignment, or higher-capacity student models. Exploring such extensions is an important direction for future work.

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

## A   Relations between Transfer Entropy, Entropy, and Mutual Information

For two discrete-time stochastic processes $X(t)$ and $Y(t)$, the transfer entropy from $X$ to $Y$ is formally defined as (Schreiber, 2000):

$$T_{X \to Y} = \sum_t p(y_{t+1}, y_t, x_t) \log \frac{p(y_{t+1} \mid y_t, x_t)}{p(y_{t+1} \mid y_t)},\tag{23}$$

where $p(\cdot)$ represents probability distributions of the respective random variables.

Transfer entropy can also be expressed in terms of conditional entropy and mutual information (Shahsavari Baboukani et al., 2020). Specifically, the transfer entropy from $X$ to $Y$, denoted $T_{X \to Y}$, measures the reduction in uncertainty about the future state $Y_{t+1}$ given the joint past of $X$ and $Y$, compared to the uncertainty given the past of $Y$ alone. Based on (23), this difference can be expressed as:

$$
\begin{aligned}
T_{X \to Y} &= \sum_t p(y_{t+1}, y_t, x_t) \log \frac{1}{p(y_{t+1} \mid y_t)} + \\
&\quad \sum_t p(y_{t+1}, y_t, x_t) \log p(y_{t+1} \mid y_t, x_t) \\
&= \sum_t p(y_{t+1}, y_t) \log \frac{1}{p(y_{t+1} \mid y_t)} + \\
&\quad \sum_t p(y_{t+1}, y_t, x_t) \log p(y_{t+1} \mid y_t, x_t) \\
&= H(Y_{t+1} \mid Y_t) - H(Y_{t+1} \mid Y_t, X_t) \tag{24} \\
&= I(Y_{t+1}; X_t \mid Y_t) \tag{25}
\end{aligned}
$$

where $H(Y_{t+1} \mid Y_t)$ is the conditional entropy of $Y_{t+1}$ given its own history $Y_t$, $H(Y_{t+1} \mid Y_t, X_t)$ is the conditional entropy of $Y_{t+1}$ given both the history of $Y$ and the history of $X$, and $I(Y_{t+1}; X_t \mid Y_t)$ is the mutual information between $Y_{t+1}$ and the history of $X$, conditioned on the history of $Y$. In this formulation, the transfer entropy quantifies the amount by which the uncertainty about the future of $Y$ is reduced by incorporating information from $X$.

## B   Prior Work on Mutual Information and Transfer Entropy Estimation

Mutual Information (MI) techniques have been employed to capture shared information between variables (Hjelm et al., 2018)(Oord et al., 2018)(Tian et al., 2020). MINE (Belghazi et al., 2018) offers a differentiable estimator for mutual information, and information-theoretic regularization has been applied in generative models for disentanglement and improved control (Chen et al., 2016). In (Gao et al., 2015), a mutual information estimator was proposed based on modified k-nearest neighbor (KNN) that is robust to local non-uniformity with limited data. A diverse set of distributions with known MI values were introduced to evaluate the performance of different MI estimators beyond traditional normal distributions (Czyż et al., 2023). McAllester and Stratos (McAllester & Stratos, 2020) highlighted the inherent difficulties in estimating mutual information from finite data, demonstrating that any distribution-free high-confidence lower bound on MI cannot exceed $O(\ln N)$, thereby underscoring the fundamental challenges in accurate mutual information estimation without strong assumptions about the data distribution. Goldfeld and Greenewald (Goldfeld & Greenewald, 2021) introduced Sliced Mutual Information, a scalable measure that projects high-dimensional distributions onto one-dimensional subspaces, effectively capturing complex dependencies while reducing computational complexity. Approximating mutual information of high-dimensional variables using learned representations was studied in (Gowri et al., 2025).

Transfer entropy is a conditional mutual information from two stochastic processes, so it's more challenging in TE estimation. In (Zhang, 2018), Low-dimensional approximation in the searching procedure was applied to transfer entropy from non-uniform embedding. In (Zhu et al., 2015), KNN was used for TE estimation. However, KNN-based approach doesn't work well if the data are noisy and long ranged. To

overcome this weakness, a perturbation model based on locality sensitive hash function was proposed for TE estimation (Garg et al., 2022). Three estimators were used for TE estimation (Lee et al., 2012), namely fixed-binning with ranking, kernel density estimation, and the Darbellay-Vajda (D-V) adaptive partitioning algorithm extended to three dimensions. In (Ma, 2019), copula entropy was applied to TE estimation. To overcome the curse of dimensionality in TE estimation, TE was decomposed into a sum of finite-dimensional contributions in (Runge et al., 2012). Recently, transformer was used for TE estimation (Luxembourg et al., 2024). In this paper, we propose TE approximation approaches which can tremendously reduce the computation cost and overcome the curse of dimensionality.

## C  Proof of Theorem 1

This section shows how a first-order (linear) expansion of TE leads to a computable surrogate based on a cosine similarity between the teacher– and student-process Jacobians. Our derivation follows the *linear–Gaussian surrogate* technique proposed in (Goldfeld et al., 2019).

*Proof.* Let $x \in \mathbb{R}^d$ be an input image–caption pair, and $f_T(x)$, $f_S(x) \in \mathbb{R}^D$ denote the teacher and student embeddings, respectively. Denote their Jacobians as $J_T(x) = \nabla_x f_T(x)$ and $J_S(x) = \nabla_x f_S(x)$, both in $\mathbb{R}^{D \times d}$.

To study the local behavior around $x$, consider a small perturbation $\delta x \sim \mathcal{N}(0, \sigma^2 I_d)$, and define

$$u := f_T(x + \delta x), \qquad v_t := f_S(x), \qquad v_{t+1} := f_S(x + \delta x).$$

The one-step transfer entropy from teacher to student becomes:

$$T_T^S(x) = I\big(v_{t+1};\, u \,|\, v_t\big). \tag{26}$$

Using a first-order Taylor expansion around $x$:

$$u \approx u_0 + J_T \delta x, \quad v_{t+1} \approx v_0 + J_S \delta x, \quad v_t = v_0 = f_S(x), \tag{27}$$

where $u_0 = f_T(x)$. Since $v_0$ is a constant shift, subtracting it from both sides does not change the conditional mutual information. Therefore:

$$T_T^S(x) \approx I\big(J_S \delta x;\, J_T \delta x\big). \tag{28}$$

Because $\delta x \sim \mathcal{N}(0, \sigma^2 I_d)$ and both $J_S$ and $J_T$ are linear maps, the pair $(J_S \delta x, J_T \delta x)$ is jointly Gaussian. Define the covariances:

$$\Sigma_S = \sigma^2 J_S J_S^\top, \quad \Sigma_T = \sigma^2 J_T J_T^\top, \quad \Sigma_{ST} = \sigma^2 J_S J_T^\top.$$

The mutual information between jointly Gaussian vectors is (Cover, 1999):

$$
\begin{aligned}
I\big(J_S \delta x;\, J_T \delta x\big) &= h\big(J_S \delta x\big) + h\big(J_T \delta x\big) - h\big(J_S \delta x,\, J_T \delta x\big) &(29) \\
&= \frac{1}{2} \log \frac{\det \Sigma_S \, \det \Sigma_T}{\det \begin{pmatrix} \Sigma_S & \Sigma_{ST} \\ \Sigma_{TS} & \Sigma_T \end{pmatrix}} &(30) \\
&= -\frac{1}{2} \log \det \left( I - \Sigma_S^{-1/2} \Sigma_{ST} \Sigma_T^{-1/2} \right). &(31)
\end{aligned}
$$

If $\Sigma_{ST}$ is small compared to the product $\Sigma_S^{1/2} \Sigma_T^{1/2}$ (which is often true in early training), we can use the approximation $\log \det(I - A) \approx -\operatorname{tr}(A)$ (Magnus & Neudecker, 1999). This gives (Goldfeld et al., 2019):

$$T_T^S(x) \approx \frac{\sigma^2}{2} \operatorname{tr}\left( (J_S J_S^\top)^{-1/2} J_S J_T^\top (J_T J_T^\top)^{-1/2} \right). \tag{32}$$

We can normalize both Jacobians by their Frobenius norms:

$$\widetilde{J}_S = \frac{J_S}{\|J_S\|_F}, \qquad \widetilde{J}_T = \frac{J_T}{\|J_T\|_F},$$

so that equation equation 32 becomes:

$$T_T^S(x) \propto \langle \widetilde{J}_S, \ \widetilde{J}_T \rangle_F = \cos\left(\widetilde{J}_S, \ \widetilde{J}_T\right), \tag{33}$$

i.e., the Frobenius inner product (cosine similarity) of the two Jacobians.

$\square$

## D    Performance Comparison: TE-Inspired Proxies versus Exact Gaussian TE

To provide a controlled sanity check for the two TE-inspired proxy measures introduced in Sections 4.2.1 and 4.2.2, we compare them against an analytic Gaussian TE reference in a synthetic teacher–student setting. This experiment is not intended as a direct validation of the optimization-step TE quantity in Eq. (3). Rather, it examines whether the proposed cosine-based proxy measures behave consistently with a simple information-theoretic reference under a linear–Gaussian model where the teacher–student dependence is explicitly controlled.

Specifically, teacher embeddings $\mathbf{T} \in \mathbb{R}^D$ are sampled from a standard normal distribution,

$$\mathbf{T} \sim \mathcal{N}(0, I),$$

and student embeddings are generated as

$$\mathbf{S} = \alpha\,\mathbf{T} + \sqrt{1 - \alpha^2}\,\mathbf{N}, \tag{34}$$

where $\mathbf{N} \sim \mathcal{N}(0, I)$ and $\alpha \in [0, 0.99]$ controls the teacher–student correlation. Each corresponding pair of teacher and student components therefore forms a jointly Gaussian random pair with Pearson correlation coefficient $\alpha$ (Lee Rodgers & Nicewander, 1988). It is a classical result in information theory that for two jointly Gaussian random variables $X$ and $Y$ with correlation $\alpha$, the mutual information is (Cover, 1999)

$$I(X;Y) = -\frac{1}{2}\log\left(1 - \alpha^2\right). \tag{35}$$

In this synthetic Gaussian setting, we define an analytic TE reference by

$$\mathrm{TE}_{\mathrm{gauss}} = I(Y_{t+1}; X_t \mid Y_t), \tag{36}$$

where $Y_{t+1}$ represents the student's updated representation, $X_t$ is the teacher's representation at time $t$, and $Y_t$ is the student's current representation. Under the assumption that these variables are jointly Gaussian and that the update of $Y_{t+1}$ depends linearly on $X_t$ after conditioning on $Y_t$, the conditional mutual information admits a closed-form expression. In particular, if the effective conditional correlation between $X_t$ and $Y_{t+1}$ is given by $\alpha$, then the mutual information per embedding dimension becomes

$$I(Y_{t+1}; X_t \mid Y_t) = -\frac{1}{2}\log\left(1 - \alpha^2\right). \tag{37}$$

When the embeddings have $D$ independent dimensions, this yields

$$\mathrm{TE}_{\mathrm{gauss}} = \frac{D}{2}\,\log\left(\frac{1}{1 - \alpha^2}\right). \tag{38}$$

For ease of comparison with our cosine-based proxy measures, we further normalize this analytic Gaussian TE reference via a logarithmic transformation to map it into the interval $[0, 1]$:

$$\mathrm{TE}_{\mathrm{norm}} = \frac{\log(1 + \mathrm{TE}_{\mathrm{gauss}})}{\log(1 + \mathrm{TE}_{\mathrm{max}})}, \tag{39}$$

where $\text{TE}_{\max}$ is computed using $\alpha_{\max} = 0.99$ to define the upper bound for normalization. This normalization allows the unbounded Gaussian reference quantity to be compared more directly with the bounded cosine-based proxy scores.

For the two approximation methods proposed in Sections 4.2.1 (Method 1) and 4.2.2 (Method 2), we vary $\alpha$ from 0 to 0.99 and compute both proxy measures together with the normalized analytic Gaussian TE reference. The results are summarized in Fig. 9. The Pearson correlation between the normalized analytic reference and both proxy measures is 0.994, indicating a very strong linear relationship in this synthetic setting. These findings suggest that both proxy measures reliably track the relative dependence trend induced by $\alpha$, even though they are not direct estimators of the analytic reference quantity.

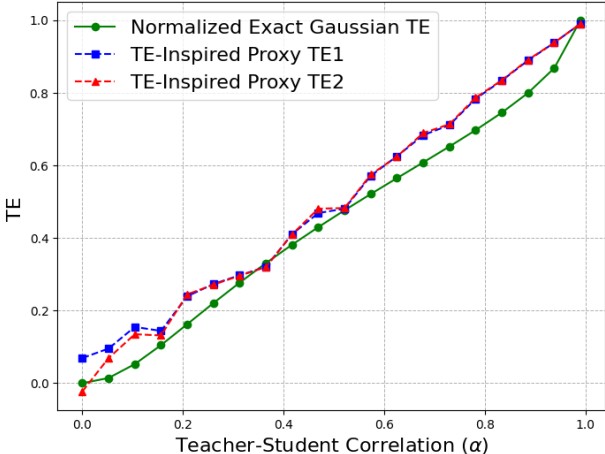

Figure 9: Comparison of TE-inspired proxy measures and the normalized analytic Gaussian TE reference.

We also examine the robustness of the two proxy methods as we vary two key factors in the same synthetic teacher–student setting:

- batch size $B$, which affects the stability of sample-based estimates; and

- embedding dimension $D$, which influences the scale of the analytic Gaussian reference.

We fix the teacher–student correlation coefficient at $\alpha = 0.8$ in Eq. (34) and perform two experiments:

1. **Varying batch size:** we fix $D = 500$ and consider batch sizes $B \in \{10, 20, 50, 100, 200, 500, 1000\}$.

2. **Varying embedding dimension:** we fix $B = 500$ and let $D \in \{10, 20, 50, 100, 200, 500, 1000\}$.

In both cases, we compute Method 1, Method 2, and the normalized analytic Gaussian TE reference.

Figure 10a shows the behavior of these quantities as a function of batch size. Both proxy measures rapidly converge to stable estimates near the normalized analytic reference. For very small $B$ (around 10–20), the sample-based cosine measures show slight deviations, but they remain close to the reference trend. As $B$ grows, the variance diminishes and both proxy measures stabilize.

Figure 10b illustrates the impact of varying embedding dimension $D$. Since the analytic Gaussian TE reference increases with $D$ due to the accumulation of information across additional dimensions, its normalized value also changes with dimension. By contrast, the two proxy measures remain relatively stable, hovering around 0.75–0.80 across the tested dimensions. This highlights a key property of the proxy measures: they capture relative geometric alignment between teacher and student, controlled here by $\alpha$, but they do not scale with embedding dimensionality in the same way as the analytic mutual-information quantity. In practice, this makes them computationally efficient and robust in high-dimensional settings, although they are not intended to quantify the absolute amount of transferred information.

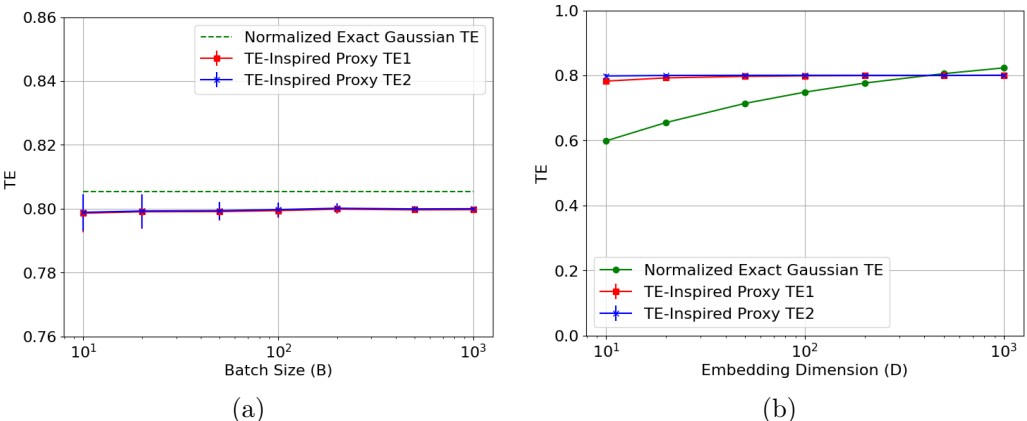

Figure 10: (a) TE-inspired proxy measures versus batch size ($B$) at fixed $D = 500$. (b) TE-inspired proxy measures versus embedding dimension ($D$) at fixed $B = 500$.

The underlying intuition behind these proxy measures is that if the student's local directional changes align closely with those of the teacher, then the student is preserving teacher-consistent geometric structure. Traditional transfer entropy relies on probabilistic modeling across random variables and conditional dependencies, which becomes difficult to estimate in high-dimensional multimodal embedding spaces. Our proxy construction instead replaces this with directional alignment in embedding space. Accordingly, the proposed measures should be interpreted as efficient geometric surrogates rather than exact information-theoretic estimators. In addition, equal weighting of image and text modalities in TE1 may not always be optimal, since the two modalities may contribute differently depending on the task and model.

Overall, the cosine-based TE-inspired proxies are effective in capturing relative teacher–student dependence trends in this synthetic Gaussian setting: they are easy to compute, stable across batch sizes, robust in high dimensions, and highly correlated with the normalized analytic Gaussian TE reference. At the same time, they do not measure exact information volume. Instead, they compress the notion of teacher–student dependence into bounded similarity-based scores. As a result, they are best viewed as scalable proxy signals for relative comparison and regularization, rather than as absolute information-theoretic quantities.

# E    Computational Cost Analysis: Exact TE versus TE-Inspired Proxies

**Computational Complexity:**

Direct estimation of transfer entropy in high-dimensional multimodal representation spaces is often computationally expensive. In general, estimating conditional mutual information or transfer entropy requires modeling dependencies among multiple variables, which may involve density estimation, kernel methods, or nearest-neighbor statistics whose cost scales poorly as the number of samples and feature dimensionality increase. This makes direct TE-style computation impractical for large-scale VLM distillation, where both the batch size and embedding dimension can be substantial.

In contrast, the TE-inspired proxy measures used in this paper are much simpler computationally. TE Proxy Method 1 in Section 4.2.1 uses cosine similarity between within-batch finite-difference directions in the teacher and student embedding spaces. Concretely, after computing teacher and student embeddings for a batch, the method forms adjacent-pair directional differences and compares them by cosine similarity. This avoids constructing any joint probability model or density estimate. For a batch of size $B$ and embedding dimension $d$, the additional computation scales approximately as $O(Bd)$, since it consists mainly of vector differences and cosine-similarity calculations on already-computed embeddings.

TE Proxy Method 2 in Section 4.2.2 similarly uses cosine similarity after concatenating image and text directional differences into a joint multimodal representation. This also avoids explicit TE estimation and

remains linear in the effective feature dimension of the concatenated vectors. In practice, both methods add only lightweight operations on top of the standard forward pass of teacher and student encoders, making their cost negligible relative to the encoder computation itself.

**Memory Usage:** Exact TE-style estimation can require storing large collections of joint samples, pairwise distances, or density-estimation structures, and these costs can grow rapidly with dimensionality. This makes direct computation infeasible for high-dimensional embeddings, since it may demand substantial storage for probability tables, kernel statistics, or large neighborhood structures. The TE-inspired proxies mitigate this issue by avoiding explicit density estimation altogether. TE Proxy Method 1 stores only the teacher and student embeddings for the current batch, together with the corresponding within-batch difference vectors, keeping memory usage approximately at $O(Bd)$. TE Proxy Method 2 operates on the same batch features and their concatenated directional differences, leading to a comparable memory requirement. These proxy measures therefore enable scalable computation in large-scale deep learning settings without overwhelming memory constraints.

**Scalability in High Dimensions:** Direct TE estimation suffers from the curse of dimensionality. As dimensionality increases, conditional density estimation becomes increasingly unreliable because high-dimensional data points become sparse, making the estimated information-theoretic quantities unstable and sample-inefficient. In contrast, cosine similarity-based proxy measures are far more scalable, since cosine similarity remains well-defined and easy to compute even in high dimensions. TE Proxy Method 1 relies only on directional comparisons between teacher and student embeddings within a batch, while TE Proxy Method 2 extends this idea to the joint multimodal direction space through concatenation. Because neither method requires explicit probabilistic modeling in the full feature space, both remain practical even for very large embeddings.

In summary, using TE-inspired cosine-similarity proxies enables analysis and regularization of high-dimensional, large-scale VLM distillation settings that would be computationally prohibitive for direct TE estimation. These proxy measures significantly improve computational feasibility and remain stable in settings where exact information-theoretic estimation would be difficult or unreliable. The trade-off, however, is that they provide only a geometric surrogate rather than an exact measure of information flow, and may therefore overlook more complex nonlinear dependencies in the data. Since the primary goal in CLIP-style distillation is to handle rich multimodal embeddings efficiently while obtaining a practical teacher-consistent alignment signal, these TE-inspired proxy measures provide a useful and scalable alternative to direct TE computation.

## F  Loss Functions in VLM Distillation

### F.1  Logit Representation in VLM Distillation

In our framework, logits represent the similarity scores between image and text embeddings, which are fundamental to contrastive learning. Given a batch of image-text pairs, let $\mathbf{v}^{(S)}, \mathbf{u}^{(S)}$ denote the image and text embeddings from the student model, and $\mathbf{v}^{(T)}, \mathbf{u}^{(T)}$ denote the corresponding embeddings from the teacher model. The logit computation follows these steps.

First, we normalize the embeddings to unit norm:

$$\hat{\mathbf{v}}^{(S)} = \frac{\mathbf{v}^{(S)}}{\|\mathbf{v}^{(S)}\|_2}, \quad \hat{\mathbf{u}}^{(S)} = \frac{\mathbf{u}^{(S)}}{\|\mathbf{u}^{(S)}\|_2}, \tag{40}$$

$$\hat{\mathbf{v}}^{(T)} = \frac{\mathbf{v}^{(T)}}{\|\mathbf{v}^{(T)}\|_2}, \quad \hat{\mathbf{u}}^{(T)} = \frac{\mathbf{u}^{(T)}}{\|\mathbf{u}^{(T)}\|_2}. \tag{41}$$

The similarity logits for the student and teacher models are then computed as the dot product between the corresponding image and text embeddings, scaled by a temperature parameter $\tau$:

$$\mathbf{z}^{(S)} = \frac{\hat{\mathbf{v}}^{(S)} \cdot (\hat{\mathbf{u}}^{(S)})^\top}{\tau}, \quad \mathbf{z}^{(T)} = \frac{\hat{\mathbf{v}}^{(T)} \cdot (\hat{\mathbf{u}}^{(T)})^\top}{\tau}. \tag{42}$$

Here, $\mathbf{z}^{(S)}$ and $\mathbf{z}^{(T)}$ are $|B| \times |B|$ matrices, where each entry $z_{ij}^{(S)}$ represents the similarity between the $i$-th image embedding and the $j$-th text embedding in the batch for the student model, and similarly for the teacher model. The temperature parameter $\tau$ controls the sharpness of the similarity distribution, with lower values making the distribution more peaky.

These logits are subsequently used in the contrastive loss and KL divergence computation to align the student's feature representations with those of the teacher, ensuring effective knowledge transfer during distillation. Several studies have explored the computation and utilization of these logits in image-text contrastive frameworks (Radford et al., 2021)(Jia et al., 2021)(Yang et al., 2022)(Hasegawa et al., 2023)(Xiao et al., 2024).

## F.2 Contrastive Loss for VLM Distillation

We employ a contrastive loss based on the InfoNCE loss formulation to align the student model's image and text representations effectively. Given a batch of $|B|$ image-text pairs, we define the contrastive loss using the computed logits. The contrastive loss for image-to-text alignment is defined as (Oord et al., 2018):

$$\mathcal{L}_{I \to T} = -\frac{1}{|B|} \sum_{k=1}^{|B|} \log \frac{\exp(z_{kk}^{(S)})}{\sum_{j=1}^{|B|} \exp(z_{kj}^{(S)})} \tag{43}$$

$z_{kj}^{(S)}$ represents the similarity between the $k$-th image embedding and the $j$-th text embedding in the batch for the student model, and $z_{kk}^{(S)}$ represents the similarity logit between the $k$-th image and its corresponding text in the batch for the student model.

Similarly, the contrastive loss for text-to-image alignment is given by:

$$\mathcal{L}_{T \to I} = -\frac{1}{|B|} \sum_{k=1}^{|B|} \log \frac{\exp(z_{kk}^{(S)})}{\sum_{j=1}^{|B|} \exp(z_{jk}^{(S)})} \tag{44}$$

$z_{jk}^{(S)}$ represents the similarity between the $j$-th image embedding and the $k$-th text embedding in the batch for the student model, and $z_{kk}^{(S)}$ is the same as that in (43).

The total contrastive loss, which balances both image-to-text and text-to-image objectives, is computed as:

$$\mathcal{L}_{\text{contrastive}} = \frac{1}{2}(\mathcal{L}_{I \to T} + \mathcal{L}_{T \to I}). \tag{45}$$

This loss function encourages the student model to align its multi-modal representations by bringing matching pairs closer in the embedding space while pushing apart non-matching pairs. Contrastive loss has been extensively applied to knowledge distillation (Chen et al., 2021)(Gao et al., 2021)(Peng et al., 2022)(Zhu et al., 2021)(Guo et al., 2023).

To enhance the effectiveness of distillation, we extend this contrastive loss with additional terms such as KL divergence and transfer entropy-based regularization. These terms further refine the student model's learning dynamics by ensuring information flow from the teacher's embeddings to the student's representations while preserving structural consistency across modalities.

## F.3 KL Divergence for VLM Distillation

To ensure that the student model effectively mimics the probability distributions of the teacher model, we include a Kullback-Leibler (KL) divergence loss term. KL divergence measures how much the student's predicted distribution deviates from the teacher's distribution, enforcing a closer alignment between their logits. KL divergence has been applied to VLM distillation (Li et al., 2024b)(Sun et al., 2024).

For a given batch of image-text pairs, let $\mathbf{z}^{(S)}$ and $\mathbf{z}^{(T)}$ represent the similarity logits of the student and teacher models, respectively. The soft probability distributions are obtained via the softmax function:

$$P_i^{(S)} = \frac{\exp(z_i^{(S)}/\tau)}{\sum_{j=1}^{|B|} \exp(z_j^{(S)}/\tau)}, \tag{46}$$

$$P_i^{(T)} = \frac{\exp(z_i^{(T)}/\tau)}{\sum_{j=1}^{|B|} \exp(z_j^{(T)}/\tau)}, \tag{47}$$

where $\tau$ is the temperature parameter that controls the sharpness of the distributions.

The KL divergence loss is computed as:

$$\mathcal{L}_{\mathrm{KL}} = \frac{1}{2} \left( D_{\mathrm{KL}} \left( P_{\mathrm{image}}^{(S)} \parallel P_{\mathrm{image}}^{(T)} \right) + D_{\mathrm{KL}} \left( P_{\mathrm{text}}^{(S)} \parallel P_{\mathrm{text}}^{(T)} \right) \right), \tag{48}$$

where the KL divergence between two probability distributions $P^{(S)}$ and $P^{(T)}$ is defined as:

$$D_{\mathrm{KL}}(P^{(S)} \parallel P^{(T)}) = \sum_{i=1}^{|B|} P_i^{(T)} \log \frac{P_i^{(T)}}{P_i^{(S)}}. \tag{49}$$

This loss encourages the student model to produce probability distributions that closely resemble those of the teacher, effectively preserving the knowledge distilled from the teacher while allowing the student to generalize efficiently.

### F.4 MSE Loss Function for VLM Distillation

To further align the feature representations of the teacher and student models, we include MSE loss that minimizes the discrepancy between their intermediate embeddings (Yang et al., 2024). The MSE loss is computed as the sum of the squared differences between the student and teacher embeddings for both modalities:

$$\mathcal{L}_{\mathrm{MSE}} = \mathcal{L}_{\mathrm{MSE}}^{\mathrm{image}} + \mathcal{L}_{\mathrm{MSE}}^{\mathrm{text}}, \tag{50}$$

where

$$\mathcal{L}_{\mathrm{MSE}}^{\mathrm{image}} = \frac{1}{|B|} \sum_{i=1}^{|B|} \left\| \hat{\mathbf{v}}_i^{(S)} - \hat{\mathbf{v}}_i^{(T)} \right\|^2, \tag{51}$$

$$\mathcal{L}_{\mathrm{MSE}}^{\mathrm{text}} = \frac{1}{|B|} \sum_{i=1}^{|B|} \left\| \hat{\mathbf{u}}_i^{(S)} - \hat{\mathbf{u}}_i^{(T)} \right\|^2. \tag{52}$$

Here, $|B|$ represents the batch size, and $\|\cdot\|^2$ denotes the squared Euclidean norm. This loss ensures that the student model's learned embeddings remain close to the teacher's representations in the feature space, facilitating effective knowledge transfer. MSE has been applied to VLM loss function in (Yang et al., 2024), and was called feature distillation.

### F.5 Interactive Contrastive Learning

Interactive Contrastive Learning (ICL) was proposed in (Yang et al., 2024) to aligns the student model's feature representations with those of the teacher by treating the student embeddings as anchors and contrasting them with the teacher embeddings.

Given a batch of image-text pairs, let $\mathbf{v}_k^{(S)}$ be the image embedding from the student model, and $\{\mathbf{u}_b^{(T)}\}_{b=1}^{|B|}$ denote the contrastive text embeddings from the teacher model. The image-to-text ICL loss is formulated as:

$$\mathcal{L}_{\text{ICL}}^{I \to T} = -\log \frac{\exp(\mathbf{v}_k^{(S)} \cdot \mathbf{u}_k^{(T)}/\tau)}{\sum_{b=1}^{|B|} \exp(\mathbf{v}_k^{(S)} \cdot \mathbf{u}_b^{(T)}/\tau)}, \tag{53}$$

where $\tau$ is the temperature parameter.

Similarly, for a student text embedding $\mathbf{u}_k^{(S)}$ and contrastive image embeddings from the teacher model $\{\mathbf{v}_b^{(T)}\}_{b=1}^{|B|}$, the text-to-image ICL loss is:

$$\mathcal{L}_{\text{ICL}}^{T \to I} = -\log \frac{\exp(\mathbf{u}_k^{(S)} \cdot \mathbf{v}_k^{(T)}/\tau)}{\sum_{b=1}^{|B|} \exp(\mathbf{u}_k^{(S)} \cdot \mathbf{v}_b^{(T)}/\tau)}. \tag{54}$$

The final ICL loss is a combination of the two:

$$\mathcal{L}_{\text{ICL}} = \frac{1}{2} \left( \mathcal{L}_{\text{ICL}}^{I \to T} + \mathcal{L}_{\text{ICL}}^{T \to I} \right). \tag{55}$$

By integrating ICL, the student model effectively learns from the teacher's structured feature space, leading to improved representation learning and knowledge transfer.

### F.6 Mutual-Information (MI) Objective

In addition to the losses above, we also evaluate mutual-information objectives as an alternative teacher-student supervision signal. This baseline helps distinguish improvements due to directed, conditional information-flow regularization from improvements that may arise from standard contrastive representation matching.

**MI baseline loss (CRD-style).** We implement an MI-driven alignment loss following the contrastive mutual-information maximization principle used in Contrastive Representation Distillation (CRD) (Tian et al., 2020). Using the same notation as above, given a batch of $|B|$ image-text pairs, let $\mathbf{v}_k^{(S)}$ and $\mathbf{u}_k^{(S)}$ denote the student image and text embeddings, and $\mathbf{v}_k^{(T)}$ and $\mathbf{u}_k^{(T)}$ the corresponding teacher embeddings. We $\ell_2$-normalize the embeddings, yielding $\hat{\mathbf{v}}_k^{(\cdot)}$ and $\hat{\mathbf{u}}_k^{(\cdot)}$, and use temperature $\tau$.

Unlike the CLIP-style contrastive loss in Eq. (43) and ICL (Yang et al., 2024), which both involve *cross-modal* matching (image↔text) either within the student or across teacher-student, this MI baseline performs *modality-matched* teacher-student alignment (image↔image and text↔text). Concretely, the image-side MI loss treats $(\hat{\mathbf{v}}_k^{(T)}, \hat{\mathbf{v}}_k^{(S)})$ as the positive pair and $(\hat{\mathbf{v}}_k^{(T)}, \hat{\mathbf{v}}_b^{(S)})$ for $b \neq k$ as negatives:

$$\mathcal{L}_{\text{MI}}^V = -\frac{1}{|B|} \sum_{k=1}^{|B|} \log \frac{\exp\left(\hat{\mathbf{v}}_k^{(T)} \cdot \hat{\mathbf{v}}_k^{(S)}/\tau\right)}{\sum_{b=1}^{|B|} \exp\left(\hat{\mathbf{v}}_k^{(T)} \cdot \hat{\mathbf{v}}_b^{(S)}/\tau\right)}. \tag{56}$$

Similarly, the text-side MI loss treats $(\hat{\mathbf{u}}_k^{(T)}, \hat{\mathbf{u}}_k^{(S)})$ as the positive pair:

$$\mathcal{L}_{\text{MI}}^U = -\frac{1}{|B|} \sum_{k=1}^{|B|} \log \frac{\exp\left(\hat{\mathbf{u}}_k^{(T)} \cdot \hat{\mathbf{u}}_k^{(S)}/\tau\right)}{\sum_{b=1}^{|B|} \exp\left(\hat{\mathbf{u}}_k^{(T)} \cdot \hat{\mathbf{u}}_b^{(S)}/\tau\right)}. \tag{57}$$

The final MI objective is the symmetric combination

$$\mathcal{L}_{\text{MI}} = \tfrac{1}{2} \left( \mathcal{L}_{\text{MI}}^V + \mathcal{L}_{\text{MI}}^U \right). \tag{58}$$

Minimizing $\mathcal{L}_{\text{MI}}$ encourages matched teacher-student representations to score higher than mismatched pairs within the batch, which corresponds to maximizing an InfoNCE lower bound on mutual information (Tian et al., 2020).

### F.7 MSE-Difference Objective

We also evaluate simple $\ell_2$ objectives in place of cosine similarity to test whether improvements can be explained by standard Euclidean change matching rather than directional similarity.

**MSE-difference baseline loss.** To mirror our change-based signals, we compare temporal differences in embeddings (across steps, layers, or time indices depending on the method):

$$\Delta\mathbf{v}_k^{(T)} = \mathbf{v}_k^{(T),(l)} - \mathbf{v}_k^{(T),(l-1)}, \qquad \Delta\mathbf{v}_k^{(S)} = \mathbf{v}_k^{(S),(l)} - \mathbf{v}_k^{(S),(l-1)}, \tag{59}$$

(and analogously for text embeddings $\Delta\mathbf{u}_k^{(T)}, \Delta\mathbf{u}_k^{(S)}$), and define

$$\mathcal{L}_{\text{MSE-}\Delta} = \frac{1}{2}\Big(\mathbb{E}_k\Big[\|\Delta\mathbf{v}_k^{(T)} - \Delta\mathbf{v}_k^{(S)}\|_2^2\Big] + \mathbb{E}_k\Big[\|\Delta\mathbf{u}_k^{(T)} - \Delta\mathbf{u}_k^{(S)}\|_2^2\Big]\Big). \tag{60}$$

This baseline isolates whether gains arise from a directed information-flow formulation versus simply matching first-order changes under an $\ell_2$ metric.

**Relation to normalized MSE-$\Delta$.** Raw MSE-$\Delta$ penalizes both directional and magnitude mismatch between teacher and student finite-difference vectors, and can therefore be dominated by differences in representation scale. A natural control is normalized MSE-$\Delta$, where each finite-difference vector is first $\ell_2$-normalized. For nonzero vectors $a$ and $b$,

$$\left\|\frac{a}{\|a\|_2 + \epsilon} - \frac{b}{\|b\|_2 + \epsilon}\right\|_2^2 \approx 2 - 2\cos(a, b),$$

with equality when $\epsilon = 0$. Hence normalized MSE-$\Delta$ is equivalent, up to an additive constant and scalar factor, to maximizing cosine similarity. Applying this identity to the image and text finite differences gives

$$L_{\text{NMSE-}\Delta}^{\text{TE1}} = 2 - 2\text{TE1},$$

and applying it to the concatenated multimodal finite differences gives

$$L_{\text{NMSE-}\Delta}^{\text{TE2}} = 2 - 2\text{TE2}.$$

Therefore, a separate normalized MSE-$\Delta$ experiment would not define a distinct training objective from the proposed cosine-based TE proxy; it would only reparameterize the same objective with a different loss weight. This clarification separates the limitation of raw MSE-$\Delta$ from the normalized directional alignment signal used by TE1/TE2.

## G Brief Introduction to the Teacher CLIP Models

We briefly summarize the CLIP architectures used in our experiments.

**CLIP ResNet-50 (RN50).** CLIP RN50 adopts a ResNet-50 backbone as the visual encoder and a Transformer-based text encoder (Radford et al., 2021). The image encoder processes $224 \times 224$ RGB inputs and produces a 1024-dimensional global image embedding, while the text encoder is a 12-layer Transformer (Vaswani et al., 2017) with hidden dimension 512. In the original CLIP implementation, the image encoder has roughly 38M parameters and the text encoder about 63M parameters, for a total of approximately 102M parameters (Radford et al., 2021). In our experiments, CLIP RN50 serves as a moderately sized vision–language teacher model that is well suited for distillation into smaller students.

**CLIP ResNet-50×16 (RN50×16).** RN50×16 is a higher-capacity variant of CLIP's ResNet family (Radford et al., 2021). It scales up RN50 using a ResNeXt-style design with $16\times$ wider convolutional blocks, substantially increasing the number of visual parameters and intermediate feature capacity. The text encoder architecture and the contrastive training objective remain unchanged, but the larger visual backbone yields stronger zero-shot performance and a richer multimodal embedding space compared to RN50.

**CLIP ViT-B/16.** CLIP ViT-B/16 replaces the convolutional backbone with a Vision Transformer (ViT-B/16) (Dosovitskiy et al., 2020; Radford et al., 2021). Images are divided into non-overlapping $16 \times 16$ patches, which are linearly projected and fed into a 12-layer Transformer encoder with hidden dimension 768, producing a global CLS-style image embedding. The text side uses the same 12-layer Transformer encoder with hidden dimension 512 as in RN50. The resulting dual-encoder model has approximately 151M parameters (about 86M in the visual encoder and 63M in the text encoder), and typically achieves stronger zero-shot performance than RN50 due to improved global reasoning and larger effective receptive fields.

**Shared CLIP training paradigm.** Across all three architectures, CLIP jointly trains the vision and text encoders with a contrastive image–text objective so that paired image–text embeddings are close in cosine-similarity space while unpaired pairs are pushed apart (Radford et al., 2021). This unified training scheme enables zero-shot recognition by comparing image embeddings to text-encoded label prompts at inference time, and provides a natural teacher for distilling smaller vision–language students.

## H Additional Experimental Results

### H.1 Comparison with Relational Geometry Matching

Relational structure is an important aspect of knowledge distillation. In addition to matching individual teacher and student embeddings, a student model can also be encouraged to preserve the teacher's pairwise representation geometry within a mini-batch. This appendix studies a relational Gram-matching baseline that aligns the within-batch cosine similarity matrices of the teacher and student. This baseline represents geometry-matching and relational-distillation objectives because it transfers sample-sample relationships rather than only pointwise feature targets.

**Relational Gram-Matching Objective.** Let $Z_S \in \mathbb{R}^{B \times d}$ and $Z_T \in \mathbb{R}^{B \times d}$ denote the student and teacher features for a mini-batch of size $B$, after row-wise $\ell_2$ normalization. The student Gram similarity matrix is defined as

$$G_S = Z_S Z_S^\top. \tag{61}$$

The teacher Gram similarity matrix is defined as

$$G_T = Z_T Z_T^\top. \tag{62}$$

For normalized features, the entries of $G_S$ and $G_T$ correspond to within-batch cosine similarities. The diagonal entries are close to one and therefore do not provide meaningful relational information. We exclude the diagonal and define the Gram-matching loss as

$$\mathcal{L}_{\mathrm{Gram}}(Z_S, Z_T) = \frac{1}{B(B-1)} \sum_{i=1}^{B} \sum_{\substack{j=1 \\ j \neq i}}^{B} (G_{S,ij} - G_{T,ij})^2. \tag{63}$$

This objective is closely related to similarity-preserving knowledge distillation (Tung & Mori, 2019) and relational knowledge distillation (Park et al., 2019), which transfer pairwise or higher-order relations among examples rather than only pointwise feature targets. More broadly, Gram-style feature relationships have also been used in knowledge distillation through flow-of-solution-procedure matrices (Yim et al., 2017).

In the vision-language setting, we apply this objective to image embeddings, text embeddings, and joint image-text embeddings. Let $v_S$ and $u_S$ denote the student image and text embeddings, respectively. Let $v_T$ and $u_T$ denote the corresponding teacher image and text embeddings. The image-level relational loss is

$$\mathcal{L}_{\mathrm{Gram}}^{\mathrm{img}} = \mathcal{L}_{\mathrm{Gram}}(v_S, v_T). \tag{64}$$

The text-level relational loss is

$$\mathcal{L}_{\mathrm{Gram}}^{\mathrm{txt}} = \mathcal{L}_{\mathrm{Gram}}(u_S, u_T). \tag{65}$$

For joint multimodal geometry matching, we concatenate the image and text embeddings for each sample and then normalize the resulting joint representation. For the $i$-th student sample, the unnormalized joint representation is

$$\widetilde{z}_{S,i}^{\text{joint}} = \text{concat}(v_{S,i}, u_{S,i}). \tag{66}$$

The normalized student joint representation is

$$z_{S,i}^{\text{joint}} = \frac{\widetilde{z}_{S,i}^{\text{joint}}}{\left\|\widetilde{z}_{S,i}^{\text{joint}}\right\|_2 + \epsilon}. \tag{67}$$

Similarly, for the teacher, the unnormalized joint representation is

$$\widetilde{z}_{T,i}^{\text{joint}} = \text{concat}(v_{T,i}, u_{T,i}). \tag{68}$$

The normalized teacher joint representation is

$$z_{T,i}^{\text{joint}} = \frac{\widetilde{z}_{T,i}^{\text{joint}}}{\left\|\widetilde{z}_{T,i}^{\text{joint}}\right\|_2 + \epsilon}. \tag{69}$$

Let $Z_S^{\text{joint}}$ and $Z_T^{\text{joint}}$ denote the matrices obtained by stacking $z_{S,i}^{\text{joint}}$ and $z_{T,i}^{\text{joint}}$ over the mini-batch. The joint Gram loss is

$$\mathcal{L}_{\text{Gram}}^{\text{joint}} = \mathcal{L}_{\text{Gram}}\left(Z_S^{\text{joint}}, Z_T^{\text{joint}}\right). \tag{70}$$

The final relational geometry-matching loss is

$$\mathcal{L}_{\text{Gram}}^{\text{total}} = \frac{1}{3}\left(\mathcal{L}_{\text{Gram}}^{\text{img}} + \mathcal{L}_{\text{Gram}}^{\text{txt}} + \mathcal{L}_{\text{Gram}}^{\text{joint}}\right). \tag{71}$$

Let the base distillation objective be

$$\mathcal{L}_{\text{base}} = \mathcal{L}_{\text{contrastive}} + \alpha\mathcal{L}_{\text{KL}} + \beta\mathcal{L}_{\text{L2}} + \delta\mathcal{L}_{\text{ICL}}. \tag{72}$$

The full objective for the Gram-matching baseline is

$$\mathcal{L}_{\text{total}} = \mathcal{L}_{\text{base}} + \mu\mathcal{L}_{\text{Gram}}^{\text{total}}. \tag{73}$$

Here, $\mu$ controls the strength of relational geometry matching.

**Relation to TE-Inspired Directional Geometry.** The Gram-matching objective transfers static pairwise similarity structure from the teacher to the student. For normalized embeddings, matching the Gram matrix corresponds to matching the within-batch angular geometry among samples. Thus, this baseline captures an important class of relational and representation-geometry distillation methods.

The proposed TE-inspired objectives provide a different geometric signal. Rather than directly matching static pairwise similarities, $\text{TE}_1$ and $\text{TE}_2$ align finite-difference directions in the teacher and student representation spaces. $\text{TE}_1$ measures per-modality directional alignment, while $\text{TE}_2$ measures joint multimodal directional alignment after concatenating image and text finite differences. The full TE objective is

$$\text{TE} = \text{TE}_1 + \text{TE}_2. \tag{74}$$

The corresponding training objective is

$$\mathcal{L}_{\text{total}} = \mathcal{L}_{\text{base}} - \gamma\left(\text{TE}_1 + \text{TE}_2\right). \tag{75}$$

Thus, Gram matching asks whether the student preserves the teacher's within-batch similarity matrix, whereas the TE-inspired objective asks whether the student's local representation changes are aligned with the teacher's local representation changes. These two forms of supervision are related but not equivalent. The finite-difference directional alignment in $\text{TE}_1$ and $\text{TE}_2$ can also be interpreted as an efficient proxy for Jacobian-action alignment, since finite differences approximate local representation changes induced by input perturbations.

| | Image-to-Text Retrieval | | | Text-to-Image Retrieval | | |
|---|---|---|---|---|---|---|
| Method | R@1 | R@5 | R@10 | R@1 | R@5 | R@10 |
| Base: CL+KL+L2+ICL | 6.08 | 18.14 | 26.93 | 5.92 | 17.06 | 24.89 |
| + Gram, $\mu = 1$ | 6.26 | 18.17 | 26.72 | 5.77 | 17.13 | 25.07 |
| + Gram, $\mu = 5$ | 6.45 | 18.42 | 27.10 | 5.70 | 16.93 | 24.84 |
| + Gram, $\mu = 50$ | 7.01 | 19.35 | 28.03 | 5.92 | 17.13 | 24.94 |
| + Gram, $\mu = 100$ | 6.91 | 19.50 | 28.16 | 6.05 | 17.37 | 25.30 |
| + TE$_1$ only | 10.10 | 25.97 | 35.76 | **7.33** | **20.22** | **28.59** |
| + TE$_2$ only | **10.32** | **26.44** | **36.43** | **7.33** | 20.03 | 28.46 |
| + TE$_1$ + TE$_2$ | 10.27 | 26.36 | 36.30 | 6.98 | 18.95 | 26.95 |

Table 18: Comparison with relational Gram geometry matching. The Gram baseline aligns within-batch teacher-student cosine similarity matrices for image, text, and joint image-text embeddings. Although Gram matching improves over the base CL+KL+L2+ICL objective, the TE-inspired finite-difference geometry objectives achieve stronger retrieval performance, especially for Image-to-Text retrieval.

**Results.** Table 18 compares the base objective, the relational Gram-matching baseline, and the TE-inspired variants. All experiments use $\alpha = 1$, $\beta = 50$, and $\delta = 1$. For Gram-matching experiments, we set $\gamma = 0$ and sweep $\mu$. For TE experiments, we set $\mu = 0$ and use $\gamma = 7.5$.

The results show that relational Gram matching provides modest improvements over the base objective. The best Gram setting improves Image-to-Text R@1 from 6.08% to 7.01%, R@5 from 18.14% to 19.50%, and R@10 from 26.93% to 28.16%. However, the TE-inspired objectives achieve substantially stronger performance. The full TE objective obtains 10.27%, 26.36%, and 36.30% on Image-to-Text R@1/R@5/R@10, respectively. TE$_1$ and TE$_2$ individually also outperform Gram matching, with TE$_2$ achieving the best Image-to-Text retrieval performance.

For Text-to-Image retrieval, the best Gram setting achieves 6.05%, 17.37%, and 25.30% on R@1/R@5/R@10. The full TE objective improves these results to 6.98%, 18.95%, and 26.95%, while the TE$_1$-only objective achieves the strongest Text-to-Image performance at 7.33%, 20.22%, and 28.59%. These results indicate that preserving static pairwise similarity geometry is helpful but does not fully account for the gains obtained by TE-inspired finite-difference alignment. The directional geometry captured by TE$_1$ and TE$_2$ provides a complementary and more effective teacher-student supervision signal for multimodal distillation.

## H.2 Isolating the Effect of the TE-Inspired Proxy

The full distillation objective combines several complementary losses, including contrastive learning, KL divergence, feature-level alignment, interactive contrastive learning, and the proposed TE-inspired proxy. To isolate the contribution of the TE-inspired proxy, we conduct a controlled ablation in which the base objective and its hyperparameters are fixed, and only the TE-related component is changed.

The base objective is defined as

$$\mathcal{L}_{\text{base}} = \mathcal{L}_{\text{contrastive}} + \alpha\mathcal{L}_{\text{KL}} + \beta\mathcal{L}_{\text{L2}} + \delta\mathcal{L}_{\text{ICL}}. \tag{76}$$

In all experiments in this subsection, we fix

$$\alpha = 1, \qquad \beta = 50, \qquad \delta = 1. \tag{77}$$

The baseline setting uses no TE-inspired proxy, corresponding to

$$\gamma = 0. \tag{78}$$

For the TE variants, we keep the same base objective and add only the TE-inspired term. For the TE1-only variant, the objective is

$$\mathcal{L}_{\text{total}} = \mathcal{L}_{\text{base}} - \gamma\text{TE}_1. \tag{79}$$

| | Image-to-Text Retrieval | | | Text-to-Image Retrieval | | |
|---|---|---|---|---|---|---|
| Method | R@1 | R@5 | R@10 | R@1 | R@5 | R@10 |
| Base: CL+KL+L2+ICL | 6.08 | 18.14 | 26.93 | 5.92 | 17.06 | 24.89 |
| + $TE_1$ only | 10.10 | 25.97 | 35.76 | **7.33** | **20.22** | **28.59** |
| + $TE_2$ only | **10.32** | **26.44** | **36.43** | **7.33** | 20.03 | 28.46 |
| + $TE_1 + TE_2$ | 10.27 | 26.36 | 36.30 | 6.98 | 18.95 | 26.95 |
| + image-side $TE_1$ only | 7.46 | 20.92 | 30.36 | 5.69 | 16.67 | 24.38 |
| + text-side $TE_2$ only | 7.49 | 20.27 | 28.96 | 7.03 | 19.40 | 27.60 |

Table 19: Ablation isolating the effect of the TE-inspired proxy. The base objective and its hyperparameters are fixed across all rows, with $\alpha = 1$, $\beta = 50$, and $\delta = 1$. The only change is whether the TE-inspired proxy is disabled, added as $TE_1$ only, added as $TE_2$ only, or added as the full $TE_1 + TE_2$ objective. The results show that the TE-inspired proxy provides substantial gains over the same base objective without changing the other loss weights.

For the TE2-only variant, the objective is

$$\mathcal{L}_{\text{total}} = \mathcal{L}_{\text{base}} - \gamma TE_2. \tag{80}$$

For the full TE objective, we use

$$\mathcal{L}_{\text{total}} = \mathcal{L}_{\text{base}} - \gamma \left( TE_1 + TE_2 \right). \tag{81}$$

We set $\gamma = 7.5$ for the TE variants. Therefore, the comparison isolates the effect of the TE-inspired proxy while keeping the remaining loss terms and hyperparameters unchanged.

Table 19 shows that adding the TE-inspired proxy substantially improves retrieval performance over the fixed base objective. The base CL+KL+L2+ICL objective obtains 6.08%, 18.14%, and 26.93% on Image-to-Text R@1/R@5/R@10, respectively. Adding $TE_1$ improves these results to 10.10%, 25.97%, and 35.76%, while adding $TE_2$ improves them to 10.32%, 26.44%, and 36.43%. The full $TE_1 + TE_2$ objective achieves 10.27%, 26.36%, and 36.30%.

For Text-to-Image retrieval, the base objective obtains 5.92%, 17.06%, and 24.89% on R@1/R@5/R@10. Adding $TE_1$ improves these results to 7.33%, 20.22%, and 28.59%, and adding $TE_2$ improves them to 7.33%, 20.03%, and 28.46%. The full $TE_1 + TE_2$ objective obtains 6.98%, 18.95%, and 26.95%.

The modality-specific ablations further show that the two TE components contribute differently. The image-side TE term improves Image-to-Text retrieval, while the text-side TE term provides stronger gains for Text-to-Image retrieval. These trends are consistent with the interpretation that the TE-inspired proxy transfers directional geometry in representation space, and that image and text modalities may affect the two retrieval directions differently. Overall, this ablation isolates the TE-inspired proxy from the other terms in the objective and shows that it is a major contributor to the observed retrieval gains.

### H.3 Pairing Strategy Ablation: Random Adjacent Pairs vs. Nearest-Neighbor Pairs

The proposed TE-inspired proxy uses finite differences between paired examples in a mini-batch. In the main implementation, we form these pairs by first randomly permuting the mini-batch and then taking adjacent examples in the permuted order. A reviewer raised the concern that such random adjacent pairs may not satisfy the small-perturbation motivation in Theorem 1 and may be disconnected from the local neighborhood structure analyzed by our kNN-based representation diagnostics. To address this concern, we add an ablation comparing the default random adjacent-pairing strategy with a nearest-neighbor (NN) pairing strategy.

For random adjacent-pairing, finite differences are formed as

$$\Delta f_i = f(x_{\pi(i+1)}) - f(x_{\pi(i)}), \tag{82}$$

Table 20: Ablation of finite-difference pairing strategies. Results are mean ± standard deviation over five random seeds. Random adjacent-pairing is the default strategy used in the main experiments. NN-pairing forms finite differences using nearest neighbors in the teacher embedding space.

| Pairing Strategy | Image-to-Text Retrieval | | | Text-to-Image Retrieval | | |
|---|---|---|---|---|---|---|
| | R@1 | R@5 | R@10 | R@1 | R@5 | R@10 |
| Random adjacent-pairing | $10.08 \pm 0.17$ | $26.10 \pm 0.19$ | $36.22 \pm 0.21$ | $7.12 \pm 0.19$ | $19.35 \pm 0.27$ | $27.85 \pm 0.23$ |
| NN-pairing | $10.35 \pm 0.16$ | $26.26 \pm 0.22$ | $36.41 \pm 0.27$ | $6.98 \pm 0.30$ | $19.16 \pm 0.26$ | $27.52 \pm 0.39$ |

where $\pi$ is a random permutation of the mini-batch. For NN-pairing, each example is paired with its nearest neighbor in the teacher embedding space:

$$j(i) = \arg\max_{j \neq i} \cos\big(z_T(x_i), z_T(x_j)\big), \tag{83}$$

and the finite difference is formed as

$$\Delta f_i = f(x_{j(i)}) - f(x_i). \tag{84}$$

This NN-pairing variant provides a more local finite-difference construction and is more directly aligned with the small-perturbation motivation behind Theorem 1 and the kNN-based representation analysis.

Table 20 reports five-seed results for the two pairing strategies under the same training objective and hyperparameter setting.

The two pairing strategies yield very similar performance. NN-pairing slightly improves Image-to-Text retrieval, while random adjacent-pairing slightly improves Text-to-Image retrieval. This suggests that the proposed TE-inspired proxy is not overly sensitive to the particular finite-difference pairing construction. We therefore interpret random adjacent-pairing as a lightweight stochastic finite-difference regularizer, rather than as a strict estimator of infinitesimal perturbations. NN-pairing provides a more local variant that better matches the small-perturbation intuition, but the comparable results indicate that the main benefit comes from teacher–student directional alignment in representation space rather than from a specific pairing heuristic.

### H.4 Teacher: RN50, Student: RN34, Dataset: Flick8k

We further evaluate our approach on the Flickr8k dataset (Marco et al., 2023), using 85% of the data for training and 15% for testing. Performance results for various loss functions are summarized in Table 21. The loss function employs weighting factors $\alpha = 1.0$, $\beta = 100$, $\delta = 1.0$, $\gamma = 5.0$, and a temperature parameter $\tau = 0.07$. These parameters were selected based on the relative contribution of each loss term to the total loss during training, ensuring balanced optimization. Given the modest size of Flickr8k, all experiments were conducted on a Google Colab instance equipped with an A100 GPU and limited system RAM. Each experiment (i.e., each row in Table 21) required approximately 20 minutes of training time. Notably, incorporating TE1 or TE2 into the loss function consistently improves both image-to-text (I2T) and text-to-image (T2I) retrieval performance compared to baselines that rely solely on standard distillation losses such as CL + KL or CL + MSE. These results underscore the effectiveness of transfer entropy approximations in guiding student model updates during distillation.

Table 22 reports the hyperparameter sensitivity of zero-shot retrieval when distilling a student RN34 from an RN50 teacher on Flickr8k. Overall, incorporating the TE reward has a clear and consistent impact: relative to the baseline without TE ($\gamma = 0$), increasing $\gamma$ improves both I2T and T2I performance, with the best overall results achieved at a moderate TE weight. In particular, $\gamma = 5$ yields the strongest bidirectional retrieval, attaining the highest I2T R@5/R@10 (65.65/77.84) and the best T2I R@1/R@5/R@10 (27.00/57.20/70.43), while slightly larger $\gamma$ (e.g., 7.5) maintains similar I2T R@1 but reduces T2I, indicating diminishing returns and mild over-regularization at high TE weights. Adjusting the other loss weights has comparatively smaller effects: increasing $\alpha$ to 5 or $\beta$ to 100 under $\gamma = 5$ produces performance close to, but not exceeding, the best setting, whereas increasing the ICL weight ($\delta = 5$) noticeably degrades retrieval, suggesting that

Table 21: Zero-shot retrieval performance (Recall@k) on **Flickr8k** of RN34-based VLM student using teacher CLIP RN50 under different loss functions. All Loss Function: CL + KL + MSE + ICL - TE1 - TE2. The weighting factors for loss functions: $\alpha = 1.0$, $\beta = 100$, $\delta = 1.0$, $\gamma = 5.0$.

| Model and Loss Function | I2T Retrieval (R) | | | T2I Retrieval (R) | | |
|---|---|---|---|---|---|---|
| | R@1 | R@5 | R@10 | R@1 | R@5 | R@10 |
| **Teacher Model (RN50)** | 51.65% | 78.17% | 87.73% | 47.28% | 75.21% | 84.60% |
| **Student Models (RN34)** | | | | | | |
| CL Only (Oord et al., 2018) | 22.73% | 48.19% | 60.87% | 18.47% | 43.76% | 56.77% |
| CL + MSE (Yang et al., 2024) | 22.98% | 49.92% | 62.52% | 17.84% | 44.71% | 57.99% |
| CL + KL (Li et al., 2024b) | 27.51% | 56.51% | 69.19% | 23.20% | 50.12% | 62.82% |
| CL + ICL (Yang et al., 2024) | 24.55% | 52.06% | 64.50% | 19.87% | 47.97% | 61.24% |
| CL - TE1 | 30.48% | 62.52% | 74.05% | 24.42% | 54.25% | 68.39% |
| CL - TE2 | 31.80% | 61.37% | 72.90% | **25.19%** | 54.66% | **68.70%** |
| CL - TE1 - TE2 | 32.29% | 62.36% | **75.29%** | 24.50% | 54.56% | 68.14% |
| CL + KL + MSE - TE1 | 31.22% | 59.31% | 71.99% | 23.64% | 53.20% | 67.27% |
| CL + KL + ICL - TE1 | 31.22% | 59.88% | 72.08% | 24.66% | 53.05% | 66.34% |
| CL + KL + MSE + ICL - TE1 | 30.23% | 58.81% | 72.24% | 24.18% | 53.16% | 66.64% |
| All Loss Function | **34.76%** | **63.43%** | 74.14% | 24.50% | **55.14%** | 68.29% |

overly strong interaction-level supervision can be harmful on this smaller dataset. Adding the MI term alone ($\eta \in \{1, 2.5, 5, 7.5\}$ with $\gamma = 0$) yields improvements over the baseline and can approach the best TE-regularized performance (e.g., $\eta = 5$ achieves I2T R@10 of 77.10 and T2I R@5 of 57.03), but combining MI and MSE-$\Delta$ with TE does not provide consistent additive gains within this sweep. Taken together, these results indicate that Flickr8k benefits most from moderate TE regularization (around $\gamma = 5$), while the overall objective remains relatively robust to moderate variations in the remaining hyperparameters.

## H.5 Teacher: RN50, Student Model: RN18

In addition to using RN34 as the student model, we also conduct experiments with RN18 as the student image encoder. The RN18 architecture is a more compact variant, containing approximately 11.7 million parameters (He et al., 2016). Similar to RN34, the final fully connected layer is modified to output 1024-dimensional features, keeping the overall parameter count stable. Given that the text encoder remains unchanged, the total number of parameters for the RN18-based student model is approximately 45-50 million. This reduction in model size compared to the RN34-based student allows for a more lightweight design while still leveraging the benefits of contrastive learning and effective knowledge transfer from the teacher model.

We applied the same loss components and hyperparameter settings as in Section 5: $\alpha = 1.0$, $\beta = 50$, $\delta = 1.0$, $\gamma = 1.0$, and a temperature parameter $\tau = 0.05$. Figure 11 presents the training losses and TE rewards over epochs for various configurations. Compared to RN34, RN18 exhibits a similar trend where the total training loss steadily decreases, and TE rewards increase over epochs, indicating effective optimization and knowledge transfer. However, due to the smaller capacity of RN18, the absolute TE rewards remain slightly lower than those observed for RN34, suggesting a less expressive feature alignment between teacher and student. Furthermore, the KL loss and MSE components show even less significant reductions over training epochs, likely due to the more limited representational capacity of RN18. This highlights that while TE-based regularization remains effective in guiding knowledge distillation, the overall learning dynamics are constrained by the smaller network size, making RN34 a more effective student model in terms of retaining structured alignment with the teacher.

We used Google Colab Pro with a T4 GPU and High-RAM for training and evaluating RN18. Due to its significantly fewer parameters compared to RN34, the student model RN18 required less training time. We trained it for 10 epochs in each loss function combination scenario, with the training and evaluation process taking approximately 11 hours per experimental setup.

Table 22: Comparison of zero-shot retrieval performance (Recall@k) of RN34-based VLM student with teacher CLIP RN50 on Flick8k.

| $\alpha$ | $\beta$ | $\delta$ | $\eta$ | $\lambda$ | $\gamma$ | I2T R@1 | I2T R@5 | I2T R@10 | T2I R@1 | T2I R@5 | T2I R@10 |
|---|---|---|---|---|---|---|---|---|---|---|---|
| 1 | 50 | 1 | 0 | 0 | 0 | 26.61% | 58.15% | 69.93% | 22.49% | 51.48% | 64.74% |
| 1 | 50 | 1 | 0 | 0 | 1 | 31.55% | 61.70% | 71.99% | 24.38% | 53.86% | 67.84% |
| 1 | 50 | 1 | 0 | 0 | 2.5 | 35.34% | 61.94% | 73.72% | 25.62% | 53.11% | 67.17% |
| 1 | 50 | 1 | 0 | 0 | 5 | 35.42% | **65.65%** | **77.84%** | **27.00%** | **57.20%** | **70.43%** |
| 1 | 50 | 1 | 0 | 0 | 7.5 | **35.50%** | 63.67% | 75.45% | 25.55% | 55.42% | 68.62% |
| 1 | 50 | 1 | 0 | 0 | 10 | 33.20% | 62.27% | 73.72% | 23.90% | 54.94% | 68.95% |
| 5 | 50 | 1 | 0 | 0 | 5 | 34.10% | 63.26% | 75.29% | 25.17% | 54.00% | 68.27% |
| 1 | 100 | 1 | 0 | 0 | 5 | 33.53% | 64.50% | 75.86% | 25.67% | 55.30% | 69.79% |
| 1 | 50 | 5 | 0 | 0 | 5 | 29.65% | 59.47% | 73.72% | 24.32% | 52.39% | 65.98% |
| 1 | 50 | 1 | 0 | 50 | 0 | 29.00% | 58.48% | 70.10% | 23.25% | 51.66% | 65.55% |
| 1 | 50 | 1 | 0 | 100 | 0 | 29.08% | 58.24% | 70.68% | 24.18% | 52.83% | 66.39% |
| 1 | 50 | 1 | 1 | 1 | 0 | 32.70% | 62.60% | 74.14% | 26.00% | 54.71% | 67.64% |
| 1 | 50 | 1 | 2.5 | 0 | 0 | 31.30% | 60.96% | 73.39% | 24.18% | 53.33% | 67.23% |
| 1 | 50 | 1 | 5 | 0 | 0 | 34.02% | 65.16% | 77.10% | 26.24% | 57.03% | 69.67% |
| 1 | 50 | 1 | 7.5 | 0 | 0 | 33.20% | 64.66% | 75.95% | 25.27% | 53.99% | 67.78% |
| 1 | 50 | 1 | 2.5 | 50 | 2.5 | 32.13% | 62.77% | 75.37% | 25.02% | 53.99% | 67.51% |
| 1 | 50 | 1 | 2.5 | 0 | 5 | 31.14% | 64.99% | 76.28% | 24.68% | 55.90% | 69.09% |
| 1 | 50 | 1 | 2.5 | 50 | 5 | 33.61% | 62.19% | 74.79% | 24.84% | 55.58% | 68.86% |
| 1 | 50 | 1 | 1 | 0 | 7.5 | 31.14% | 62.44% | 74.63% | 24.81% | 55.14% | 68.55% |
| 1 | 50 | 1 | 1 | 50 | 7.5 | 32.95% | 62.44% | 74.14% | 26.33% | 55.67% | 68.65% |
| 1 | 50 | 1 | 5 | 50 | 7.5 | 30.15% | 60.87% | 74.55% | 23.84% | 54.55% | 68.01% |

We summarize the zero-shot retrieval performance for the trained RN18 student model in Table 23. Similar observations we can make that the experiment with loss function (Contrastive - TE1 -TE2) achieved the best performance for Image-to-Text retrieval, while the experiment with loss function (Contrastive + KL + MSE + ICL - TE1 - TE2) achieved the best performance in Text-to-Image Retrieval. Comparing Table 23 with Table 1, the results indicate that while RN18 achieves competitive performance across different loss function combinations, it underperforms compared to RN34 for all loss configurations, with RN34 consistently yielding higher Recall@k values. However, the best-performing RN18 model (Contrastive - TE1 - TE2) achieves Recall@1 of 6.65% for image-to-text retrieval, which is not far behind RN34's highest Recall@1 values of 8.24% under the same loss formulation. This suggests that while RN18 is a lighter-weight alternative, RN34 remains a better choice for preserving retrieval performance during distillation. The trade-off between model complexity and retrieval accuracy highlights the importance of selecting an appropriate student architecture based on deployment constraints and performance requirements.

# I  Statement of Broader Impact

The development of efficient distillation frameworks for VLMs, such as our proposed TE-inspired proxy regularization approach, has the potential to yield several positive societal impacts. By enabling the deployment of compact yet high-performing multimodal models on resource-constrained devices, this work may broaden access to advanced AI technologies for individuals and communities with limited computational resources. Such efficiency gains could support wider adoption in applications such as assistive technologies for people with disabilities, low-cost language translation tools, educational platforms, and on-device multimodal interfaces. In addition, parameter-efficient VLM distillation can reduce energy consumption and the environmental footprint associated with large-scale model training and inference, thereby contributing to more sustainable AI deployment.

At the same time, broader deployment of efficient VLMs raises important risks. Increased accessibility to vision–language technology may amplify concerns related to privacy, surveillance, and misuse, including

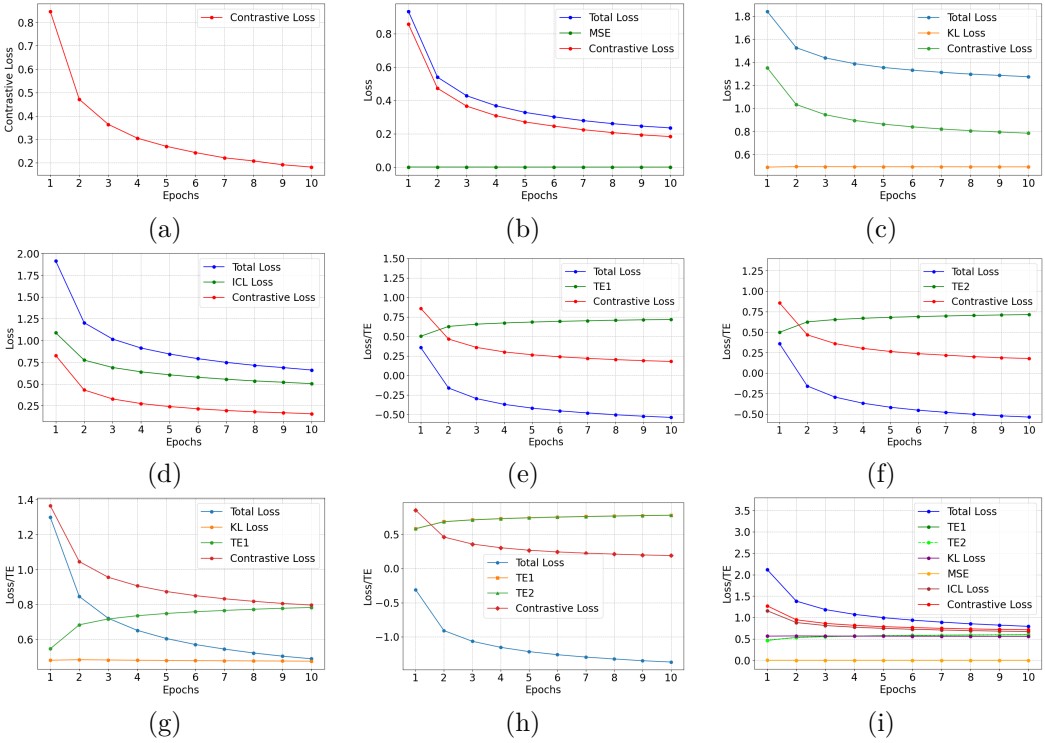

Figure 11: The training losses and TE for different loss functions in the training of RN18-based VLM Student with teacher CLIP RN50. (a) Contrastive only, (b) Contrastive + MSE, (c) Contrastive + KL, (d) Contrastive + ICL, (e) Contrastive - TE1, (f) Contrastive - TE2, (g) Contrastive + KL - TE1, (h) Contrastive - TE1 - TE2, (i) Contrastive + KL + ICL + MSE - TE1 - TE2.

Table 23: Comparison of zero-shot retrieval performance (Recall@k) of RN18-based VLM student with teacher CLIP RN50 for different loss function combinations in VLM distillation using MSCOCO. All Loss Function: CL + KL + MSE + ICL - TE1 - TE2.

| Model and Loss Function | I2T Retrieval (R) | | | T2I Retrieval (R) | | |
|---|---|---|---|---|---|---|
| | R@1 | R@5 | R@10 | R@1 | R@5 | R@10 |
| **Teacher Model (RN50)** | 15.27% | 30.73% | 39.05% | 11.68% | 25.52% | 33.50% |
| **Student Models (RN18)** | | | | | | |
| CL Only (Oord et al., 2018) | 4.38% | 13.28% | 20.40% | 3.39% | 11.07% | 17.22% |
| CL + MSE (Yang et al., 2024) | 4.27% | 13.29% | 20.15% | 3.47% | 11.17% | 17.28% |
| CL + KL (Li et al., 2024b) | 4.89% | 15.23% | 22.90% | 4.58% | 13.99% | 21.05% |
| CL + ICL (Yang et al., 2024) | 5.39% | 15.48% | 22.95% | 4.32% | 13.23% | 19.96% |
| CL - TE1 | 5.48% | 16.43% | 24.59% | 4.60% | 13.86% | 20.80% |
| CL - TE2 | 5.57% | 16.67% | 24.78% | 4.67% | 14.08% | 20.97% |
| CL - TE1 - TE2 | **6.65%** | **18.75%** | **27.33%** | 5.18% | 15.09% | 22.35% |
| CL + KL - TE1 | 6.49% | 18.37% | 26.83% | 5.18% | 15.17% | 22.52% |
| All Loss Function | 6.52% | 18.60% | 27.16% | **5.79%** | **16.78%** | **24.47%** |

unauthorized content analysis, large-scale monitoring, or the generation of misleading multimodal content. Moreover, distillation can propagate or even amplify biases already present in the teacher model, potentially leading to unfair or discriminatory outcomes for underrepresented groups. Compression may also remove subtle but important information, which could degrade robustness, calibration, or fairness in real-world settings, especially when models are deployed under distribution shift.

55

To help mitigate these risks, we advocate several practical safeguards. First, bias and fairness audits should be incorporated throughout both teacher evaluation and student distillation, particularly for sensitive or high-impact applications. Second, strong data governance and privacy-preserving practices are important when training or deploying VLMs on user-facing or sensitive data. Third, transparency should be maintained through model cards, clear reporting of evaluation settings and limitations, and documentation of the distillation pipeline. Finally, interdisciplinary collaboration with ethicists, domain experts, and affected stakeholders can help identify harms early and guide more responsible deployment. By proactively addressing these concerns, TE-inspired VLM distillation may be developed and applied in a safer, more equitable, and more socially beneficial manner.

