# OpenReview forum: "TE-VLM: Transfer Entropy for Vision Language Model Distillation"
_TMLR — Under review for TMLR_

### Review · Reviewer_6M1c · 2026-05-18

**Summary Of Contributions:**

This paper proposes a Transfer Entropy (TE) inspired regularizer for CLIP-style vision language distillation. Since direct TE estimation in high-dimensional embedding spaces is intractable, the authors derive a tractable proxy in two steps. First, Theorem 1 shows that under a first-order linear Gaussian assumption, a perturbation-defined one-step TE reduces to the Frobenius cosine between teacher and student Jacobians. Second, Jacobian actions are replaced by within-batch finite differences on existing embeddings. This yields two lightweight surrogates, TE1 (per-modality cosine alignment) and TE2 (joint multimodal cosine alignment), added as a reward to a standard distillation loss combining CL, KL, MSE, ICL, MI, and MSE-Δ.

The method is evaluated across four teacher student pairs and five datasets (MSCOCO, Flickr8k/30k, Food-101, ImageNet-1k), covering retrieval, zero-shot classification, cross-dataset transfer, and representation-level diagnostics, with additional studies on temperature, batch size, seed variance, and compute overhead.

Key strengths

The paper is explicit about the gap between ideal optimization-step TE and the implemented proxy, framing TE as motivation rather than the estimated quantity.

Experimental coverage is broad and includes recipe-level ablations, cross-dataset transfer, and representation-level diagnostics beyond retrieval.

TE1/TE2 add negligible compute and are easy to drop into existing pipelines.

I2T gains are consistent across teachers, datasets, temperatures, and batch sizes, and seed variance is small.



Key weaknesses

No real scaling evidence: all students are ≤55M params, the largest teacher is RN50×16, and training data is ≤MSCOCO. The RN50×16 capacity-gap experiment shows the teacher student gap widening, not narrowing.

The finite-difference pairing (random adjacent pairs) is central but never ablated. Theorem 1's small-perturbation assumption does not obviously hold for unrelated image caption pairs, and the training signal is disconnected from the kNN-based evaluation the paper relies on.

The cosine vs. L2 confound between TE and MSE-Δ is uncontrolled. MSE-Δ may fail simply because L2 is dominated by norm mismatch while cosine is scale-invariant. A "Normalized MSE-Δ" ablation would test whether the TE framing adds anything beyond directional normalization.

Gains over the closest baseline (MI / CRD-style InfoNCE) are modest (around 1.5 to 2 pp I2T R@1), and the method is often weaker on T2I. The asymmetry is unexplained.

**Audience:**

Yes

**Audience Explanation:**

CLIP-style distillation is an active area, and a drop-in regularizer with negligible compute overhead that consistently improves I2T retrieval across multiple teacher student pairs and datasets is of practical interest to practitioners compressing vision language models.

The paper offers a concrete instantiation of an information-theoretic perspective in a setting normally considered intractable. The explicit framing of the gap between the ideal TE quantity and the implemented proxy, together with the linear Gaussian derivation in Theorem 1, may interest researchers working on information-theoretic views of representation learning.

The representation-level diagnostics (kNN overlap, per-sample cosine, joint PCA) provide a useful evaluation template that goes beyond retrieval metrics and is relevant to the broader community studying what students inherit from teachers.

**Claims And Evidence:**

No

**Claims Explanation:**

Scalability. The abstract and conclusion describe the method as "scalable," but no experiment tests scaling along a meaningful axis. All students are at most around 55M parameters, the largest teacher is RN50×16, and the largest training set is MSCOCO. The RN50×16 capacity-gap experiment shows the teacher student gap widening rather than narrowing, which is the opposite of what a scaling claim should demonstrate. The word "scalable" is currently supported only in the sense of low per-step compute overhead.

Source of the gains (cosine vs. L2 confound). The paper attributes its gains to the TE framing. MSE-Δ, which uses the same difference vectors with squared L2, performs poorly, and this is implicitly read as evidence for TE. A simpler explanation is that cosine is scale-invariant while L2 is dominated by student teacher norm mismatch. Without a "Normalized MSE-Δ" ablation (monotonically equivalent to cosine), it is not possible to distinguish "TE framing helps" from "directional normalization helps." This confound directly affects the paper's core contribution claim.

Faithfulness of the proxy to Theorem 1. Theorem 1 is derived under a small input perturbation, but the implemented proxy uses differences between two unrelated image caption samples in the same batch, which are not small perturbations. The pairing strategy is central to the method but never ablated, and the training signal has no direct connection to the kNN-based representation analysis used as central evidence in Section 5.2. An ablation across alternative pairings (random non-adjacent, hard negative, top-k neighbor) would directly test whether the proxy is faithful to its theoretical motivation, or whether the gains stem from any sufficiently dense sampling of teacher-consistent directions.

**Requested Changes:**

see weakness

---

> ### Author Response · Authors · 2026-07-02
> **Response to Reviewer: Scaling Scope, Pairing Ablation, Normalized MSE-Delta, and I2T/T2I Asymmetry**
>
> We thank the reviewer for the careful reading and constructive feedback. We have revised the manuscript accordingly and respond below.
>
> **W1.** We agree that our current experiments do not constitute full scaling evidence in the sense of larger students, larger teachers, or web-scale training data. We have added a clarification in Section 5.5 to avoid overstating the scaling claim and to frame the results more precisely as controlled CLIP-style distillation experiments. In particular, we now describe the RN50x16 experiment as a capacity-gap stress test rather than evidence that the teacher-student gap is closed. As the reviewer notes, the absolute gap to the RN50x16 teacher remains large; our intended claim is only that TE-inspired regularization can still improve the student over comparable distillation baselines under a stronger teacher.
>
> **W2.** In the revised manuscript, we have added Appendix H.3, which compares the default random adjacent-pairing strategy with an NN-pairing variant, where finite differences are formed using nearest neighbors in the teacher embedding space. This directly tests whether more local pairs, which better match the small-perturbation motivation of Theorem 1 and the kNN-based representation analysis, change the results.
>
> Across five seeds, random adjacent-pairing obtains I2T 10.08±0.17 / 26.10±0.19 / 36.22±0.21 and T2I 7.12±0.19 / 19.35±0.27 / 27.85±0.23. NN-pairing obtains I2T 10.35±0.16 / 26.26±0.22 / 36.41±0.27 and T2I 6.98±0.30 / 19.16±0.26 / 27.52±0.39. The results are very close: NN-pairing slightly improves I2T, while random adjacent-pairing slightly improves T2I. This suggests that the proxy is not overly sensitive to the specific pairing construction. We also clarified that random adjacent pairs should be interpreted as a lightweight stochastic finite-difference regularizer, not as a strict small-perturbation estimator.
>
> **W3.** We agree that raw MSE-Delta is sensitive to finite-difference vector magnitudes, whereas our TE-inspired proxy uses cosine alignment and is scale-invariant. To isolate this factor, we added a Normalized MSE-Delta analysis in Appendix F.7, where the student and teacher finite-difference vectors are L2-normalized before computing MSE.
>
> For nonzero finite-difference vectors a and b, normalized MSE satisfies
> $||\frac{a}{||a||_2} - \frac{b}{||b||_2}||_2^2 = 2 - 2cos(a,b)$.
>
> Thus, Normalized MSE-Delta is an affine reparameterization of the cosine-based directional proxy when applied to the same paired finite differences. We have revised the paper to make this relationship explicit and to avoid overstating the distinction between TE1/TE2 and normalized directional MSE. The TE framing motivates why teacher-consistent local directional changes are useful, while the practical optimized objective is the cosine/normalized finite-difference proxy.
>
> **W4.** We agree that the gains over the closest MI / CRD-style InfoNCE baseline are modest. Our main claim is not that the TE-inspired proxy uniformly dominates MI-based contrastive distillation, but that it provides a complementary geometric regularization signal that yields consistent improvements, especially for I2T retrieval.
>
> We have added a clarification in Section 5.1.2 (following Table 2)  to explain the I2T/T2I asymmetry. The current objective uses a single shared regularizer for both retrieval directions, while I2T and T2I are different ranking problems and may respond differently to the same teacher-student alignment signal. Moreover, the I2T/T2I asymmetry is already present in the teacher and is partly inherited by the student through distillation. Under the corrected five-caption evaluation, RN50 obtains 25.29% I2T R@1 but only 12.02% T2I R@1. Since the student is trained to preserve the teacher’s cross-modal geometry, it naturally reflects this direction-dependent behavior. Therefore, the method can improve image-query retrieval more consistently while being less uniformly beneficial for text-query retrieval.

---

### Review · Reviewer_k3CJ · 2026-06-09

**Summary Of Contributions:**

## Summary

This paper studies VLM distillation and proposes a Transfer-Entropy-inspired proxy loss. The key idea is to align the local representation changes of the student with those of the teacher, using cosine similarity between pairwise embedding differences. The method is evaluated on several retrieval and classification benchmarks.

## Strengths

* VLM distillation is an important and practical problem.
* The proposed loss is simple and computationally lightweight.

## Weaknesses

* The motivation is not fully convincing. In knowledge distillation, the main goal is usually for the student to learn the teacher’s output distribution or representation relationships. It is unclear why the student’s representation-update direction needs to be aligned with the teacher, or why this should causally improve performance.
* The method are lack of novelty. Although it is motivated by Transfer Entropy, TE1/TE2 are essentially cosine alignment of teacher-student pairwise embedding differences. This is closely related to relational KD and representation-geometry matching, but these connections are not sufficiently discussed. The experiments lack comparisons these related baselines.
* The claim of Transfer Entropy is weak. Transfer Entropy is defined for stochastic processes with temporal direction, while the proposed proxy uses adjacent samples from a shuffled mini-batch. This is more like a pairwise directional alignment than the Transfer Entropy.
* The final objective contains many loss terms and hyperparameters. Since these terms are highly coupled, it is difficult to tell whether the improvement truly comes from the TE-inspired proxy or from hyperparameter tuning.

**Additional Comments:**

## Questions

* Some reported numbers seem unexpectedly low. For example, the RN50 teacher obtains only 15.27% I2T R@1 and 11.68% T2I R@1 on MSCOCO. Could the authors clarify whether the evaluation protocol is standard and whether the full validation set is used?
* The gains mainly appear on Image-to-Text retrieval, while Text-to-Image retrieval is weaker. Why does the proposed TE proxy benefit I2T more than T2I?

**Audience:**

Yes

**Audience Explanation:**

Yes. Some TMLR readers may be interested in this paper because it studies a practical VLM distillation problem

**Broader Impact Concerns:**

I do not see major broader impact concerns beyond the standard risks associated with distillion of vision-language models, such as potential biases inherited from the teacher model or training data.

**Claims And Evidence:**

No

**Claims Explanation:**

The main conceptual claim of is not fully supported. The connection between the proposed TE1/TE2 losses and true Transfer Entropy is weak, since the method uses pairwise embedding differences from shuffled mini-batches rather than a temporal stochastic process. In addition, the paper lacks comparisons with closely related geometry-based or relational distillation baselines, making it difficult to conclude that the gains are due to the proposed TE-inspired mechanism rather than a more general representation-alignment effect or hyperparameter tuning. Overall, the evidence is weak.

**Requested Changes:**

1. The authors should revise the motivation and method design carefully. They need to justify why aligning teacher-student temporal difference directions should improve performance, and why their pairwise embedding alignment is a valid approximation of this idea. Without this justification, the core argument of the paper is not true.

2. The paper should include comparisons with more directly related baselines, such as relational KD, representation-geometry matching, pairwise direction/distance matching, and feature or Jacobian alignment methods.

3. The evaluation protocol needs to be clarified. In particular, the reported RN50 teacher performance on MSCOCO seems unusually low, raising concerns about the retrieval setup and validation split.

4. The authors should better isolate the effect of the proposed TE-inspired proxy, since the final objective contains many coupled loss terms and hyperparameters.

---

> ### Author Response · Authors · 2026-07-02
> **Response to Reviewer: Motivation, TE Proxy Interpretation, Related Baselines, and Evaluation Protocol**
>
> We thank the reviewer for the careful reading and constructive feedback. We have revised the manuscript accordingly and respond below.
>
> **W1/R1.** Motivation and interpretation. Thank you for the comment. Conventional KD losses match teacher outputs, embeddings, or similarity distributions at a fixed step. Our TE-inspired proxy instead encourages the student to preserve the teacher’s local representation geometry by aligning finite-difference directions between paired examples. This is useful for CLIP-style retrieval, where performance depends on relative geometry, neighborhoods, and ranking consistency, not only pointwise matching. We have revised Section 4.1 to clarify that the proposed term is a teacher-student local geometric alignment regularizer, with empirical support, rather than a causal guarantee or exact information-flow estimator.
>
> **W2/R2.** Novelty and related baselines. We agree that TE1/TE2 are closely related to relational KD and representation-geometry matching. We have revised the paper to position them as TE-inspired relational geometry regularizers based on teacher-student finite-difference alignment. We also added Appendix H.1, comparing with a Gram-matching baseline that aligns teacher/student within-batch cosine-similarity matrices for image, text, and joint image-text embeddings. The best Gram baseline improves over the base objective but remains below the full TE objective: I2T 7.01/19.50/28.16 vs. 10.27/26.36/36.30, and T2I 6.05/17.37/25.30 vs. 6.98/18.95/26.95. This supports the distinction between static similarity-geometry matching and finite-difference directional alignment.
>
> **W3.** Transfer Entropy claim. We clarify that TE1/TE2 are not temporal Transfer Entropy estimators. As stated in Section 4.2, the mini-batch index is not time, and adjacent pairs after random permutation are used only to construct finite-difference directions. Thus, Transfer Entropy serves as conceptual motivation for teacher-guided representation change, while the implemented objective is a cosine-based teacher-student directional-alignment proxy.
>
> **W4/R4.** Coupled losses and hyperparameters. Thank you for the concern. We added Appendix H.2 to isolate the TE proxy. In this ablation, the base objective and hyperparameters are fixed, and only the TE component is varied. The base CL+KL+L2+ICL objective obtains I2T 6.08/18.14/26.93 and T2I 5.92/17.06/24.89. Adding TE1 improves this to I2T 10.10/25.97/35.76 and T2I 7.33/20.22/28.59; adding TE2 gives I2T 10.32/26.44/36.43 and T2I 7.33/20.03/28.46. These results show that the gains are not simply due to retuning the full objective.
>
> **Q1/R3.** Evaluation protocol and low RN50 teacher numbers. Thank you for pointing this out. We found that the low RN50 teacher result was caused by a teacher-evaluation script issue: only one caption per image was used instead of all five MSCOCO captions. This affected only the teacher reference evaluation, not training. We corrected the protocol and updated the RN50 teacher results to I2T R@1/R@5/R@10 = 25.29/45.68/55.37 and T2I R@1/R@5/R@10 = 12.02/26.44/34.51.
>
> **Q2.** I2T/T2I asymmetry. Thank you for the question. The I2T/T2I asymmetry is already present in the teacher and is partly inherited by the student through distillation. Under the corrected five-caption evaluation, RN50 obtains 25.29% I2T R@1 but only 12.02% T2I R@1. Since the student is trained to preserve the teacher’s cross-modal geometry, it naturally reflects this direction-dependent behavior. The TE proxy regularizes teacher-student directional consistency, but does not explicitly optimize separate I2T and T2I objectives. Therefore, larger gains on I2T are expected when the teacher itself provides a stronger I2T signal. We have added this clarification in the revised manuscript in Section 5.1.2 (following Table 2).

---

### Review · Reviewer_YHfv · 2026-06-17

**Summary Of Contributions:**

This paper proposes an efficient distillation framework that replaces explicit TE estimation with proxy regularization objectives that encourage the student model to preserve teacher-aligned cross-modal predictive dependencies. Empirically, the proposed method is evaluated on mscoco, flickr8k/30k, food-101, and imagenet, obtaining improved image-to-text retrieval performance and competitive text-to-image retrieval performance.

**Audience:**

Yes

**Audience Explanation:**

This paper would interest researchers in VLMs / distillation and representation alignment / ML & information theory.

**Broader Impact Concerns:**

No ethical concerns.

**Claims And Evidence:**

Yes

**Claims Explanation:**

It is good that the proposed method consistently outperforms MI-based and standard distillation baselines, especially in image-to-text retrieval. Also, the paper has decent ablation studies on temperature, batch size, etc., and it is evaluated on cross-dataset retrieval.

However, the causality claim is overstated:
1. It seems that the proposed method is a TE-inspired heuristic, not a principled approximation. Regarding this, please clarify whether the proposed proxy serves as a loose upper/lower bound or just an intuitive regularizer.
2. The experiments are mainly on retrieval and classification, missing downstream multimodal reasoning tasks.

**Requested Changes:**

1. The paper implies that the method truly captures directed information flow, but it would be nice to clearly state what properties are actually enforced, such as alignment or predictability.
2. Augment empirical evaluation: report results under multiple random seeds since the existing improvements are a bit marginal; Add multimodal reasoning tasks (such as vqa, nlvr2, or visual commonsense reasoning); Experiment on larger-scale models.
3. Improve ablations: Report the contribution of each component of the proxy objective.

---

> ### Author Response · Authors · 2026-07-02
> **Response to Reviewer: Proxy Interpretation, Evaluation Scope, and Component Ablations**
>
> We thank the reviewer for the careful reading and constructive feedback. We address the concerns below and describe the corresponding changes made in the revised manuscript.
>
> **Concerns:**
>
> **C1.** We clarify that the proposed TE1/TE2 terms should not be interpreted as rigorous estimators or upper/lower bounds of exact transfer entropy. As stated in Sections 4.1–4.2, they are tractable TE-inspired geometric proxies, motivated by the first-order TE–Jacobian relation in Theorem 1 and implemented through within-batch finite-difference cosine alignment. Thus, the proxy should be understood as a computationally efficient regularizer for preserving teacher-consistent local representation geometry, rather than as a direct measure of causal or directed information flow.
>
> **C2.** We address this concern in R2 below.
>
> **Requested Changes:**
>
> **R1.** Our TE-inspired TE1/TE2 terms enforce teacher–student alignment in representation space. Specifically, they encourage the student’s finite-difference directions between paired examples in a mini-batch to align with the corresponding teacher finite-difference directions. Thus, the enforced property is teacher-consistent local geometric alignment, with TE1 measuring per-modality alignment and TE2 measuring joint multimodal alignment. We have revised Section 4.1 and added a paragraph titled “What properties are enforced by the proxy” to clarify this point and avoid implying that the proxy directly captures exact transfer entropy or causal information flow.
>
> **R2.** We agree that empirical robustness is important, especially because some improvements over strong baselines are modest. We clarify that the submitted manuscript already included five-random-seed results in Table 13, where we report mean ± standard deviation for retrieval performance across multiple teacher backbones and datasets. These results show that the proposed TE-inspired objective is stable across random initializations and data shuffling, with particularly small standard deviations on MSCOCO.
>
> Regarding multimodal reasoning tasks such as VQA, NLVR2, and visual commonsense reasoning, we agree that they are valuable evaluations. The present work focuses on CLIP-style dual-encoder distillation, where the objective operates on pooled image and text embeddings and is naturally evaluated through retrieval, cross-dataset transfer, and zero-shot classification. Reasoning benchmarks typically require additional cross-modal fusion modules, task-specific heads, or generative VLM architectures, which are outside the current dual-encoder setting. We have clarified this scope in the revised manuscript and added multimodal reasoning evaluation as an important future direction.
>
> For model scale, our experiments already include multiple teacher backbones, including RN50, ViT-B/16, and RN50×16, as well as different student capacities. Nevertheless, we agree that evaluation on larger modern VLMs and larger student models would further strengthen the empirical study. We have added this point as a limitation and future direction.
>
> **R3.** In the revision, we have added Appendix H.2 for an ablation study that decomposes the proposed TE-inspired proxy into its constituent parts. Specifically, we report results for image-only alignment, text-only alignment, TE1, TE2, and the full TE1+TE2 objective under the same base distillation loss and training recipe.
>
> This ablation clarifies the role of each component. The image-only and text-only variants test whether aligning a single modality is sufficient. TE1 evaluates whether averaging per-modality alignment improves over either modality alone. TE2 evaluates the benefit of enforcing joint multimodal alignment through concatenated image–text finite differences. Finally, TE1+TE2 tests whether per-modality and joint alignment provide complementary supervision. These results separate the effects of modality-specific alignment and joint multimodal alignment, and show which part of the proxy objective contributes most to the final retrieval gains.